# Selective Task Group Updates for Multi-Task Optimization

**Wooseong Jeong & Kuk-Jin Yoon**
Korea Advanced Institute of Science and Technology
{stk14570, kjyoon}@kaist.ac.kr

## Abstract

Multi-task learning enables the acquisition of task-generic knowledge by training multiple tasks within a unified architecture. However, training all tasks together in a single architecture can lead to performance degradation, known as negative transfer, which is a main concern in multi-task learning. Previous works have addressed this issue by optimizing the multi-task network through gradient manipulation or weighted loss adjustments. However, their optimization strategy focuses on addressing task imbalance in shared parameters, neglecting the learning of task-specific parameters. As a result, they show limitations in mitigating negative transfer, since the learning of shared space and task-specific information influences each other during optimization. To address this, we propose a different approach to enhance multi-task performance by selectively grouping tasks and updating them for each batch during optimization. We introduce an algorithm that adaptively determines how to effectively group tasks and update them during the learning process. To track inter-task relations and optimize multi-task networks simultaneously, we propose proximal inter-task affinity, which can be measured during the optimization process. We provide a theoretical analysis on how dividing tasks into multiple groups and updating them sequentially significantly affects multi-task performance by enhancing the learning of task-specific parameters. Our methods substantially outperform previous multi-task optimization approaches and are scalable to different architectures and various numbers of tasks.

## 1 Introduction

Multi-task learning (MTL) stands out as a key approach for crafting efficient and robust deep learning models that can adeptly manage numerous tasks within a unified architecture (Caruana, 1997). By training related tasks within a single network, MTL facilitates the acquisition of universal knowledge spanning multiple tasks, thereby enhancing generalization and accelerating convergence. Additionally, MTL reduces the need for expensive computing and storage resources by employing a shared network across tasks, making it a favorable choice for future generalized networks across many applications.

The goal of MTL is to mitigate negative transfer (Crawshaw, 2020) between tasks. Since each task possesses its own distinct objective function, improving performance in one task can potentially impede the performance of others. This phenomenon, characterized by a trade-off among tasks' performances, is referred to as negative transfer. To mitigate negative transfer during optimization, task-specific gradients are manipulated (Désidéri, 2012; Sener & Koltun, 2018; Yu et al., 2020; Liu et al., 2021a;b; Navon et al., 2022; Senushkin et al., 2023), tasks' losses are adaptively weighted (Kendall et al., 2018; Guo et al., 2018; Liu et al., 2019; 2024), or a combination of both approaches is used (Liu et al., 2021b). Previous research focused on balancing the influence of different tasks by considering unbalance in gradients of shared parameters. However, these analyses overlooked the role of task-specific parameters. Since the learning of shared space and task-specific information influence each other during optimization, a balanced shared space across tasks cannot be achieved without learning task-specific information in the task-specific parameters.

Therefore, we take a fundamentally different approach to mitigate negative transfer in MTL (see Figure 1). Our experiments reveal a notable difference in multi-task performance depending on whether task losses are updated collectively or sequentially. By grouping tasks and updating them

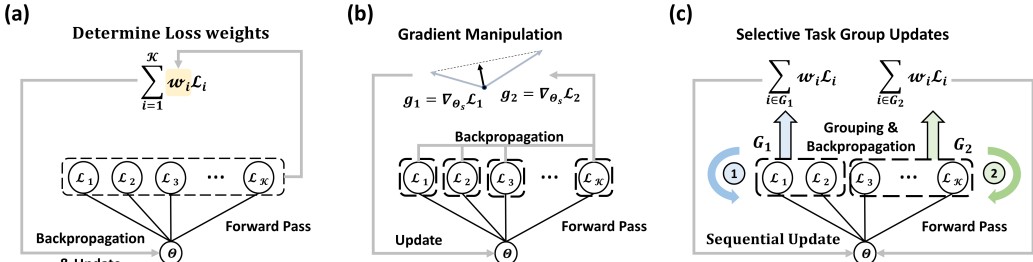

Figure 1: Comparison of multi-task optimization methods. $\Theta$ represents the network parameters, and $\{\mathcal{L}\}_{i=1}^{\mathcal{K}}$ denotes the task-specific losses for $\mathcal{K}$ tasks. (a) Loss-based approaches balance the loss by adjusting the weights $\{w_i\}_{i=1}^{\mathcal{K}}$ during optimization. (b) Gradient-based approaches modify the task-specific gradients $\{g_i\}_{i=1}^{\mathcal{K}}$ with respect to $\Theta$. (c) Our method divides the tasks into $\mathcal{M}$ groups (in this case, $\mathcal{M} = 2$) and updates them sequentially for each batch during optimization.

sequentially, the network can focus on specific task groups in turn, facilitating the learning of task-specific parameters. Thus, our main objective is to identify optimal strategies for grouping and updating tasks during optimization. Inter-task affinity (Fifty et al., 2021), which evaluates the loss change of one task when updating another task's gradients in the shared parameters, is a useful metric for understanding task relations. However, directly incorporating inter-task affinity into optimization presents several challenges: (i) It only considers updates to shared parameters, ignoring the influence of task-specific parameter updates. (ii) Inter-task affinity varies significantly throughout optimization, so relying on the average across time steps may not fully capture task relations. (iii) Measuring inter-task affinity is computationally intensive because it involves recursive gradient updates, tracking changes in loss, and reverting to previous parameter states. To address these issues, we introduce proximal inter-task affinity (Section 3.1), which concurrently explains the updates of shared and task-specific parameters. This metric can be tracked over time to approximate inter-task relations during optimization without significant computational overhead.

By capturing task relations, we introduce an algorithm that dynamically updates task groups using proximal inter-task affinity to effectively mitigate negative transfer between tasks (Section 3.2). Additionally, we present practical methods for inferring proximal inter-task affinity during task group updates. In the theoretical analysis (Section 4), we explain how this sequential update strategy can improve multi-task performance from an optimization standpoint. Previous approaches have demonstrated convergence to Pareto-stationary points, where the sum of all task-specific gradients equals zero (Désidéri, 2012; Sener & Koltun, 2018; Yu et al., 2020; Liu et al., 2021a;b; Navon et al., 2022; Senushkin et al., 2023). However, conventional convergence analysis cannot explain why sequential updates yield better multi-task performance with comparable stability, as they handle the learning of shared and task-specific parameters separately. Instead, we provide a theoretical explanation of the relations between the updates of shared and task-specific parameters and how the update sequence affects this. This perspective is not addressed in traditional multi-task optimization, which typically deals with the learning of shared and task-specific parameters independently. As a result, our approach facilitates the learning of task-specific parameters with stability similar to other optimization methods, while achieving faster convergence compared to previous gradient-based approaches. Experimental results demonstrate that adaptively partitioning the task set and sequentially updating them during the optimization process significantly enhances multi-task performance compared to previous works. Our main contributions are threefold:

- We propose an algorithm that dynamically groups tasks for updating each batch during multi-task optimization. To do this, we introduce proximal inter-task affinity, which we can track throughout the optimization process to group tasks accordingly.

- We provide a theoretical explanation of the advantages of sequentially updating task groups based on proximal inter-task affinity. We discuss how sequential updates can enhance the learning of task-specific parameters and improve multi-task performance, a result that traditional multi-task optimization analyses do not explain.

- Our methods demonstrate superior performance across various benchmarks when compared to previous multi-task optimization techniques, including both loss-based and gradient-based approaches.

## 2    RELATED WORK

Optimization methods in MTL can be broadly categorized into those that manipulate task-specific gradients (Chen et al., 2018; 2020; Javaloy & Valera, 2021; Désidéri, 2012; Sener & Koltun, 2018; Yu et al., 2020; Liu et al., 2021a;b; Phan et al., 2022; Navon et al., 2022; Senushkin et al., 2023; Jeong & Yoon, 2024) and those that adjust the weighting of task-specific losses (Kendall et al., 2018; Guo et al., 2018; Liu et al., 2019; 2024). Addressing the imbalance in task influence, normalized gradients are utilized (Chen et al., 2018) to prevent task spillover. Introducing stochasticity to the network's parameters based on gradient consistency, GradDrop (Chen et al., 2020) drops gradients. RotoGrad (Javaloy & Valera, 2021) rotates the network's feature space to align tasks. MGDA (Sener & Koltun, 2018) treats MTL as a multi-objective problem, minimizing the norm point in the convex hull. CAGrad (Liu et al., 2021a) minimizes multiple loss functions and regulates trajectory using worst local improvements of individual tasks. Aligned-MTL (Senushkin et al., 2023) stabilizes optimization by aligning principal components of gradient matrix . The weighting of losses significantly impacts multi-task performance as tasks with higher losses can dominate training. Addressing this, tasks' losses are weighted based on task-dependent uncertainty (Kendall et al., 2018). Prioritizing tasks according to difficulty by evaluating validation results (Guo et al., 2018), or balancing multi-task loss reflecting loss descent rates (Liu et al., 2019), are also proposed strategies. Previous works on multi-task optimization that directly modify task-specific gradients Chen et al. (2018; 2020); Javaloy & Valera (2021); Désidéri (2012); Sener & Koltun (2018); Yu et al. (2020); Liu et al. (2021a;b); Phan et al. (2022); Navon et al. (2022); Senushkin et al. (2023); Jeong & Yoon (2024) focus on learning the shared parameters of the network without considering their interaction with task-specific parameters during the optimization step. In contrast, we propose fundamentally different strategies to mitigate negative transfer by sequentially updating task groups which lead to better multi-task performance, a phenomenon not explained by conventional multi-task optimization analysis.

## 3    SELECTIVE TASK GROUP UPDATES FOR MULTI-TASK OPTIMIZATION

### 3.1    THE GOAL OF MULTI-TASK LEARNING AND INTER-TASK AFFINITY

In the context of MTL, we deal with multiple tasks with its own objective function denoted as $\{\mathcal{L}_i\}_{i=1}^{\mathcal{K}}$, where $\mathcal{K}$ represents the number of tasks. The network parameters $\Theta$ are partitioned into $\{\Theta_s, \Theta_1, \Theta_2, \ldots, \Theta_{\mathcal{K}}\}$, where $\Theta_s$ refers to the shared parameters while $\Theta_i$ represents the task-specific parameters for the task $i$. To simplify the discussion, we assumed that $\Theta_s$ is shared across all tasks. However, the following discussion can still apply even when it is only partially shared for specific task sets. The goal of MTL is to find Pareto-optimal parameters (Sener & Koltun, 2018), denoted as $\Theta^*$, that minimize the weighted sum of task-specific losses. This can be expressed as $\Theta^* = \arg\min_{\Theta} \sum_{i=1}^{\mathcal{K}} w_i \mathcal{L}_i(\Theta_s, \Theta_i)$, where $w_i$ represents the weight for the loss of task $i$.

The concept of inter-task affinity (Fifty et al., 2021) aims to identify tasks that enhance each other's performance when learned together on shared networks. It measures the extent to which the gradient of a specific task with respect to the shared parameters $\Theta_s$, affects the loss of other tasks.

**Definition 1** (Inter-Task Affinity). *Consider a multi-task network shared by tasks $i$ and $k$. For a data sample $z^t$ and a learning rate $\eta$, the task-specific gradients from $\mathcal{L}_i$ are applied to update the shared parameters of the network as follows: $\Theta_{s|i}^{t+1} = \Theta_s^t - \eta \nabla_{\Theta_s^t} \mathcal{L}_i(z^t, \Theta_s^t, \Theta_i^t)$. The inter-task affinity from task $i$ to task $k$ at time step $t$ is then defined as:*

$$\mathcal{A}_{i \to k}^t = 1 - \frac{\mathcal{L}_k(z^t, \Theta_{s|i}^{t+1}, \Theta_k^t)}{\mathcal{L}_k(z^t, \Theta_s^t, \Theta_k^t)} \tag{1}$$

In the context of task grouping (Fifty et al., 2021), task affinity is computed and utilized in two stages. Initially, task affinity within a shared network is assessed by monitoring how the loss for each task changes when updating the loss of the other task at each step. Subsequently, using the averaged task affinity, task groups are organized, and each group is trained in separate networks. However, directly incorporating inter-task affinity for multi-task optimization presents several challenges. First, it only tracks the learning of shared parameters, $\Theta_s$, and their influence on task losses, without considering task-specific parameters, $\{\Theta_i\}_{i=1}^{\mathcal{K}}$. In practice, both shared and task-specific parameters need to be optimized simultaneously. Second, since inter-task affinity fluctuates significantly during

optimization, it must be tracked continuously to capture evolving task relations. This process is computationally expensive, requiring recursive gradient updates, monitoring loss changes, and reverting to previous parameter states.

## 3.2 PROXIMAL INTER-TASK AFFINITY FOR MULTI-TASK OPTIMIZATION

To address the issues mentioned above, we propose practical methods based on the concept of proximal inter-task affinity to track task relations and use this information for concurrent multi-task optimization. Firstly, we integrate the learning of task-specific parameters into the estimation of inter-task affinity to incorporate this affinity into ongoing optimization. Secondly, we extend the concept of inter-task affinity to elucidate relations between task groups rather than individual pairs of tasks. This allows us to monitor these affinities when a set of tasks is collectively backpropagated. Practically, proximal inter-task affinity serves as an approximation of inter-task affinity; however, it also plays a crucial role in theoretical analysis, as discussed in Section 4, explaining why updating task groups sequentially yields distinct multi-task performance. To be specific, we lay out guidelines for updating proximal inter-task affinity by distinguishing between inter-group relations and intra-group relations. We then incorporate the concept of proximal inter-task affinity to explore the task update strategy, grouping tasks based on their affinity, which has a significant impact on multi-task performance.

Before delving into the algorithm, let's establish some notations building upon those introduced in Section 3.1. Our objective is to partition tasks into several groups, denoted as $\{G_1, G_2, ..., G_{\mathcal{M}}\}$, where $\mathcal{M}$ is the number of task groups. Each group set contains tasks' indices as their components, for example, $G_k = \{i, j\}$. When selecting tasks to form these groups, we require the task groups to be mutually exclusive, meaning that $G_i \cap G_j = \emptyset$ for any $i$ and $j$ ($i \neq j$).

Tracking inter-task affinity during optimization, as defined in eq. (1), is impractical. This is because it is difficult to assess the influence of individual tasks on shared network parameters when multiple tasks are updated simultaneously. Additionally, task-specific parameters $\{\Theta_i\}_{i=1}^{\mathcal{K}}$ must also be updated concurrently, which is essential for MTL settings. Therefore, we practically approximate inter-task affinity by utilizing proximal inter-task affinity, taking into account updates of multiple tasks with task-specific parameters. The difference between inter-task affinity and proximal inter-task affinity is difficult to detect, so we provide a more detailed discussion in Appendix A.1.

**Definition 2** (**Proximal Inter-Task Affinity**). *Consider a multi-task network shared by the task set $G$, with their respective losses defined as $\mathcal{L}_G$. For a data sample $z^t$ and a learning rate $\eta$, the gradients of task set $G$ are updated to the parameters of the network as follows: $\Theta_{s|G}^{t+1} = \Theta_s^t - \eta \nabla_{\Theta_s^t} \mathcal{L}_G(z^t, \Theta_s^t, \Theta_G^t)$ and $\Theta_k^{t+1} = \Theta_k^t - \eta \nabla_{\Theta_k^t} \mathcal{L}_k(z^t, \Theta_s^t, \Theta_k^t)$ for $k \in G$. Then, the proximal inter-task affinity from the task set $G$ to $\tau_k$ at time step $t$ is defined as:*

$$\mathcal{B}_{G \to k}^t = 1 - \frac{\mathcal{L}_k(z^t, \Theta_{s|G}^{t+1}, \Theta_k^{t+1})}{\mathcal{L}_k(z^t, \Theta_s^t, \Theta_k^t)} \tag{2}$$

As it's not feasible to simultaneously track inter-task affinity and optimize networks, we instead track the proximal inter-task affinity and utilize it for multi-task optimization. We explain the benefits of integrating inter-task affinity into optimization in the view of conventional multi-task optimization. Theorem 1 suggests that grouping tasks with high inter-task affinity leads to better alignment of their gradients compared to grouping tasks with lower affinity. In analysis, we use the extended version of $\mathcal{A}$ for multiple tasks (e.g., $\mathcal{A}_{G \to k}$) for ease of notation.

**Theorem 1.** *Let $g_k$ denote the task-specific gradients backpropagated from the loss function $\mathcal{L}_k$ with respect to the parameters $\Theta_s^t$. At a given time step $t$, if the inter-task affinity from task group $\{i, k\}$ to task $k$ is greater than or equal to the inter-task affinity from group $\{j, k\}$ to task $k$, denoted as $A_{i,k \to k}^t \geq A_{j,k \to k}^t$. Then for a sufficiently small learning rate $\eta \ll 1$, it follows that $g_i \cdot g_k \geq g_j \cdot g_k$.*

The results also apply to proximal inter-task affinity $\mathcal{B}$ instead of $\mathcal{A}$, as the only difference is the inclusion of task-specific parameters for task $k$.

We analyze how this alignment in task-specific gradients reduces the loss of task $k$ in Theorem 2.

**Theorem 2.** *Let $g_k$ denote the task-specific gradients backpropagated from the loss function $\mathcal{L}_k$ with respect to the parameters $\Theta_s^t$. Let $\Theta_{s|k}^{t+1}$ represent the updated parameters after applying the*

*gradients. Assume that for tasks $i$, $j$, and $k$, the inequality $g_i \cdot g_k \geq g_j \cdot g_k$ holds. Then, for a sufficiently small learning rate $\eta \ll 1$, the inequality $\mathcal{L}_k(z^t, \Theta_{s|i,k}^{t+1}, \Theta_k^t) \leq \mathcal{L}_k(z^t, \Theta_{s|j,k}^{t+1}, \Theta_k^t)$ holds.*

The result indicates that when the gradients $g_i$ align better with $g_k$ than with $g_j$, the loss on the reference task $k$ using the updated gradients $g_i + g_k$ is lower than that using the updated gradients $g_j + g_k$. Theorems 1 and 2 suggest that updating tasks jointly with high proximal inter-task affinity at each optimization step is a reasonable approach.

## 3.3 TRACKING PROXIMAL INTER-TASK AFFINITY IN SELECTIVE GROUP UPDATES

From now on, we'll explain how we track proximal inter-task affinity in the sequential updates of task groups. For ease of explanation, we start with an already clustered (or initialized) task set $\{G_i\}_{i=1}^{\mathcal{M}}$ obtained from the tracked proximal inter-task affinity in the previous time step. $\mathcal{M}$ is the number of task groups. We'll provide more explanation at the end of this section on how the tracked proximal inter-task affinity is utilized in clustering the task set.

**Inter-Group Relations.** We begin by forwarding the sample $z^t$ to the shared multi-task network, from which we obtain the multi-task loss $\{\mathcal{L}_i(z^t, \Theta_s^t, \Theta_i^t)\}_{i=1}^{\mathcal{K}}$. Based on the task grouping $\{G_i\}_{i=1}^{\mathcal{M}}$, we divide each time step into $\mathcal{M}$ steps to update each task set. For simplicity of discussion, we update the first group $G_1$ in steps from $t$ to $t + 1/\mathcal{M}$, but the following discussion can be extended to all sequential updates between $t$ and $t + 1$.

$$\Theta_{s|G_1}^{t+1/\mathcal{M}} = \Theta_s^t - \eta \sum_{i \in G_1} w_i \nabla_{\Theta_s^t} \mathcal{L}_i(z^t, \Theta_s^t, \Theta_i^t) \tag{3}$$

$\Theta_{s|G_1}^{t+1/\mathcal{M}}$ represents the resulting parameter after updating the gradient from task group $G_1$ to the parameter $\Theta_s^t$. Upon forwarding $z^t$ once more, we can access $\{\mathcal{L}_i(z^t, \Theta_{s|G_1}^{t+1/\mathcal{M}}, \Theta_i^{t+1/\mathcal{M}})\}_{i=1}^{\mathcal{K}}$. For tasks that do not belong to $G_1$, their task-specific parameters remain unchanged, satisfying $\Theta_i^{t+1/\mathcal{M}} = \Theta_i^t$. Consequently, we can assess the proximal inter-task affinity from group $G_1$ to other tasks. In case where target task $\tau_j$ does not belong to $G_1$, we refer to this as inter-group relations as follows:

$$\mathcal{B}_{G_1 \to j}^t \simeq \mathcal{B}_{i \to j}^t = \mathcal{A}_{i \to j}^t = 1 - \frac{\mathcal{L}_j(z^t, \Theta_{s|G_1}^{t+1/\mathcal{M}}, \Theta_j^t)}{\mathcal{L}_j(z^t, \Theta_s^t, \Theta_j^t)} \qquad \text{where } \tau_i \in G_1 \text{ and } \tau_j \notin G_1 \tag{4}$$

To be precise, eq. (4) represents the proximal inter-task affinity from $G_1$ to $\tau_j$. However, since $\tau_i$ belongs to $G_1$, which is a group of tasks similar to $\tau_i$, we approximate the relations between $\tau_i$ and $\tau_j$ using $G_1$ and $\tau_j$. We explain how approximation $\mathcal{B}_{G_1 \to j}^t \simeq \mathcal{B}_{i \to j}^t$ can be justified in task group updates from the results of Theorem 3. We can repeat this process iteratively $\mathcal{M}$ times, with each step updating each task group in $\{G_i\}_{i=1}^{\mathcal{M}}$.

**Intra-Group Relations.** Inter-group relation addresses inter-task relations where the source task $\tau_i$ and the target task $\tau_j$ do not belong to the same group. Conversely, intra-group relation refers to the influence between tasks within the same task group, and it follows a slightly different process. Since both the source and target tasks are updated simultaneously, we must infer inter-task relations based on how the tasks' losses change. Let's suppose we're updating the task set $G_1$. The proximal inter-task affinity between tasks within the same group can be calculated as follows:

$$\mathcal{B}_{G_1 \to j}^t \simeq \mathcal{B}_{i \to j}^t = 1 - \frac{\mathcal{L}_j(z^t, \Theta_{s|G_1}^{t+1/\mathcal{M}}, \Theta_j^{t+1/\mathcal{M}})}{\mathcal{L}_j(z^t, \Theta_s^t, \Theta_j^t)} \qquad \text{where } \tau_i \in G_1 \text{ and } \tau_j \in G_1 \tag{5}$$

Similarly to the inter-group case, we approximate the relations from $\tau_i$ to $\tau_j$ using the relations from $G_1$ to $\tau_j$ and this also can be justified by Theorem 3 when we use it to cluster task sets. Since the target task $\tau_j$ is included in $G_1$, we infer the inter-task relations by dividing cases into two. If both $\mathcal{B}_{i \to j}^t$ and $\mathcal{B}_{j \to i}^t$ have positive signs, we infer that the inter-task affinity between $\tau_i$ and $\tau_j$ is positive. Otherwise, if at least one of $\mathcal{B}_{i \to j}^t$ or $\mathcal{B}_{j \to i}^t$ has a negative sign, the inter-task affinity between $\tau_i$ and $\tau_j$ is negative. This inference is intuitive, as an increase in one task's loss during the decrease of the other task's loss implies that they transfer negative influences. However, it results in faster convergence compared to previous gradient-based optimization methods, which require a number

---

**Algorithm 1:** Tracking Proximal Inter-Task Affinity for Task Group Updates

---

**Require:** Loss function $\{\mathcal{L}_i\}_{i=1}^{\mathcal{K}}$, Dataset $\{z_i\}_{i=1}^{n}$, Total iteration $T$,
Proximal inter-task affinity (w/ decay) $\mathcal{B}^t$ ($\tilde{\mathcal{B}}^t$), Task group set $\{G_i\}_{i=1}^{\mathcal{M}}$, Decay rate $\beta \in (0, 1)$
**Initialize:** Proximal inter-task affinity $\tilde{\mathcal{B}}^0 = 0_{\mathcal{K} \times \mathcal{K}}$,
Task group $G_i = \{\tau_i\}$ for $i = 1, ..., \mathcal{M}$ ($\mathcal{M} = \mathcal{K}$ for initialization)

1  **for** $t = 1, \ldots, T$ **do**
2     Randomly mix the sequence of task group $\{G_i\}_{i=1}^{\mathcal{M}}$ and calculate $\{\mathcal{L}_i(z^t, \Theta_s^t, \Theta_i^t)\}_{i=1}^{\mathcal{K}}$
3     **for** $k = 1, \ldots, \mathcal{M}$ **do**
4         Update gradients $\sum_{\tau_i \in G_k} w_i \nabla \mathcal{L}_i$ backpropagated from losses in $G_k$
5         Forward $z^t$ and calculate $\mathcal{L}_i(z^t, \Theta_{s|G_k}^{t+k/\mathcal{M}}, \Theta_i^{t+k/\mathcal{M}})$ for all $i = 1, ..., \mathcal{K}$
6         Calculate $\mathcal{B}_{i \to j}^{t+(k-1)/\mathcal{M}}$ and $\mathcal{B}_{j \to i}^{t+(k-1)/\mathcal{M}}$ where $\tau_i \in G_k$ for all $j = 1, ..., \mathcal{K}$
7         **if** $\tau_j \notin G_k$ **or** $\mathcal{B}_{i \to j}^{t+(k-1)/\mathcal{M}} \cdot \mathcal{B}_{j \to i}^{t+(k-1)/\mathcal{M}} \geq 0$ **then**
8           $\tilde{\mathcal{B}}_{i \to j}^{t+(k-1)/\mathcal{M}} = (1 - \beta)\tilde{\mathcal{B}}_{i \to j}^{t+(k-2)/\mathcal{M}} + \beta \mathcal{B}_{i \to j}^{t+(k-1)/\mathcal{M}}$
9         **else**
10          $\tilde{\mathcal{B}}_{i \to j}^{t+(k-1)/\mathcal{M}} = (1 - \beta)\tilde{\mathcal{B}}_{i \to j}^{t+(k-2)/\mathcal{M}} - \beta \max(|\mathcal{B}_{i \to j}^{t+(k-1)/\mathcal{M}}|, |\mathcal{B}_{j \to i}^{t+(k-1)/\mathcal{M}}|)$
11     Divide task groups $\{G_i\}_{i=1}^{\mathcal{M}'}$ based on proximal inter-task affinity $\tilde{\mathcal{B}}^{(t+\mathcal{M}-1)/\mathcal{M}}$

---

of backpropagations equal to the number of tasks, along with gradient manipulation. This is further analyzed in Section 5. In the initial phase of learning, we initialize task groups by assigning each task to a different group to facilitate the learning of proximal inter-task affinity. For stable tracking, we employ a decay rate $\beta$. Using the tracked proximal inter-task affinity, the next task groups $\{G_i\}_{i=1}^{\mathcal{M}'}$ are formed by grouping tasks with positive affinity, where $\mathcal{M}'$ is the number of task sets for the next step. During optimization, the number of clustered task sets $\mathcal{M}'$ continuously fluctuates along with their composition. The entire process is outlined in Algorithm 1.

We explain how it effectively approximates inter-task affinity in selective task group updates in Theorem 3. To be specific, we compare the inter-task affinity when the target task is included in the source task group versus when it is not.

**Theorem 3.** *The affinity between* $\{i, k\} \to k$ *and* $i \to k$ *satisfies* $\mathcal{A}_{i,k \to k}^t \geq \mathcal{A}_{i \to k}^t$.

When task $i$ and $k$ are within the same task group, we can only access $\mathcal{B}_{i,k \to k}^t$ rather than $\mathcal{B}_{i \to k}^t$ during the optimization process. If $\mathcal{B}_{i,k \to k}^t \leq 0$, the inter-task affinity also satisfies $\mathcal{A}_{i,k \to k}^t \leq 0$ as $\mathcal{A}_{i,k \to k}^t \leq \mathcal{B}_{i,k \to k}^t$ (see the end of appendix A.1). According to Theorem 3, this condition implies $\mathcal{A}_{i \to k}^t \leq 0$. The proposed algorithm separates these tasks into different groups when $\mathcal{B}_{i,k \to k}^t \leq 0$ which justifies our grouping rules.

Conversely, when tasks $\tau_i$ and $\tau_j$ belong to separate task groups, we only have access to $\mathcal{B}_{i \to k}^t$ instead of $\mathcal{B}_{i,k \to k}^t$. In this scenario, the proposed algorithm merges these tasks into the same group if $\mathcal{B}_{i \to k}^t \geq 0$. This inequality also implies $\mathcal{A}_{i \to k}^t \geq 0$ as $\mathcal{B}_{i \to k}^t = \mathcal{A}_{i \to k}^t$ (see the end of appendix A.1), justifying the merging of tasks $\tau_i$ and $\tau_k$ based on $\mathcal{B}_{i \to k}^t$ during optimization.

## 4   Theoretical Analysis

We outline the benefits of grouping tasks with positive inter-task affinity and updating them in multi-task optimization in Theorem 1 and Theorem 2. We validate the feasibility of tracking proximal inter-task affinity and forming task groups based on this in our grouping updates strategy in Theorem 3. In this section, we clarify why, from the viewpoint of standard convergence analysis in Theorem 4, it's difficult to demonstrate that updating task groups sequentially leads to better performance than joint learning. We then explore how sequentially updating grouped task sets can improve multi-task performance, as discussed in Theorem 5.

## 4.1 Convergence Analysis

We perform conventional convergence analysis to determine the convergence points of the sequential updates of task groups, as outlined in Theorem 4. Conventional analyses treat the learning of shared and task-specific parameters separately, which fails to explain why sequential updates result in better multi-task performance while maintaining comparable stability.

**Theorem 4** (Convergence Analysis). *Given a set of differentiable losses $\{\mathcal{L}_i\}_{i=1}^{\mathcal{K}}$ and Lipschitz continuous gradients with a constant $H > 0$, $||\nabla \mathcal{L}_i(x) - \nabla \mathcal{L}_i(y)|| \leq H||x-y||$ for all $i = 1, 2, ..., \mathcal{K}$. If we sequentially update task groups $\{G_i\}_{i=1}^{\mathcal{M}}$ based on inter-task affinity, employing a sufficiently small step size $\eta \leq \min(\frac{2}{H\mathcal{K}}, \frac{1}{H|G_m|})$ where $|G_m|$ is the number of tasks in $G_m$, then the following inequalities are satisfied for any task group $G_m$.*

$$\sum_{k=1}^{\mathcal{M}} \mathcal{L}_{G_k}(z^t, \Theta_{s|G_m}^{t+m/\mathcal{M}}, \Theta_{G_k}^{t+1}) \leq \sum_{k=1}^{\mathcal{M}} \mathcal{L}_{G_k}(z^t, \Theta_{s|G_m}^{t+(m-1)/\mathcal{M}}, \Theta_{G_k}^t) \tag{6}$$

$$- \eta g_{s,G_m}^{t+(m-1)/\mathcal{M}} \cdot (\sum_{k=1}^{\mathcal{M}} g_{s,G_k}^{t+(m-1)/\mathcal{M}} - g_{s,G_m}^{t+(m-1)/\mathcal{M}}) - \frac{\eta}{2}||g_{ts,G_m}^t||^2 \tag{7}$$

where $g_{s,G_m}$ and $g_{ts,G_m}$ represent the gradients of the shared parameters and task-specific parameters, respectively, for group $G_m$. Previous approaches, which handle updates of shared and task-specific parameters independently, failing to capture their interdependence during optimization. The term, $g_{s,G_m}^{t+(m-1)/\mathcal{M}} \cdot (\sum_{k=1}^{\mathcal{M}} g_{s,G_k}^{t+(m-1)/\mathcal{M}})$, fluctuates during optimization. When the gradients of group $G_m$ align well with the gradients of the other groups $\{G_k\}_{i=1, i \neq m}^{\mathcal{M}}$, their dot product yields a positive value, leading to a decrease in multi-task losses. In practice, the sequential update strategy demonstrates a similar level of stability in optimization, which appears to contradict the conventional results. Thus, we assume a correlation between the learning of shared parameters and task-specific parameters, where the learning of task-specific parameters reduces gradient conflicts in shared parameters. Under this assumption, the sequential update strategy can guarantee convergence to Pareto-stationary points. This assumption is reasonable, as task-specific parameters capture task-specific information, thereby reducing conflicts in the shared parameters across tasks.

## 4.2 Advantages of Selective Task Group Updates

Nonetheless, sequentially updating task groups facilitates the learning of task-specific parameters, a concept not covered by conventional analysis. We directly compare the loss of joint learning with the loss of the sequential group updates strategy. To do this, we introduce two-step proximal inter-task affinity, an expanded concept of proximal inter-task affinity over multiple steps, as defined in Appendix A.6. We analyze the benefits of sequential updates using this two-step proximal inter-task affinity based on the plausible assumptions delineated in Theorem 5.

**Theorem 5.** *Consider three tasks $\{i, j, k\}$, where task groups are formed with positive inter-task affinity as $\{i, k\}$ and $\{j\}$. Assume all losses are convex and differentiable, and the change in affinity during a single step from $t + (m-1)/\mathcal{M}$ to $t + m/\mathcal{M}$ is negligible. The affinity, which learns all tasks jointly, is denoted as $\mathcal{B}_{i,j,k \rightarrow k}^{t+(m-1)/\mathcal{M}}$. The affinity for the updating sequence $(\{i, k\}, \{j\})$ is represented as $\mathcal{B}_{i,k;j \rightarrow k}^{t+(m-1)/\mathcal{M}}$, while for the sequence $(\{j\}, \{i, k\})$, it is represented as $\mathcal{B}_{j;i,k \rightarrow k}^{t+(m-1)/\mathcal{M}}$. Then, for a sufficiently small learning rate $\eta \ll 1$, the following holds:*

$$\mathcal{B}_{i,j,k \rightarrow k}^{t+(m-1)/\mathcal{M}} \simeq \mathcal{B}_{i,k;j \rightarrow k}^{t+(m-1)/\mathcal{M}} \qquad \mathcal{B}_{i,j,k \rightarrow k}^{t+(m-1)/\mathcal{M}} \leq \mathcal{B}_{j;i,k \rightarrow k}^{t+(m-1)/\mathcal{M}} \tag{8}$$

This suggests that grouping tasks with proximal inter-task affinity and subsequently updating these groups sequentially result in lower multi-task loss compared to jointly backpropagating all tasks. This disparity arises because the network can discern superior task-specific parameters to accommodate task-specific information during sequential learning. Notably, the order in which tasks are updated impacts multi-task outcomes within a single batch. However, as the optimization progresses, this influence diminishes, as demonstrated in the following experimental results.

Table 1: Comparison of time complexity and memory consumption between our optimization methods and other multi-task optimization approaches, including loss-based and gradient-based methods.

| Method | Forward Pass | Backpropagation | Gradient Manipulation | Optimizer Step | Affinity Update | Memory |
|---|---|---|---|---|---|---|
| Loss-based Methods | $\mathcal{O}(1)$ | $\mathcal{O}(1)$ | ✗ | $\mathcal{O}(1)$ | ✗ | $\mathcal{O}(1)$ |
| Gradient-based Methods | $\mathcal{O}(1)$ | $\mathcal{O}(\mathcal{K})$ | ✓ | $\mathcal{O}(1)$ | ✗ | $\mathcal{O}(\mathcal{K})$ |
| Ours | $\mathcal{O}(\mathcal{M})$ | $\mathcal{O}(\mathcal{M})$ | ✗ | $\mathcal{O}(\mathcal{M})$ | ✓ | $\mathcal{O}(1)$ |

Figure 2: Comparison of the average time required by each optimization process to handle a single batch for 5 tasks on PASCAL-Context (left) and 11 tasks on Taskonomy (right).

## 5 EXPERIMENTS

**Experimental settings.** We assess the proposed techniques using three datasets: NYUD-V2 for indoor vision tasks (Silberman et al., 2012), PASCAL-Context for outdoor scenarios (Mottaghi et al., 2014), and Taskonomy (Zamir et al., 2018) for large number of tasks. Multi-task performance is compared using the metric introduced by (Maninis et al., 2019). This metric calculates the per-task performance by averaging it relative to the single-task baseline $b$: $\triangle_m = (1/\mathcal{K}) \sum_{i=1}^{\mathcal{K}} (-1)^{l_i} (M_{m,i} - M_{b,i})/M_{b,i}$ where $l_i = 1$ if a lower value of measure $M_i$ means indicates better performance for task $\tau_i$, and 0 otherwise. More details are introduced in Appendix C.

**Baselines.** We selected conventional multi-task optimization methods as our baselines: 1) Single-task learning, where each task is trained independently; 2) GD, where all task gradients are updated jointly without manipulation; 3) Gradient-based optimization methods that include gradient manipulation, such as GradDrop (Chen et al., 2020), MGDA (Sener & Koltun, 2018), PCGrad (Yu et al., 2020), CAGrad (Liu et al., 2021a), Aligned-MTL (Senushkin et al., 2023), and Nash-MTL (Navon et al., 2022); 4) Loss-based optimization methods, including UW (Kendall et al., 2018), DTP (Guo et al., 2018), DWA (Liu et al., 2019), and FAMO (Liu et al., 2024); and 5) A hybrid approach combining gradient and loss-based methods, specifically IMTL (Liu et al., 2021b). Each experiment was conducted three times with different random seeds to ensure a fair comparison.

**Architectures.** We use the most common multi-task architecture, employing a shared encoder and multiple decoders, each dedicated to a specific task. For our encoder, we mainly use ViT Dosovitskiy et al. (2020), coupled with a single convolutional layer as the decoder.

We conduct experimental analysis to address several key statements and questions. For additional and comprehensive experimental results, please refer to Appendix D.

**Computational cost and memory consumption.** In Table 1, we compare the time complexity and memory consumption of previous approaches with our methods. Additionally, Figure 2 presents the average time required by each optimization process to handle a single batch. Loss-based approaches, such as FAMO, DWA, DTP, and UW, exhibit the lowest computational cost and memory usage because they do not require multiple forward passes or backpropagation. In contrast, gradient-based approaches, including Nash-MTL, Aligned-MTL, CAGrad, PCGrad, and GradDrop, incur significantly higher computational costs due to the iterative backpropagation required for each task ($\mathcal{K}$), even though they have the potential for better performance. These approaches also need to store task-specific gradients, which demands $\mathcal{K}$ times more memory. Our methods, however, require $\mathcal{M}$ (number of groups) forward passes and backward passes, where $\mathcal{M} < \mathcal{K}$. As shown in Figure 3c, the number of groups changes during optimization on Taskonomy, with our methods maintaining an average $\mathcal{M} = 1.8$ with ViT-L, much lower than $\mathcal{K} = 11$ for the Taskonomy benchmark. Despite the need for multiple forward passes, our methods remain computationally competitive because the majority of the computational load is concentrated on backpropagation and gradient manipulation.

Table 2: Experimental results on the Taskonomy dataset using ViT-L. The best results in each column are shown in bold, while convergence failures are indicated with a dash.

| Task | DE | DZ | EO | ET | K2 | K3 | N | C | R | S2 | S2.5 | $\triangle_m$ (↑) |
| | L1 Dist. ↓ | L1 Dist. ↓ | L1 Dist. ↓ | L1 Dist. ↓ | L1 Dist. ↓ | L1 Dist. ↓ | L1 Dist. ↓ | RMSE ↓ | L1 Dist. ↓ | L1 Dist. ↓ | L1 Dist. ↓ | |
|---|---|---|---|---|---|---|---|---|---|---|---|---|
| Single Task | 0.0155 | 0.0160 | 0.1012 | 0.1713 | 0.1620 | 0.082 | 0.2169 | 0.7103 | 0.1357 | 0.1700 | 0.1435 | - |
| GD | 0.0163 | 0.0167 | 0.1211 | 0.1742 | 0.1715 | 0.093 | 0.2333 | 0.7527 | 0.1625 | 0.1837 | 0.1487 | -8.65 ±0.229 |
| GradDrop | 0.0168 | 0.0172 | 0.1229 | 0.1744 | 0.1727 | 0.091 | 0.2562 | 0.7615 | 0.1656 | 0.1862 | 0.1511 | -10.81 ±0.377 |
| MGDA | - | - | - | - | - | - | - | - | - | - | - | - |
| UW | 0.0167 | 0.0151 | 0.1212 | 0.1728 | 0.1712 | 0.089 | 0.2360 | 0.7471 | 0.1607 | 0.1829 | 0.1538 | -7.65 ±0.087 |
| DTP | 0.0169 | 0.0153 | 0.1213 | **0.1720** | 0.1707 | 0.089 | 0.2517 | 0.7481 | 0.1603 | 0.1814 | 0.1503 | -8.16 ±0.081 |
| DWA | 0.0147 | 0.0155 | 0.1209 | 0.1725 | 0.1711 | 0.089 | 0.2619 | 0.7486 | 0.1613 | 0.1845 | 0.1543 | -7.92 ±0.077 |
| PCGrad | 0.0161 | 0.0165 | 0.1206 | 0.1735 | 0.1696 | 0.090 | 0.2301 | 0.7540 | 0.1625 | 0.1830 | 0.1483 | -7.72 ±0.206 |
| CAGrad | 0.0162 | 0.0166 | 0.1202 | 0.1769 | 0.1651 | 0.091 | 0.2565 | 0.7653 | 0.1661 | 0.1861 | 0.1571 | -10.05 ±0.346 |
| IMTL | 0.0162 | 0.0165 | 0.1206 | 0.1741 | 0.1710 | 0.090 | 0.2268 | 0.7497 | 0.1617 | 0.1832 | 0.1543 | -8.03 ±0.179 |
| Aligned-MTL | 0.0150 | 0.0155 | **0.1135** | 0.1725 | **0.1630** | **0.086** | 0.2513 | 0.8039 | 0.1646 | 0.1800 | **0.1438** | -6.22 ±0.285 |
| Nash-MTL | - | - | - | - | - | - | - | - | - | - | - | - |
| FAMO | 0.0157 | 0.0155 | 0.1211 | 0.1730 | 0.1702 | 0.090 | 0.2433 | 0.7479 | 0.1610 | 0.1823 | 0.1527 | -7.58 ±0.211 |
| Ours | **0.0140** | **0.0145** | 0.1136 | 0.1735 | 0.1679 | 0.087 | **0.2029** | **0.7166** | **0.1500** | **0.1769** | 0.1469 | **-1.42 ±0.208** |

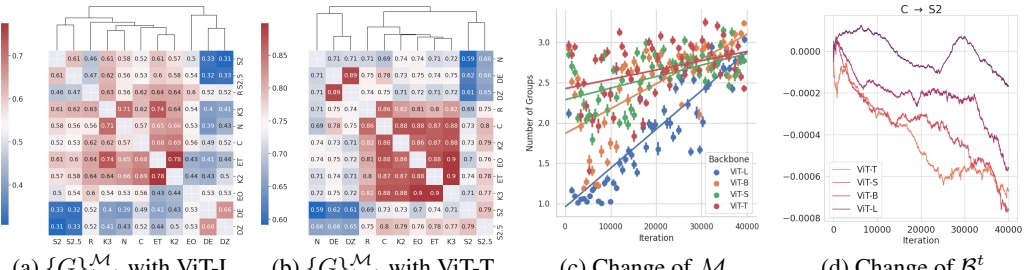

(a) $\{G\}_{i=1}^{\mathcal{M}}$ with ViT-L  (b) $\{G\}_{i=1}^{\mathcal{M}}$ with ViT-T  (c) Change of $\mathcal{M}$  (d) Change of $\mathcal{B}^t$

Figure 3: The averaged grouping results $\{G\}_{i=1}^{\mathcal{M}}$ on the Taskonomy benchmark are shown for ViT-L in (a) and for ViT-T in (b). (c) illustrates how the number of task groups, $\mathcal{M}$, changes during optimization. (d) shows the change in proximal inter-task affinity from DE to C.

This trend becomes more advantageous as the number of tasks increases. Our methods achieve a computational cost that falls between loss-based and gradient-based methods, while maintaining similar memory consumption to loss-based methods.

**Optimization results comparison.** We compare the results of multi-task optimization on Taskonomy in Table 2 and on NYUD-v2 and PASCAL-Context in Table 3. Our methods achieve superior multi-task performance across all benchmarks, demonstrating their practical effectiveness. One key observation is that the tasks showing the greatest performance improvements differ between our methods and previous approaches, reflecting the distinct motivations behind each optimization design. Recent methods like Nash-MTL, IMTL, and Aligned-MTL focus on improving edge detection, often at the expense of other tasks. This is due to their emphasis on balancing task losses or gradients, which are affected by unbalanced loss scales. Since edge detection shows the lowest loss scale across the optimization process, these methods prioritize enhancing its performance. Loss-based approaches such as FAMO, DWA, DTP, and UW share a similar motivation but exhibit limited performance compared to gradient-based methods. In contrast, our methods enhance performance across all tasks compared to GD by optimizing task-specific parameters in a grouping update, which benefits all tasks. Furthermore, our methods achieve stable convergence with many tasks, unlike MGDA or Nash-MTL, which struggle to converge in such scenarios on Taskonomy.

**Grouping results of selective task group updates.** In Figure 3, we illustrate how the grouping strategy evolves during the optimization process. Specifically, in Figures 3a and 3b, we show the frequency with which different tasks are grouped together throughout the entire optimization process. Similar tasks, such as depth euclidean (DE) and depth zbuffer (DZ), tend to be grouped more often, whereas depth-related tasks are less frequently grouped with tasks of other types. Overall, the trends in task grouping are consistent across different backbone network sizes. However, one notable observation is that the overall level of grouping increases as the backbone size grows. This suggests that larger models are better at extracting generalizable features across multiple tasks, leading to higher inter-task affinity. In Figure 3c, we present the average number of task groups $\mathcal{M}$ throughout the optimization process. During optimization, $\mathcal{M}$ tends to increase as proximal inter-task affinity decreases, suggesting that task competition grows more intense over time. Moreover, $\mathcal{M}$ grows as the backbone network's capacity increases, which aligns with the grouping patterns observed for different backbone sizes in Figures 3a and 3b. We provide an example of proximal inter-task affinity in Figure 3d, with all task pairs shown in Figure D.2.

Table 3: We assess the proposed method alongside previous multi-task optimization approaches on both NYUD-v2 and PASCAL-Context. The best results are highlighted in bold.

| Method | NYUD-v2 | | | | | PASCAL-Context | | | | | |
| --- | --- | --- | --- | --- | --- | --- | --- | --- | --- | --- | --- |
| | Semseg mIoU ↑ | Depth RMSE ↓ | Normal mErr ↓ | Edge odsF ↑ | $\triangle_m$ % ↑ | Semseg mIoU ↑ | Parsing mIoU ↑ | Saliency maxF ↑ | Normal mErr ↓ | Edge odsF ↑ | $\triangle_m$ % ↑ |
| Single Task | 39.35 | 0.661 | 22.14 | 59.7 | - | 67.96 | 58.90 | 83.76 | 15.65 | 47.7 | - |
| GD | 38.51 | 0.641 | 25.64 | 53.0 | -6.54 ±0.171 | 67.48 | 55.46 | 81.36 | 18.41 | 39.0 | -9.06 ±0.095 |
| GradDrop | 38.42 | 0.638 | 25.75 | 53.0 | -6.60 ±0.264 | 67.18 | 55.35 | 81.32 | 18.53 | 39.0 | -9.35 ±0.092 |
| MGDA | 20.93 | 0.870 | 36.66 | 59.8 | -35.96 ±0.117 | - | - | - | - | - | - |
| UW | 38.20 | 0.631 | 25.32 | 53.0 | -5.99 ±0.196 | 67.11 | 55.22 | 81.33 | 18.44 | 38.6 | -9.46 ±0.113 |
| DTP | 38.52 | 0.633 | 25.60 | 52.8 | -6.26 ±0.214 | 67.09 | 55.23 | 81.35 | 18.40 | 38.6 | -9.41 ±0.108 |
| DWA | 38.56 | 0.634 | 25.62 | 52.8 | -6.30 ±0.220 | 67.11 | 55.23 | 81.34 | 18.44 | 38.6 | -9.46 ±0.116 |
| PCGrad | 38.26 | 0.633 | 25.40 | 54.0 | -5.70 ±0.083 | 66.62 | 55.11 | 81.82 | 18.16 | 40.0 | -8.58 ±0.101 |
| CAGrad | 38.31 | 0.641 | 26.11 | 58.0 | -5.10 ±0.243 | 66.31 | 54.96 | 81.28 | 18.68 | 44.9 | -7.46 ±0.067 |
| Aligned-MTL | 36.20 | 0.655 | 23.77 | **58.5** | -4.12 ±0.121 | 60.78 | 54.42 | 82.29 | 17.44 | **45.5** | -7.20 ±0.075 |
| IMTL | 37.19 | 0.652 | **23.45** | 57.8 | -3.31 ±0.213 | 63.91 | 55.23 | 82.23 | 17.83 | 44.2 | -7.07 ±0.104 |
| Nash-MTL | 36.94 | 0.641 | 23.60 | 58.0 | -3.14 ±0.115 | **68.50** | 10.32 | 75.63 | 22.25 | 35.3 | -31.91 ±0.166 |
| FAMO | 38.57 | 0.636 | 25.61 | 53.1 | -6.23 ±0.166 | 67.12 | 55.23 | 81.34 | 18.44 | 38.6 | -9.46 ±0.115 |
| Ours | **40.02** | **0.618** | 24.09 | 53.9 | **-2.58** ±0.205 | 68.14 | **57.15** | **82.52** | **17.19** | 39.5 | **-6.24** ±0.192 |

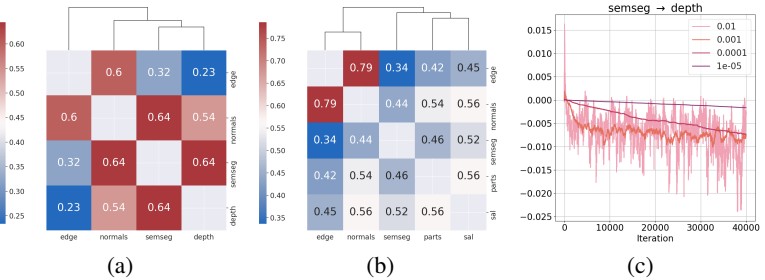

(a)  (b)  (c)

Table 5: Ablation studies on Taskonomy exploring the impact of group order and type of grouping.

| Method | $\triangle_m$ (↑) |
| --- | --- |
| Order of Group | |
| Forward | -1.41 ±0.211 |
| Backward | -1.44 ±0.209 |
| Grouping Methods | |
| Joint (GD) | -8.65 ±0.229 |
| Separate | -2.48 ±0.651 |
| Ours | -1.42 ±0.208 |

Figure 4: The averaged grouping results $\{G\}_{i=1}^{\mathcal{M}}$ are shown for NYUD-v2 in (a) and PASCAL-Context in (b). (c) illustrates how the decaying factor $\beta$ influences the stable tracking of proximal inter-task affinity.

**Stability of optimization.** In Tables 2 and 3, we also report the variance in multi-task performance. Our methods demonstrate a similar level of variance compared to previous optimization approaches. Additionally, we assess how the order of task group updates and the type of grouping affect performance and optimization stability in Table 5. Using the task groups identified from tracked proximal inter-task affinity, we compare performance when updating in forward order, backward order, and randomly selected order. The results show that the order of task group updates does not significantly impact the final multi-task performance or algorithm stability. When comparing our method to gradient descent (GD), where all tasks are grouped together ($\mathcal{M} = 1$), and a scenario where each task is placed in a separate group ($\mathcal{M} = \mathcal{K}$), our approach demonstrates better performance and greater stability by grouping tasks with high proximal inter-task affinity.

**The effect of the decay rate on tracking affinity.** The affinity decay rate $\beta$ influences the tracking of proximal inter-task affinity as shown in Figure 4c. A larger $\beta$ results in more dynamic fluctuations in affinity, resulting in more frequent changes in task groupings. In contrast, a smaller $\beta$ leads to slower adjustments in task groupings. We observe that the affinity decay rate $\beta$ does not significantly impact multi-task performance, which enhances the method's applicability. In our experiments, we set $\beta$ to 0.001. We anticipate that $\beta$ can be easily adjusted by monitoring the variation of proximal inter-task affinity. The overall results are shown in Figure D.5.

## 6 CONCLUSION

We propose a novel approach to mitigate negative transfer by dynamically dividing task groups during optimization. Through empirical experimentation, we observed significant differences in multi-task performance depending on whether task losses are backpropagated collectively or updated sequentially. Building on this insight, we introduce an algorithm that adaptively separates task sets and updates them within a single shared architecture during optimization. To facilitate simultaneous tracking of inter-task relations and network optimization, we introduce proximal inter-task affinity, which can be measured throughout the optimization process. Our analysis highlights the profound impact of sequential updates on the learning of task-specific representations. Ultimately, our methods substantially enhance multi-task performance compared to previous multi-task optimization.

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

# A    PROOFS OF THEORETICAL ANALYSIS

Before diving into the theoretical analysis proof, we'll briefly introduce the basic concepts used in our proof to ensure the paper is self-contained.

**Definition 3** (Lipschitz continuity). *$f$ is $\lambda$-Lipschitz if for any two points $u, v$ in the domain of $f$, we have following inequality:*

$$|f(u) - f(v)| \leq \lambda ||u - v|| \tag{9}$$

## A.1    DIFFERENCE BETWEEN INTER-TASK AFFINITY AND PROXIMAL INTER-TASK AFFINITY

Firstly, let's reiterate the definitions of inter-task affinity and proximal inter-task affinity from the main paper.

In a typical SGD process for task $i$ at time step $t$ with input $z^t$, the update rule for $\Theta_s$ is as follows: $\Theta_{s|i}^{t+1} = \Theta_s^t - \eta w_i \nabla_{\Theta_s^t} \mathcal{L}_i(z^t, \Theta_s^t, \Theta_i^t)$ where $z^t$ represents the input data and $\eta$ is the learning rate, $\Theta_{s|i}^{t+1}$ is the updated shared parameters with loss $\mathcal{L}_i$. Then the affinity from task $i$ to $k$ at time step $t$, denoted as $\mathcal{A}_{i \to k}^t$, is:

$$\mathcal{A}_{i \to k}^t = 1 - \frac{\mathcal{L}_k(z^t, \Theta_{s|i}^{t+1}, \Theta_k^t)}{\mathcal{L}_k(z^t, \Theta_s^t, \Theta_k^t)} \tag{10}$$

For proximal inter-task affinity, let's consider a multi-task network shared by the task set $G$, with their respective losses defined as $\mathcal{L}_G$. For a data sample $z^t$ and a learning rate $\eta$, the gradients of task set $G$ are updated to the parameters of the network as follows: $\Theta_{s|G}^{t+1} = \Theta_s^t - \eta \nabla_{\Theta_s^t} \mathcal{L}_G(z^t, \Theta_s^t, \Theta_G^t)$ and $\Theta_k^{t+1} = \Theta_k^t - \eta \nabla_{\Theta_k^t} \mathcal{L}_k(z^t, \Theta_s^t, \Theta_k^t)$ for $k \in G$. Then, the proximal inter-task affinity from group $G$ to task $k$ at time step $t$ is defined as:

$$\mathcal{B}_{G \to k}^t = 1 - \frac{\mathcal{L}_k(z^t, \Theta_{s|G}^{t+1}, \Theta_k^{t+1})}{\mathcal{L}_k(z^t, \Theta_s^t, \Theta_k^t)} \tag{11}$$

The primary distinction between the two affinities lies in the incorporation of the task set and the update of task-specific parameters (indicated by red letters). Proximal inter-task affinity is an expanded concept that integrates the task set rather than individual tasks as the source task. This difference is evident from the notation, where $i \to k$ in Equation (10) and $G \to k$ in Equation (11).

The second main difference lies in the update of task-specific parameters. In the inter-task affinity in Equation (10), the denominator includes $\Theta_k^t$, while in the proximal inter-task affinity in Equation (11), it includes $\Theta_k^{t+1}$, which is a subtle distinction that may not be noticed by readers. These two modifications allow us to track proximal inter-task affinity while simultaneously optimizing multi-task networks.

When measuring affinity under the assumption of a convex objective, the proximal inter-task affinity is equal to or greater than the inter-task affinity. This also aligns well with real-world scenarios, as proximal inter-task affinity reflects updates to task-specific parameters, as shown below.

$$\mathcal{A}_{i \to k}^t = 1 - \frac{\mathcal{L}_k(z^t, \Theta_{s|i}^{t+1}, \Theta_k^t)}{\mathcal{L}_k(z^t, \Theta_s^t, \Theta_k^t)} \leq 1 - \frac{\mathcal{L}_k(z^t, \Theta_{s|i}^{t+1}, \Theta_k^{t+1})}{\mathcal{L}_k(z^t, \Theta_s^t, \Theta_k^t)} = \mathcal{B}_{i \to k}^t \tag{12}$$

This inequality can also be applied to an expanded setting that incorporates task sets. If we expand the concept of inter-task affinity from individual task to task set as $\mathcal{A}_{G \to k}^t$, then the inequality $\mathcal{A}_{G \to k}^t \leq \mathcal{B}_{G \to k}^t$ is satisfied. If $k \notin G$ then, $\mathcal{A}_{G \to k}^t = \mathcal{B}_{G \to k}^t$ holds.

For ease of notation, we use the expanded version of the affinity for multiple tasks throughout the proof, as follows:

$$\mathcal{A}_{G \to k}^t = 1 - \frac{\mathcal{L}_k(z^t, \Theta_{s|G}^{t+1}, \Theta_k^t)}{\mathcal{L}_k(z^t, \Theta_s^t, \Theta_k^t)} \tag{13}$$

This differs from proximal inter-task affinity, as it does not consider the update of task-specific parameters.

## A.2 PROOF OF THEOREM 1

**Theorem 1.** *Let $g_k$ denote the task-specific gradients backpropagated from the loss function $\mathcal{L}_k$ with respect to the parameters $\Theta_s^t$. At a given time step $t$, if the inter-task affinity from task group $\{i, k\}$ to task $k$ is greater than or equal to the inter-task affinity from group $\{j, k\}$ to task $k$, denoted as $A_{i,k\to k}^t \geq A_{j,k\to k}^t$. Then for a sufficiently small learning rate $\eta \ll 1$, it follows that $g_i \cdot g_k \geq g_j \cdot g_k$.*

*Proof.* Let's consider a scenario where we update the network parameters $\Theta$ with task-specific losses $\mathcal{L}_i$ and $\mathcal{L}_k$ simultaneously at time step $t$ with input $z^t$. Applying the Taylor expansion, we obtain the following:

$$\mathcal{L}_k(z^t, \Theta_{s|i,k}^{t+1}, \Theta_k^t) \simeq \mathcal{L}_k(z^t, \Theta_s^t, \Theta_k^t) + (\Theta_{s|i,k}^{t+1} - \Theta_s^t)\nabla_{\Theta_s^t}\mathcal{L}_k(z^t, \Theta_s^t, \Theta_k^t) + O(\eta^2) \quad (14)$$

$$= \mathcal{L}_k(z^t, \Theta_s^t, \Theta_k^t) - \eta g_k \cdot (g_i + g_k) + O(\eta^2) \quad (15)$$

where $g_i$ and $g_k$ represent the gradients backpropagated from the losses $\mathcal{L}_i$ and $\mathcal{L}_k$, respectively, with respect to the shared parameters $\Theta_s^t$. For instance, $g_i = \nabla_{\Theta_s^t}\mathcal{L}_i(z^t, \Theta_s^t, \Theta_i)$.

Reorganizing the inequality to align with the format of inter-task affinity, we obtain:

$$\mathcal{A}_{i,k\to k}^t = 1 - \frac{\mathcal{L}_k(z^t, \Theta_{s|i,k}^{t+1}, \Theta_k^t)}{\mathcal{L}_k(z^t, \Theta_s^t, \Theta_k^t)} \simeq \frac{1}{\mathcal{L}_k(z^t, \Theta_s^t, \Theta_k^t)}\left(\eta g_k \cdot (g_i + g_k) + O(\eta^2)\right) \quad (16)$$

Similar results can be obtained for $A_{j,k\to k}^t$.

$$\mathcal{A}_{j,k\to k}^t = 1 - \frac{\mathcal{L}_k(z^t, \Theta_{s|j,k}^{t+1}, \Theta_k^t)}{\mathcal{L}_k(z^t, \Theta_s^t, \Theta_k^t)} \simeq \frac{1}{\mathcal{L}_k(z^t, \Theta_s^t, \Theta_k^t)}\left(\eta g_k \cdot (g_j + g_k) + O(\eta^2)\right) \quad (17)$$

From $\mathcal{A}_{i,k\to k}^t \geq \mathcal{A}_{j,k\to k}^t$ and by ignoring the $O(\eta^2)$ term with a sufficiently small learning rate $\eta \ll 1$, we can derive the result:

$$g_i \cdot g_k \geq g_j \cdot g_k \quad (18)$$

$\square$

The findings indicate that grouping tasks with positive inter-task affinity exhibits better alignment in task-specific gradients compared to grouping tasks with negative inter-task affinity, thereby validating the grouping strategies employed by our algorithm. Furthermore, we analyze how this alignment in task-specific gradients contributes to reducing the loss of task $k$ in Theorem 2.

## A.3 PROOF OF THEOREM 2

**Theorem 2.** *Let $g_k$ denote the task-specific gradients backpropagated from the loss function $\mathcal{L}_k$ with respect to the parameters $\Theta_s^t$. Let $\Theta_{s|k}^{t+1}$ represent the updated parameters after applying the gradients. Assume that for tasks $i$, $j$, and $k$, the inequality $g_i \cdot g_k \geq g_j \cdot g_k$ holds. Then, for a sufficiently small learning rate $\eta \ll 1$, the inequality $\mathcal{L}_k(z^t, \Theta_{s|i,k}^{t+1}, \Theta_k^t) \leq \mathcal{L}_k(z^t, \Theta_{s|j,k}^{t+1}, \Theta_k^t)$ holds.*

*Proof.* Let's consider a scenario where we update the network parameters $\Theta_s^t$ with task-specific losses $\mathcal{L}_i$ and $\mathcal{L}_k$ simultaneously at time step $t$ with input $z^t$. Let $g_i$ denote the gradients backpropagated from the loss $\mathcal{L}_i$ with respect to the shared parameters $\Theta_s^t$, expressed as $g_i = \nabla\Theta_s^t\mathcal{L}_i(z^t, \Theta_s^t, \Theta_i)$.

Using the first-order Taylor approximation of $\mathcal{L}_k$ for $\Theta_s^t$, we obtain:

$$\mathcal{L}_k(z^t, \Theta_{s|i,k}^{t+1}, \Theta_k^t) = \mathcal{L}_k(z^t, \Theta_s^t, \Theta_k^t) + (\Theta_{s|i,k}^{t+1} - \Theta_s^t)\nabla_{\Theta_s^t}\mathcal{L}_k(z^t, \Theta_s^t, \Theta_k^t) + O(\eta^2) \quad (19)$$

$$= \mathcal{L}_k(z^t, \Theta_s^t, \Theta_k^t) - \eta(g_i + g_k) \cdot g_k + O(\eta^2) \quad (20)$$

For task $j$, we can follow a similar process as follows:

$$\mathcal{L}_k(z^t, \Theta_{s|j,k}^{t+1}, \Theta_k^t) = \mathcal{L}_k(z^t, \Theta_s^t, \Theta_k^t) - \eta(g_j + g_k) \cdot g_k + O(\eta^2) \quad (21)$$

With a sufficiently small learning rate $\eta \ll 1$, subtract eq. (21) from eq. (20), then:

$$\mathcal{L}_k(z^t, \Theta_{s|i,k}^{t+1}, \Theta_k^t) - \mathcal{L}_k(z^t, \Theta_{s|j,k}^{t+1}, \Theta_k^t) = -\eta(g_i + g_k) \cdot g_k + \eta(g_j + g_k) \cdot g_k \tag{22}$$

$$= -\eta(g_i - g_j) \cdot g_k \leq 0 \tag{23}$$

which proves the results. $\qquad\square$

The result indicates that when the gradients $g_i$ from task $i$ align better with those of the reference task $k$ compared to task $j$, the loss on the reference task $k$ tends to be lower with updated gradients $g_i + g_k$ compared to $g_j + g_k$, especially for sufficiently small learning rates $\eta$.

## A.4 PROOF OF THEOREM 3

**Theorem 3.** *The affinity between $\{i, k\} \to k$ and $i \to k$ satisfies $\mathcal{A}_{i,k\to k}^t \geq \mathcal{A}_{i\to k}^t$.*

*Proof.* Let's begin with the definition of inter-task affinity between $\{i, j\} \to k$ and $i \to k$ as follows:

$$\mathcal{A}_{i,k\to k}^t = 1 - \frac{\mathcal{L}_k(z^t, \Theta_{s|i,k}^{t+1}, \Theta_k^t)}{\mathcal{L}_k(z^t, \Theta_s^t, \Theta_k^t)} \qquad\qquad \mathcal{A}_{i\to k}^t = 1 - \frac{\mathcal{L}_k(z^t, \Theta_{s|i}^{t+1}, \Theta_k^t)}{\mathcal{L}_k(z^t, \Theta_s^t, \Theta_k^t)} \tag{24}$$

When updating $i$ and $k$ simultaneously, we can derive the first-order Taylor approximation of $\mathcal{L}_k$ for $\Theta_s^t$ as follows:

$$\mathcal{L}_k(z^t, \Theta_{s|i,k}^{t+1}, \Theta_k^t) = \mathcal{L}_k(z^t, \Theta_s^t, \Theta_k^t) + (\Theta_{s|i,k}^{t+1} - \Theta_s^t)\nabla_{\Theta_s^t}\mathcal{L}_k(z^t, \Theta_s^t, \Theta_k^t) + O(\eta^2) \tag{25}$$

$$= \mathcal{L}_k(z^t, \Theta_s^t, \Theta_k^t) - \eta(g_i + g_k) \cdot g_k + O(\eta^2) \tag{26}$$

Similarly, when updating $i$ alone, the first-order Taylor approximation of $\mathcal{L}_k$ for $\Theta_s^t$ is as follows:

$$\mathcal{L}_k(z^t, \Theta_{s|i}^{t+1}, \Theta_k^t) = \mathcal{L}_k(z^t, \Theta_s^t, \Theta_k^t) + (\Theta_{s|i}^{t+1} - \Theta_s^t)\nabla_{\Theta_s^t}\mathcal{L}_k(z^t, \Theta_s^t, \Theta_k^t) + O(\eta^2) \tag{27}$$

$$= \mathcal{L}_k(z^t, \Theta_s^t, \Theta_k^t) - \eta g_i \cdot g_k + O(\eta^2) \tag{28}$$

With a sufficiently small learning rate $\eta$, the difference between the two inter-task affinities in eq. (24) can be expressed as follows:

$$\mathcal{A}_{i,k\to k}^t - \mathcal{A}_{i\to k}^t = 1 - \frac{\mathcal{L}_k(z^t, \Theta_{s|i,k}^{t+1}, \Theta_k^t)}{\mathcal{L}_k(z^t, \Theta_s^t, \Theta_k^t)} - \left(1 - \frac{\mathcal{L}_k(z^t, \Theta_{s|i}^{t+1}, \Theta_k^t)}{\mathcal{L}_k(z^t, \Theta_s^t, \Theta_k^t)}\right) \tag{29}$$

$$= \frac{\mathcal{L}_k(z^t, \Theta_{s|i}^{t+1}, \Theta_k^t) - \mathcal{L}_k(z^t, \Theta_{s|i,k}^{t+1}, \Theta_k^t)}{\mathcal{L}_k(z^t, \Theta_s^t, \Theta_k^t)} \tag{30}$$

$$= \frac{\mathcal{L}_k(z^t, \Theta_s^t, \Theta_k^t) - \eta g_i \cdot g_k - (\mathcal{L}_k(z^t, \Theta_s^t, \Theta_k^t) - \eta(g_i + g_k) \cdot g_k)}{\mathcal{L}_k(z^t, \Theta_s^t, \Theta_k^t)} \tag{31}$$

$$= \frac{\eta \|g_k\|^2}{\mathcal{L}_k(z^t, \Theta_s^t, \Theta_k^t)} \tag{32}$$

$$\geq 0 \tag{33}$$

The inequality in eq. (33) proves that $\mathcal{A}_{i,k\to k}^t \geq \mathcal{A}_{i\to k}^t$. $\qquad\square$

When tasks $i$ and $k$ are within the same task group, we can access $\mathcal{B}_{i,k\to k}^t$ during the optimization process. If $\mathcal{B}_{i,k\to k}^t \leq 0$, the inter-task affinity also satisfies $\mathcal{A}_{i,k\to k}^t \leq 0$ as $\mathcal{A}_{i,k\to k}^t \leq \mathcal{B}_{i,k\to k}^t$. According to Theorem 3, this condition implies $\mathcal{A}_{i\to k}^t \leq 0$. The proposed algorithm separates these tasks into different groups when $\mathcal{B}_{i,k\to k}^t \leq 0$ which justifies our grouping rules.

Conversely, when tasks $i$ and $j$ belong to separate task groups, we only have access to $\mathcal{B}_{i\to k}^t$ instead of $\mathcal{B}_{i,k\to k}^t$. In this scenario, the proposed algorithm merges these tasks into the same group if $\mathcal{B}_{i\to k}^t = \mathcal{A}_{i\to k}^t \geq 0$. This inequality also implies $\mathcal{A}_{i,k\to k}^t \geq 0$, justifying the merging of tasks $i$ and $k$ based on $\mathcal{B}_{i\to k}^t$ during optimization.

A.5 PROOF OF THEOREM 4

**Theorem 4** (Convergence Analysis). *Given a set of differentiable losses $\{\mathcal{L}_i\}_{i=1}^{\mathcal{K}}$ and Lipschitz continuous gradients with a constant $H > 0$, $||\nabla\mathcal{L}_i(x) - \nabla\mathcal{L}_i(y)|| \leq H||x - y||$ for all $i = 1, 2, ..., \mathcal{K}$. If we sequentially update task groups $\{G_i\}_{i=1}^{\mathcal{M}}$ based on inter-task affinity, employing a sufficiently small step size $\eta \leq \min(\frac{2}{H\mathcal{K}}, \frac{1}{H|G_m|})$ where $|G_m|$ is the number of tasks in $G_m$, then the following inequalities are satisfied for any task group $G_m$.*

$$\sum_{k=1}^{\mathcal{M}} \mathcal{L}_{G_k}(z^t, \Theta_{s|G_m}^{t+m/\mathcal{M}}, \Theta_{G_k}^{t+1}) \leq \sum_{k=1}^{\mathcal{M}} \mathcal{L}_{G_k}(z^t, \Theta_{s|G_m}^{t+(m-1)/\mathcal{M}}, \Theta_{G_k}^t) \tag{6}$$

$$- \eta g_{s,G_m}^{t+(m-1)/\mathcal{M}} \cdot (\sum_{k=1}^{\mathcal{M}} g_{s,G_k}^{t+(m-1)/\mathcal{M}} - g_{s,G_m}^{t+(m-1)/\mathcal{M}}) - \frac{\eta}{2}||g_{ts,G_m}^t||^2 \tag{7}$$

Let's represent the sum of losses of tasks included in $G_m$ as $\mathcal{L}_{G_m}$, defined as follows:

$$\mathcal{L}_{G_m}(z^t, \Theta_{s|G_m}^{t+m/\mathcal{M}}, \Theta_{G_m}^{t+m/\mathcal{M}}) = \sum_{k \in G_m} \mathcal{L}_k(z^t, \Theta_{s|G_m}^{t+m/\mathcal{M}}, \Theta_k^{t+m/\mathcal{M}}) \tag{34}$$

where $\Theta_{s|G_m}^{t+m/\mathcal{M}}$ denotes the shared parameters, while $\Theta_{G_m}^{t+m/\mathcal{M}}$ represents the set of task-specific parameters within $G_m$ after updating tasks in $G_m$.

We begin by expanding the task-specific loss $\mathcal{L}_{G_m}$ in terms of the shared parameter $\Theta_{s|G_m}^{t+m/\mathcal{M}}$ and the task-specific parameters $\Theta_{G_m}^{t+m/\mathcal{M}}$ using a quadratic expansion. During this process, the task-specific parameters $\Theta_{G_m}^{t+(m-1)/\mathcal{M}} = \Theta_{G_m}^t$ and $\Theta_{G_m}^{t+m/\mathcal{M}} = \Theta_{G_m}^{t+1}$, since the task-specific parameters in $G_m$ are updated only once from $\Theta_{G_m}^{t+(m-1)/\mathcal{M}}$ to $\Theta_{G_m}^{t+m/\mathcal{M}}$.

$$\mathcal{L}_{G_m}(z^t, \Theta_{s|G_m}^{t+m/\mathcal{M}}, \Theta_{G_m}^{t+1}) \leq \mathcal{L}_{G_m}(z^t, \Theta_{s|G_{m-1}}^{t+(m-1)/\mathcal{M}}, \Theta_{G_m}^t) \tag{35}$$

$$+ \nabla_{\Theta_{s|G_{m-1}}^{t+(m-1)/\mathcal{M}}} \mathcal{L}_{G_m}(z^t, \Theta_{s|G_{m-1}}^{t+(m-1)/\mathcal{M}}, \Theta_{G_m}^t)(\Theta_{s|G_m}^{t+m/\mathcal{M}} - \Theta_{s|G_{m-1}}^{t+(m-1)/\mathcal{M}}) \tag{36}$$

$$+ \frac{1}{2}\nabla^2_{\Theta_{s|G_{m-1}}^{t+(m-1)/\mathcal{M}}} \mathcal{L}_{G_m}(z^t, \Theta_{s|G_{m-1}}^{t+(m-1)/\mathcal{M}}, \Theta_{G_m}^t)(\Theta_{s|G_m}^{t+m/\mathcal{M}} - \Theta_{s|G_m}^{t+(m-1)/\mathcal{M}})^2 \tag{37}$$

$$+ \nabla_{\Theta_{G_m}^t} \mathcal{L}_{G_m}(z^t, \Theta_{s|G_{m-1}}^{t+(m-1)/\mathcal{M}}, \Theta_{G_m}^t)(\Theta_{G_m}^{t+1} - \Theta_{G_m}^t) \tag{38}$$

$$+ \frac{1}{2}\nabla^2_{\Theta_{G_m}^t} \mathcal{L}_{G_m}(z^t, \Theta_{s|G_{m-1}}^{t+(m-1)/\mathcal{M}}, \Theta_{G_m}^t)(\Theta_{G_m}^{t+1} - \Theta_{G_m}^t)^2 \tag{39}$$

$$\leq \mathcal{L}_{G_m}(z^t, \Theta_{s|G_m}^{t+(m-1)/\mathcal{M}}, \Theta_{G_m}^t) \tag{40}$$

$$+ \nabla_{\Theta_{s|G_{m-1}}^{t+(m-1)/\mathcal{M}}} \mathcal{L}_k(z^t, \Theta_{s|G_{m-1}}^{t+(m-1)/\mathcal{M}}, \Theta_{G_m}^t)(\Theta_{s|G_m}^{t+m/\mathcal{M}} - \Theta_{s|G_{m-1}}^{t+(m-1)/\mathcal{M}}) \tag{41}$$

$$+ \frac{1}{2}H|G_m|(\Theta_{s|G_m}^{t+m/\mathcal{M}} - \Theta_{s|G_{m-1}}^{t+(m-1)/\mathcal{M}})^2 \tag{42}$$

$$+ \nabla_{\Theta_{G_m}^t} \mathcal{L}_{G_m}(z^t, \Theta_{s|G_{m-1}}^{t+(m-1)/\mathcal{M}}, \Theta_{G_m}^t)(\Theta_{G_m}^{t+1} - \Theta_{G_m}^t) \tag{43}$$

$$+ \frac{1}{2}H|G_m|(\Theta_{G_m}^{t+1} - \Theta_{G_m}^t)^2 \tag{44}$$

where $|G_m|$ represents the number of tasks in $G_m$. The inequality holds with the Lipschitz continuity of $\nabla\mathcal{L}$ with a constant $H$.

For the shared parameters of the network, the update rule is as follows:

$$\Theta_{s|G_m}^{t+m/\mathcal{M}} = \Theta_{s|G_{m-1}}^{t+(m-1)/\mathcal{M}} - \eta\nabla_{\Theta_{s|G_{m-1}}^{t+(m-1)/\mathcal{M}}} \mathcal{L}_{G_m}(z^t, \Theta_{s|G_{m-1}}^{t+(m-1)/\mathcal{M}}, \Theta_{G_m}^t) \tag{45}$$

$$= \Theta_{s|G_{m-1}}^{t+(m-1)/\mathcal{M}} - \eta g_{s,G_m}^{t+(m-1)/\mathcal{M}} \tag{46}$$

where $g_{s,G_m}^{t+(m-1)/\mathcal{M}}$ is the gradients of the shared parameters with respect to loss of tasks in $G_m$.

Similarly, the task-specific parameters of the network, the update rule is as follows:

$$\Theta_{G_m}^{t+1} = \Theta_{G_m}^t - \eta \nabla_{\Theta_{G_m}^t} \mathcal{L}_{G_m}(z^t, \Theta_{s|G_{m-1}}^{t+(m-1)/\mathcal{M}}, \Theta_{G_m}^t) = \Theta_{G_m}^t - \eta g_{ts,G_m}^t \tag{47}$$

where $g_{ts,G_m}^t$ is the gradients of the task-specific parameters with respect to the loss of tasks in $G_m$.

If we substitute eq. (46) and eq. (47) into the result of eq. (35), it become as follows:

$$\mathcal{L}_{G_m}(z^t, \Theta_{s|G_m}^{t+m/\mathcal{M}}, \Theta_{G_m}^{t+1}) \leq \mathcal{L}_{G_m}(z^t, \Theta_{s|G_{m-1}}^{t+(m-1)/\mathcal{M}}, \Theta_{G_m}^t) \tag{48}$$

$$- \eta \|g_{s,G_m}^{t+(m-1)/\mathcal{M}}\|^2 + \frac{\eta^2 H |G_m|}{2} \|g_{s,G_m}^{t+(m-1)/\mathcal{M}}\|^2 \tag{49}$$

$$- \eta \|g_{ts,G_m}^t\|^2 + \frac{\eta^2 H |G_m|}{2} \|g_{ts,G_m}^t\|^2 \tag{50}$$

We can derive similar results for the loss of task group $G_i$, where the index $i$ is not the same as the updating group sequence $m$ ($i \neq m$). This process follows similarly to the one described above. For the step from $t + (m-1)/\mathcal{M}$ to $t + m/\mathcal{M}$, the task-specific parameters in $G_i$ remain unchanged.

$$\mathcal{L}_{G_i}(z^t, \Theta_{s|G_m}^{t+m/\mathcal{M}}, \Theta_{G_i}^t) \leq \mathcal{L}_{G_i}(z^t, \Theta_{s|G_{m-1}}^{t+(m-1)/\mathcal{M}}, \Theta_{G_i}^t) \tag{51}$$

$$+ \nabla_{\Theta_{s|G_{m-1}}^{t+(m-1)/\mathcal{M}}} \mathcal{L}_{G_i}(z^t, \Theta_{s|G_{m-1}}^{t+(m-1)/\mathcal{M}}, \Theta_{G_i}^t)(\Theta_{s|G_m}^{t+m/\mathcal{M}} - \Theta_{s|G_{m-1}}^{t+(m-1)/\mathcal{M}}) \tag{52}$$

$$+ \frac{1}{2} \nabla_{\Theta_{s|G_{m-1}}^{t+(m-1)/\mathcal{M}}}^2 \mathcal{L}_{G_i}(z^t, \Theta_{s|G_{m-1}}^{t+(m-1)/\mathcal{M}}, \Theta_{G_i}^t)(\Theta_{s|G_m}^{t+m/\mathcal{M}} - \Theta_{s|G_m}^{t+(m-1)/\mathcal{M}})^2 \tag{53}$$

$$\leq \mathcal{L}_{G_i}(z^t, \Theta_{s|G_m}^{t+(m-1)/\mathcal{M}}, \Theta_{G_i}^t) \tag{54}$$

$$+ \nabla_{\Theta_{s|G_{m-1}}^{t+(m-1)/\mathcal{M}}} \mathcal{L}_{G_i}(z^t, \Theta_{s|G_{m-1}}^{t+(m-1)/\mathcal{M}}, \Theta_{G_i}^t)(\Theta_{s|G_m}^{t+m/\mathcal{M}} - \Theta_{s|G_{m-1}}^{t+(m-1)/\mathcal{M}}) \tag{55}$$

$$+ \frac{1}{2} H |G_i|(\Theta_{s|G_m}^{t+m/\mathcal{M}} - \Theta_{s|G_{m-1}}^{t+(m-1)/\mathcal{M}})^2 \tag{56}$$

$$\leq \mathcal{L}_{G_i}(z^t, \Theta_{s|G_m}^{t+(m-1)/\mathcal{M}}, \Theta_{G_i}^t) \tag{57}$$

$$- \eta g_{s,G_i}^{t+(m-1)/\mathcal{M}} \cdot g_{s,G_m}^{t+(m-1)/\mathcal{M}} + \frac{\eta^2 H |G_i|}{2} \|g_{s,G_m}^{t+(m-1)/\mathcal{M}}\|^2 \tag{58}$$

$$\tag{59}$$

Then the total loss of multiple task groups can be expressed as follows:

$$\sum_{k=1}^{\mathcal{M}} \mathcal{L}_{G_k}(z^t, \Theta_{s|G_m}^{t+m/\mathcal{M}}, \Theta_{G_k}^{t+1}) \leq \sum_{k=1}^{\mathcal{M}} \mathcal{L}_{G_k}(z^t, \Theta_{s|G_m}^{t+(m-1)/\mathcal{M}}, \Theta_{G_k}^t) \tag{60}$$

$$- \eta \sum_{k=1}^{\mathcal{M}} g_{s,G_k}^{t+(m-1)/\mathcal{M}} \cdot g_{s,G_m}^{t+(m-1)/\mathcal{M}} + \frac{\eta^2 H}{2} ||g_{s,G_m}^{t+(m-1)/\mathcal{M}}||^2 \sum_{k=1}^{\mathcal{M}} |G_k| \tag{61}$$

$$- \eta ||g_{ts,G_m}^t||^2 + \frac{\eta^2 H |G_m|}{2} ||g_{ts,G_m}^t||^2 \tag{62}$$

$$\leq \sum_{k=1}^{\mathcal{M}} \mathcal{L}_{G_k}(z^t, \Theta_{s|G_m}^{t+(m-1)/\mathcal{M}}, \Theta_{G_k}^t) \tag{63}$$

$$- \eta \sum_{k=1}^{\mathcal{M}} g_{s,G_k}^{t+(m-1)/\mathcal{M}} \cdot g_{s,G_m}^{t+(m-1)/\mathcal{M}} + \eta ||g_{s,G_m}^{t+(m-1)/\mathcal{M}}||^2 - \frac{\eta}{2} ||g_{ts,G_m}^t||^2 \tag{64}$$

$$= \sum_{k=1}^{\mathcal{M}} \mathcal{L}_{G_k}(z^t, \Theta_{s|G_m}^{t+(m-1)/\mathcal{M}}, \Theta_{G_k}^t) \tag{65}$$

$$- \eta g_{s,G_m}^{t+(m-1)/\mathcal{M}} \cdot (\sum_{k=1}^{\mathcal{M}} g_{s,G_k}^{t+(m-1)/\mathcal{M}} - g_{s,G_m}^{t+(m-1)/\mathcal{M}}) - \frac{\eta}{2} ||g_{ts,G_m}^t||^2 \tag{66}$$

The inequality between eq. (61) and the first term in eq. (64) requires $\eta \leq \frac{2}{H \cdot \sum_{k=1}^{\mathcal{M}} |G_k|} = \frac{2}{H\mathcal{K}}$, while the inequality between eq. (62) and the second term in eq. (64) requires $\eta \leq \frac{1}{H|G_M|}$. Therefore, the inequality in eq. (63) holds when $\eta \leq \min(\frac{2}{H\mathcal{K}}, \frac{1}{H|G_M|})$. Previous approaches, which handle updates of shared and task-specific parameters independently, failing to capture their interdependence during optimization. The term, $g_{s,G_m}^{t+(m-1)/\mathcal{M}} \cdot (\sum_{k=1}^{\mathcal{M}} g_{s,G_k}^{t+(m-1)/\mathcal{M}})$, fluctuates during optimization. When the gradients of group $G_m$ align well with the gradients of the other groups $\{G_k\}_{i=1,i\neq m}^{\mathcal{M}}$, their dot product yields a positive value, leading to a decrease in multi-task losses. However, in practice, the sequential update strategy demonstrates a similar level of stability in optimization, which appears to contradict the conventional results. Thus, we assume a correlation between the learning of shared parameters and task-specific parameters, where the learning of task-specific parameters reduces gradient conflicts in shared parameters. Under this assumption, the sequential update strategy can guarantee convergence to Pareto-stationary points. This assumption is reasonable, as task-specific parameters capture task-specific information, thereby reducing conflicts in the shared parameters across tasks.

## A.6 Two-Step Proximal Inter-task Affinity

Before delving into the proof of Theorem 5, let's introduce the concept of two-step proximal inter-task affinity, which extends the notion of proximal inter-task affinity over two update steps.

**Definition 4** (Two-Step Proximal Inter-Task Affinity). *Consider a multi-task network shared by the tasks $i, j, k$, with their respective losses denoted as $\mathcal{L}_i, \mathcal{L}_j, \mathcal{L}_k$. Sequential updates of $(\{j\}, \{i, k\})$ result in parameters being updated from $\Theta_s^t \to \Theta_{s|j}^{t+1} \to \Theta_{s|i,k}^{t+2}$ and $\Theta_k^t \to \Theta_k^{t+1} \to \Theta_k^{t+2}$. Then, the two-step proximal inter-task affinity from sequential update $(\{j\}, \{i, k\})$ to $k$ at time step $t$ is defined as follows:*

$$\mathcal{B}_{j;i,k\to k}^t = 1 - (1 - \mathcal{B}_{j\to k}^t)(1 - \mathcal{B}_{i,k\to k}^{t+1}) \tag{67}$$

$$= 1 - \frac{\mathcal{L}_k(z^t, \Theta_{s|j}^{t+1}, \Theta_k^{t+1})}{\mathcal{L}_k(z^t, \Theta_s^t, \Theta_k^t)} \cdot \frac{\mathcal{L}_k(z^t, \Theta_{s|i,k}^{t+2}, \Theta_k^{t+2})}{\mathcal{L}_k(z^t, \Theta_{s|j}^{t+1}, \Theta_k^{t+1})} = 1 - \frac{\mathcal{L}_k(z^t, \Theta_{s|i,k}^{t+2}, \Theta_k^{t+2})}{\mathcal{L}_k(z^t, \Theta_s^t, \Theta_k^t)} \tag{68}$$

## A.7 Proof of Theorem 5

**Theorem 5.** *Consider three tasks $\{i, j, k\}$, where task groups are formed with positive inter-task affinity as $\{i, k\}$ and $\{j\}$. Assume all losses are convex and differentiable, and the change in affinity*

*during a single step from $t + (m-1)/\mathcal{M}$ to $t + m/\mathcal{M}$ is negligible. The affinity, which learns all tasks jointly, is denoted as $\mathcal{B}_{i,j,k\to k}^{t+(m-1)/\mathcal{M}}$. The affinity for the updating sequence $(\{i,k\}, \{j\})$ is represented as $\mathcal{B}_{i,k;j\to k}^{t+(m-1)/\mathcal{M}}$, while for the sequence $(\{j\}, \{i,k\})$, it is represented as $\mathcal{B}_{j;i,k\to k}^{t+(m-1)/\mathcal{M}}$. Then, for a sufficiently small learning rate $\eta \ll 1$, the following holds:*

$$\mathcal{B}_{i,j,k\to k}^{t+(m-1)/\mathcal{M}} \simeq \mathcal{B}_{i,k;j\to k}^{t+(m-1)/\mathcal{M}} \qquad\qquad \mathcal{B}_{i,j,k\to k}^{t+(m-1)/\mathcal{M}} \leq \mathcal{B}_{j;i,k\to k}^{t+(m-1)/\mathcal{M}} \tag{8}$$

*Proof.* We compare the loss after jointly updating three tasks $\{i,j,k\}$ with the loss after sequentially updating the task sets $\{i,k\}$ and $\{j\}$. To assess the impact of the updating order of task sets, we also conduct the analysis on the reverse order of task set $\{j\}, \{i,k\}$.

(i) Let's begin with the definition of proximal inter-task affinity between $\{i,k\} \to k$ and $j \to k$, taking into account the updates of task-specific parameters as follows:

$$\mathcal{B}_{i,j,k\to k}^{t+(m-1)/\mathcal{M}} = 1 - \frac{\mathcal{L}_k(z^t, \Theta_{s|i,j,k}^{t+m/\mathcal{M}}, \Theta_k^{t+m/\mathcal{M}})}{\mathcal{L}_k(z^t, \Theta_s^{t+(m-1)/\mathcal{M}}, \Theta_k^{t+(m-1)/\mathcal{M}})} \tag{69}$$

$$\mathcal{B}_{i,k\to k}^{t+(m-1)/\mathcal{M}} = 1 - \frac{\mathcal{L}_k(z^t, \Theta_{s|i,k}^{t+m/\mathcal{M}}, \hat{\Theta}_k^{t+m/\mathcal{M}})}{\mathcal{L}_k(z^t, \Theta_s^{t+(m-1)/\mathcal{M}}, \Theta_k^{t+(m-1)/\mathcal{M}})} \tag{70}$$

$$\mathcal{B}_{j\to k}^{t+m/\mathcal{M}} = 1 - \frac{\mathcal{L}_k(z^t, \Theta_{s|j}^{t+(m+1)/\mathcal{M}}, \Theta_k^{t+(m+1)/\mathcal{M}})}{\mathcal{L}_k(z^t, \Theta_{s|i,k}^{t+m/\mathcal{M}}, \hat{\Theta}_k^{t+m/\mathcal{M}})} \tag{71}$$

$$= 1 - \frac{\mathcal{L}_k(z^t, \Theta_{s|j}^{t+(m+1)/\mathcal{M}}, \hat{\Theta}_k^{t+m/\mathcal{M}})}{\mathcal{L}_k(z^t, \Theta_{s|i,k}^{t+m/\mathcal{M}}, \hat{\Theta}_k^{t+m/\mathcal{M}})} \tag{72}$$

where $\hat{\Theta}_k^{t+m/\mathcal{M}}$ represents the resulting task-specific parameters of $k$ immediately after updating the task set $\{i,j\}$. This notation is used to differentiate it from the task-specific parameter $\Theta_k^{t+m/\mathcal{M}}$ obtained after jointly updating all tasks.

The two-step proximal inter-task affinity with the sequence $\{i,k\}$ and $\{j\}$ can be represented as follows:

$$\mathcal{B}_{i,k;j\to k}^{t+(m-1)/\mathcal{M}} = 1 - (1 - \mathcal{B}_{i,k\to k}^{t+(m-1)/\mathcal{M}}) \cdot (1 - \mathcal{B}_{j\to k}^{t+m/\mathcal{M}}) \tag{73}$$

$$= 1 - \frac{\mathcal{L}_k(z^t, \Theta_{s|i,k}^{t+m/\mathcal{M}}, \hat{\Theta}_k^{t+m/\mathcal{M}})}{\mathcal{L}_k(z^t, \Theta_s^{t+(m-1)/\mathcal{M}}, \Theta_k^{t+(m-1)/\mathcal{M}})} \cdot \frac{\mathcal{L}_k(z^t, \Theta_{s|j}^{t+(m+1)/\mathcal{M}}, \hat{\Theta}_k^{t+m/\mathcal{M}})}{\mathcal{L}_k(z^t, \Theta_{s|i,k}^{t+m/\mathcal{M}}, \hat{\Theta}_k^{t+m/\mathcal{M}})} \tag{74}$$

$$= 1 - \frac{\mathcal{L}_k(z^t, \Theta_{s|j}^{t+(m+1)/\mathcal{M}}, \hat{\Theta}_k^{t+m/\mathcal{M}})}{\mathcal{L}_k(z^t, \Theta_s^{t+(m-1)/\mathcal{M}}, \Theta_k^{t+(m-1)/\mathcal{M}})} \tag{75}$$

Our objective is to compare $\mathcal{B}_{i,j,k\to k}^{t+(m-1)/\mathcal{M}}$ from eq. (69) with $\mathcal{B}_{i,k;j\to k}^{t+(m-1)/\mathcal{M}}$ from eq. (75) to assess each update's effect on the final loss. Since both equations share a common denominator, we only need to compare the numerators of each equation. Using the first-order Taylor approximation of $\mathcal{L}_k(z^t, \Theta_{s|j}^{t+(m+1)/\mathcal{M}}, \hat{\Theta}_k^{t+m/\mathcal{M}})$ in eq. (75), we have:

$$\mathcal{L}_k(z^t, \Theta_{s|j}^{t+(m+1)/\mathcal{M}}, \hat{\Theta}_k^{t+m/\mathcal{M}}) = \mathcal{L}_k(z^t, \Theta_{s|i,k}^{t+m/\mathcal{M}}, \hat{\Theta}_k^{t+m/\mathcal{M}}) \tag{76}$$

$$+ (\Theta_{s|j}^{t+(m+1)/\mathcal{M}} - \Theta_{s|i,k}^{t+m/\mathcal{M}})\nabla_{\Theta_{s|i,k}^{t+m/\mathcal{M}}}\mathcal{L}_k(z^t, \Theta_{s|i,k}^{t+m/\mathcal{M}}, \hat{\Theta}_k^{t+m/\mathcal{M}}) + O(\eta^2) \tag{77}$$

$$= \mathcal{L}_k(z^t, \Theta_{s|i,k}^{t+m/\mathcal{M}}, \hat{\Theta}_k^{t+m/\mathcal{M}}) - \eta g_{s;j}^{t+m/\mathcal{M}} \cdot g_{s;k}^{t+m/\mathcal{M}} + O(\eta^2) \tag{78}$$

The subscript $s$ in gradients indicates that it represents the gradients of the shared parameters of the network. Conversely, we will use the subscript $ts$ for gradients of the task-specific network in the following derivation. Similarly, $\mathcal{L}_k(z^t, \Theta_{s|i,k}^{t+m/\mathcal{M}}, \hat{\Theta}_k^{t+m/\mathcal{M}})$ in eq. (76) can also be further expanded using Taylor expansion as follows:

$$\mathcal{L}_k(z^t, \Theta_{s|i,k}^{t+m/\mathcal{M}}, \hat{\Theta}_k^{t+m/\mathcal{M}}) = \mathcal{L}_k(z^t, \Theta_s^{t+(m-1)/\mathcal{M}}, \Theta_k^{t+(m-1)/\mathcal{M}}) \tag{79}$$

$$+ (\Theta_{s|i,k}^{t+m/\mathcal{M}} - \Theta_s^{t+(m-1)/\mathcal{M}}) \nabla_{\Theta_s^{t+(m-1)/\mathcal{M}}} \mathcal{L}_k(z^t, \Theta_s^{t+(m-1)/\mathcal{M}}, \Theta_k^{t+(m-1)/\mathcal{M}}) \tag{80}$$

$$+ (\hat{\Theta}_k^{t+m/\mathcal{M}} - \Theta_k^{t+(m-1)/\mathcal{M}}) \nabla_{\Theta_k^{t+(m-1)/\mathcal{M}}} \mathcal{L}_k(z^t, \Theta_s^{t+(m-1)/\mathcal{M}}, \Theta_k^{t+(m-1)/\mathcal{M}}) + O(\eta^2) \tag{81}$$

$$=\mathcal{L}_k(z^t, \Theta_s^{t+(m-1)/\mathcal{M}}, \Theta_k^{t+(m-1)/\mathcal{M}}) - \eta(g_{s;i}^{t+(m-1)/\mathcal{M}} + g_{s;k}^{t+(m-1)/\mathcal{M}}) \cdot g_{s;k}^{t+(m-1)/\mathcal{M}} \tag{82}$$

$$- \eta g_{ts;k}^{t+(m-1)/\mathcal{M}} \cdot g_{ts;k}^{t+(m-1)/\mathcal{M}} + O(\eta^2) \tag{83}$$

By substituting eq. (76) with the results of eq. (82) and eq. (83), we can obtain the following results:

$$\mathcal{L}_k(z^t, \Theta_{s|j}^{t+(m+1)/\mathcal{M}}, \hat{\Theta}_k^{t+m/\mathcal{M}}) = \mathcal{L}_k(z^t, \Theta_s^{t+(m-1)/\mathcal{M}}, \Theta_k^{t+(m-1)/\mathcal{M}}) \tag{84}$$

$$- \eta g_{s;j}^{t+m/\mathcal{M}} \cdot g_{s;k}^{t+m/\mathcal{M}} - \eta(g_{s;i}^{t+(m-1)/\mathcal{M}} + g_{s;k}^{t+(m-1)/\mathcal{M}}) \cdot g_{s;k}^{t+(m-1)/\mathcal{M}} \tag{85}$$

$$- \eta g_{ts;k}^{t+(m-1)/\mathcal{M}} \cdot g_{ts;k}^{t+(m-1)/\mathcal{M}} + O(\eta^2) \tag{86}$$

For the scenario where all tasks $\{i, j, k\}$ are jointly updated, the numerator of eq. (69) can also be expanded as follows:

$$\mathcal{L}_k(z^t, \Theta_{s|i,j,k}^{t+m/\mathcal{M}}, \Theta_k^{t+m/\mathcal{M}}) = \mathcal{L}_k(z^t, \Theta_s^{t+(m-1)/\mathcal{M}}, \Theta_k^{t+(m-1)/\mathcal{M}}) \tag{87}$$

$$+ (\Theta_{s|i,j,k}^{t+m/\mathcal{M}} - \Theta_s^{t+(m-1)/\mathcal{M}}) \nabla_{\Theta_s^{t+(m-1)/\mathcal{M}}} \mathcal{L}_k(z^t, \Theta_s^{t+(m-1)/\mathcal{M}}, \Theta_k^{t+(m-1)/\mathcal{M}}) \tag{88}$$

$$+ (\Theta_k^{t+m/\mathcal{M}} - \Theta_k^{t+(m-1)/\mathcal{M}}) \nabla_{\Theta_k^{t+(m-1)/\mathcal{M}}} \mathcal{L}_k(z^t, \Theta_s^{t+(m-1)/\mathcal{M}}, \Theta_k^{t+(m-1)/\mathcal{M}}) + O(\eta^2) \tag{89}$$

$$=\mathcal{L}_k(z^t, \Theta_s^{t+(m-1)/\mathcal{M}}, \Theta_k^{t+(m-1)/\mathcal{M}}) \tag{90}$$

$$- \eta(g_{s,i}^{t+(m-1)/\mathcal{M}} + g_{s,j}^{t+(m-1)/\mathcal{M}} + g_{s,k}^{t+(m-1)/\mathcal{M}}) \cdot g_{s,k}^{t+(m-1)/\mathcal{M}} \tag{91}$$

$$- \eta g_{ts,k}^{t+(m-1)/\mathcal{M}} \cdot g_{ts,k}^{t+(m-1)/\mathcal{M}} + O(\eta^2) \tag{92}$$

Finally, we can compare $\mathcal{B}_{i,j,k \to k}^{t+(m-1)/\mathcal{M}}$ with $\mathcal{B}_{i,k;j \to k}^{t+(m-1)/\mathcal{M}}$ by comparing the losses we obtained: $\mathcal{L}_k(z^t, \Theta_{s|j}^{t+(m+1)/\mathcal{M}}, \Theta_k^{t+(m-1)/\mathcal{M}})$ with $\mathcal{L}_k(z^t, \Theta_{s|i,j,k}^{t+m/\mathcal{M}}, \Theta_k^{t+(m-1)/\mathcal{M}})$. We assume a sufficiently small learning rate $\eta$ that allows us to ignore terms larger than order two with $\eta$.

$$\mathcal{L}_k(z^t, \Theta_{s|j}^{t+(m+1)/\mathcal{M}}, \hat{\Theta}_k^{t+m/\mathcal{M}}) - \mathcal{L}_k(z^t, \Theta_{s|i,j,k}^{t+m/\mathcal{M}}, \Theta_k^{t+m/\mathcal{M}}) \tag{93}$$

$$= - \eta g_{s,j}^{t+m/\mathcal{M}} \cdot g_{s,k}^{t+m/\mathcal{M}} - \eta(g_{s,i}^{t+(m-1)/\mathcal{M}} + g_{s,k}^{t+(m-1)/\mathcal{M}}) \cdot g_{s,k}^{t+(m-1)/\mathcal{M}} \tag{94}$$

$$- \eta g_{ts,k}^{t+(m-1)/\mathcal{M}} \cdot g_{ts,k}^{t+(m-1)/\mathcal{M}} \tag{95}$$

$$+ \eta(g_{s,i}^{t+(m-1)/\mathcal{M}} + g_{s,j}^{t+(m-1)/\mathcal{M}} + g_{s,k}^{t+(m-1)/\mathcal{M}}) \cdot g_{s,k}^{t+(m-1)/\mathcal{M}} \tag{96}$$

$$+ \eta g_{ts,k}^{t+(m-1)/\mathcal{M}} \cdot g_{ts,k}^{t+(m-1)/\mathcal{M}} \tag{97}$$

$$= \eta(g_{s,j}^{t+(m-1)/\mathcal{M}} \cdot g_{s,k}^{t+(m-1)/\mathcal{M}} - g_{s,j}^{t+m/\mathcal{M}} \cdot g_{s,k}^{t+m/\mathcal{M}}) \tag{98}$$

$$\simeq 0 \tag{99}$$

The approximation in eq. (99) holds as we assume inter-task affinity change during a single time step from $t + (m-1)/\mathcal{M}$ to $t + m/\mathcal{M}$ is negligible.

(ii) In case we update task groups in reverse order the results would differ with (i). Similarly, we begin with the definition of proximal inter-task affinity with reverse order between $j \to k$ and $\{i, j\} \to k$ as follows:

$$\mathcal{B}_{j \to k}^{t+(m-1)/\mathcal{M}} = 1 - \frac{\mathcal{L}_k(z^t, \Theta_{s|j}^{t+m/\mathcal{M}}, \Theta_k^{t+m/\mathcal{M}})}{\mathcal{L}_k(z^t, \Theta_{s|i,k}^{t+(m-1)/\mathcal{M}}, \Theta_k^{t+(m-1)/\mathcal{M}})} \tag{100}$$

$$= 1 - \frac{\mathcal{L}_k(z^t, \Theta_{s|j}^{t+m/\mathcal{M}}, \Theta_k^{t+(m-1)/\mathcal{M}})}{\mathcal{L}_k(z^t, \Theta_s^{t+(m-1)/\mathcal{M}}, \Theta_k^{t+(m-1)/\mathcal{M}})} \tag{101}$$

$$\mathcal{B}_{i,k \to k}^{t+m/\mathcal{M}} = 1 - \frac{\mathcal{L}_k(z^t, \Theta_{s|i,k}^{t+(m+1)/\mathcal{M}}, \hat{\Theta}_k^{t+(m+1)/\mathcal{M}})}{\mathcal{L}_k(z^t, \Theta_{s|j}^{t+m/\mathcal{M}}, \Theta_k^{t+m/\mathcal{M}})} \tag{102}$$

$$= 1 - \frac{\mathcal{L}_k(z^t, \Theta_{s|i,k}^{t+(m+1)/\mathcal{M}}, \hat{\Theta}_k^{t+(m+1)/\mathcal{M}})}{\mathcal{L}_k(z^t, \Theta_{s|j}^{t+m/\mathcal{M}}, \Theta_k^{t+(m-1)/\mathcal{M}})} \tag{103}$$

The two-step proximal inter-task affinity with the sequence $\{j\}$ and $\{i, k\}$ can be represented as follows:

$$\mathcal{B}_{j;i,k \to k}^{t+(m-1)/\mathcal{M}} = 1 - (1 - \mathcal{B}_{j \to k}^{t+(m-1)/\mathcal{M}}) \cdot (1 - \mathcal{B}_{i,k \to k}^{t+m/\mathcal{M}}) \tag{104}$$

$$= 1 - \frac{\mathcal{L}_k(z^t, \Theta_{s|j}^{t+m/\mathcal{M}}, \Theta_k^{t+(m-1)/\mathcal{M}})}{\mathcal{L}_k(z^t, \Theta_s^{t+(m-1)/\mathcal{M}}, \Theta_k^{t+(m-1)/\mathcal{M}})} \cdot \frac{\mathcal{L}_k(z^t, \Theta_{s|i,k}^{t+(m+1)/\mathcal{M}}, \hat{\Theta}_k^{t+(m+1)/\mathcal{M}})}{\mathcal{L}_k(z^t, \Theta_{s|j}^{t+m/\mathcal{M}}, \Theta_k^{t+(m-1)/\mathcal{M}})} \tag{105}$$

$$= 1 - \frac{\mathcal{L}_k(z^t, \Theta_{s|i,k}^{t+(m+1)/\mathcal{M}}, \hat{\Theta}_k^{t+(m+1)/\mathcal{M}})}{\mathcal{L}_k(z^t, \Theta_s^{t+(m-1)/\mathcal{M}}, \Theta_k^{t+(m-1)/\mathcal{M}})} \tag{106}$$

Our objective is to compare $\mathcal{B}_{i,j,k \to k}^{t+(m-1)/\mathcal{M}}$ from eq. (69) with $\mathcal{B}_{j;i,k \to k}^{t+(m-1)/\mathcal{M}}$ from eq. (106) to assess each update's effect on the final loss. Since both equations share a common denominator, we only need to compare the numerators of each equation. Using the first-order Taylor approximation of $\mathcal{L}_k(z^t, \Theta_{s|i,k}^{t+(m+1)/\mathcal{M}}, \hat{\Theta}_k^{t+(m+1)/\mathcal{M}})$ in eq. (106), we have:

$$\mathcal{L}_k(z^t, \Theta_{s|i,k}^{t+(m+1)/\mathcal{M}}, \hat{\Theta}_k^{t+(m+1)/\mathcal{M}}) = \mathcal{L}_k(z^t, \Theta_{s|j}^{t+m/\mathcal{M}}, \Theta_k^{t+m/\mathcal{M}}) \tag{107}$$

$$+ (\Theta_{s|i,k}^{t+(m+1)/\mathcal{M}} - \Theta_{s|j}^{t+m/\mathcal{M}}) \nabla_{\Theta_{s|j}^{t+m/\mathcal{M}}} \mathcal{L}_k(z^t, \Theta_{s|j}^{t+m/\mathcal{M}}, \Theta_k^{t+m/\mathcal{M}}) \tag{108}$$

$$+ (\hat{\Theta}_k^{t+(m+1)/\mathcal{M}} - \Theta_k^{t+m/\mathcal{M}}) \nabla_{\Theta_k^{t+m/\mathcal{M}}} \mathcal{L}_k(z^t, \Theta_{s|j}^{t+m/\mathcal{M}}, \Theta_k^{t+m/\mathcal{M}}) + O(\eta^2) \tag{109}$$

$$= \mathcal{L}_k(z^t, \Theta_{s|j}^{t+m/\mathcal{M}}, \Theta_k^{t+m/\mathcal{M}}) - \eta(g_{s;i}^{t+m/\mathcal{M}} + g_{s;k}^{t+m/\mathcal{M}}) \cdot g_{s;k}^{t+m/\mathcal{M}} \tag{110}$$

$$- \eta g_{ts;k}^{t+m/\mathcal{M}} \cdot g_{ts;k}^{t+m/\mathcal{M}} + O(\eta^2) \tag{111}$$

Similarly, $\mathcal{L}_k(z^t, \Theta_{s|i,k}^{t+m/\mathcal{M}}, \Theta_k^{t+m/\mathcal{M}})$ in eq. (107) can also be further expanded using Taylor expansion as follows:

$$\mathcal{L}_k(z^t, \Theta_{s|j}^{t+m/\mathcal{M}}, \Theta_k^{t+m/\mathcal{M}}) = \mathcal{L}_k(z^t, \Theta_{s|j}^{t+m/\mathcal{M}}, \Theta_k^{t+(m-1)/\mathcal{M}}) \tag{112}$$

$$= \mathcal{L}_k(z^t, \Theta_s^{t+(m-1)/\mathcal{M}}, \Theta_k^{t+(m-1)/\mathcal{M}}) \tag{113}$$

$$+ (\Theta_{s|j}^{t+m/\mathcal{M}} - \Theta_s^{t+(m-1)/\mathcal{M}}) \nabla_{\Theta_s^{t+(m-1)/\mathcal{M}}} \mathcal{L}_k(z^t, \Theta_{s|j}^{t+m/\mathcal{M}}, \Theta_k^{t+(m-1)/\mathcal{M}}) \tag{114}$$

$$= \mathcal{L}_k(z^t, \Theta_s^{t+(m-1)/\mathcal{M}}, \Theta_k^{t+(m-1)/\mathcal{M}}) \tag{115}$$

$$- \eta g_{s;j}^{t+(m-1)/\mathcal{M}} \cdot g_{s;k}^{t+(m-1)/\mathcal{M}} + O(\eta^2) \tag{116}$$

By substituting eq. (107) with the results of eq. (115) and eq. (116), we can obtain the following results:

$$\mathcal{L}_k(z^t, \Theta_{s|i,k}^{t+(m+1)/\mathcal{M}}, \hat{\Theta}_k^{t+(m+1)/\mathcal{M}}) = \mathcal{L}_k(z^t, \Theta_s^{t+(m-1)/\mathcal{M}}, \Theta_k^{t+(m-1)/\mathcal{M}}) \tag{117}$$

$$- \eta(g_{s;i}^{t+m/\mathcal{M}} + g_{s;k}^{t+m/\mathcal{M}}) \cdot g_{s;k}^{t+m/\mathcal{M}} - \eta g_{ts;k}^{t+m/\mathcal{M}} \cdot g_{ts;k}^{t+m/\mathcal{M}} \tag{118}$$

$$- \eta g_{s;j}^{t+(m-1)/\mathcal{M}} \cdot g_{s;k}^{t+(m-1)/\mathcal{M}} + O(\eta^2) \tag{119}$$

Finally, we can compare $\mathcal{B}_{i,j,k\to k}^{t+(m-1)/\mathcal{M}}$ with $\mathcal{B}_{j;i,k\to k}^{t+(m-1)/\mathcal{M}}$ by comparing the losses we obtained: $\mathcal{L}_k(z^t, \Theta_{s|j}^{t+(m+1)/\mathcal{M}}, \Theta_k^{t+(m-1)/\mathcal{M}})$ with $\mathcal{L}_k(z^t, \Theta_{s|i,j,k}^{t+m/\mathcal{M}}, \Theta_k^{t+(m-1)/\mathcal{M}})$. We assume a sufficiently small learning rate $\eta$ that allows us to ignore terms larger than order two with $\eta$.

$$\mathcal{L}_k(z^t, \Theta_{s|i,k}^{t+(m+1)/\mathcal{M}}, \hat{\Theta}_k^{t+(m+1)/\mathcal{M}}) - \mathcal{L}_k(z^t, \Theta_{s|i,j,k}^{t+m/\mathcal{M}}, \Theta_k^{t+m/\mathcal{M}}) \tag{120}$$

$$= - \eta(g_{s;i}^{t+m/\mathcal{M}} + g_{s;k}^{t+m/\mathcal{M}}) \cdot g_{s;k}^{t+m/\mathcal{M}} - \eta g_{ts;k}^{t+m/\mathcal{M}} \cdot g_{ts;k}^{t+m/\mathcal{M}} \tag{121}$$

$$- \eta g_{s;j}^{t+(m-1)/\mathcal{M}} \cdot g_{s;k}^{t+(m-1)/\mathcal{M}} \tag{122}$$

$$+ \eta(g_{s,i}^{t+(m-1)/\mathcal{M}} + g_{s,j}^{t+(m-1)/\mathcal{M}} + g_{s,k}^{t+(m-1)/\mathcal{M}}) \cdot g_{s,k}^{t+(m-1)/\mathcal{M}} \tag{123}$$

$$+ \eta g_{ts,k}^{t+(m-1)/\mathcal{M}} \cdot g_{ts,k}^{t+(m-1)/\mathcal{M}} \tag{124}$$

$$= - \eta(g_{s;i}^{t+m/\mathcal{M}} + g_{s;k}^{t+m/\mathcal{M}}) \cdot g_{s;k}^{t+m/\mathcal{M}} + \eta(g_{s,i}^{t+(m-1)/\mathcal{M}} + g_{s,k}^{t+(m-1)/\mathcal{M}}) \cdot g_{s,k}^{t+(m-1)/\mathcal{M}} \tag{125}$$

$$- \eta g_{ts;k}^{t+m/\mathcal{M}} \cdot g_{ts;k}^{t+m/\mathcal{M}} + \eta g_{ts,k}^{t+(m-1)/\mathcal{M}} \cdot g_{ts,k}^{t+(m-1)/\mathcal{M}} \tag{126}$$

$$\simeq - \eta g_{ts;k}^{t+m/\mathcal{M}} \cdot g_{ts;k}^{t+m/\mathcal{M}} + \eta g_{ts,k}^{t+(m-1)/\mathcal{M}} \cdot g_{ts,k}^{t+(m-1)/\mathcal{M}} \tag{127}$$

$$\leq 0 \tag{128}$$

The approximation in eq. (127) holds under the assumption that the change in inter-task affinity during a single time step from $t + (m-1)/\mathcal{M}$ to $t + m/\mathcal{M}$ is negligible. Since we assume convex loss functions, the magnitude of task-specific gradients $g_{ts;k}$ would increase after updating the loss of $j$, which exhibits negative inter-task affinity with $k$ ($\mathcal{A}_{j\to k} < 0$). Therefore, the inequality in eq. (128) holds. $\square$

This suggest that grouping tasks with proximal inter-task affinity and subsequently updating these groups sequentially result in lower multi-task loss compared to jointly backpropagating all tasks. This disparity arises because the network can discern superior task-specific parameters to accommodate task-specific information during sequential learning.

## B ADDITIONAL RELATED WORKS

**Task Grouping.** Early Multi-Task Learning research is founded on the belief that simultaneous learning of similar tasks within a multi-task framework can enhance overall performance. Kang et al. Kang et al. (2011) identify tasks that contribute to improved multi-task performance through the clustering of related tasks with online stochastic gradient descent. This strategy challenges the prevailing assumption that all tasks are inherently interrelated. In parallel, Kumar et al. Kumar & Daume III (2012) present a framework for MTL designed to enable selective sharing of information across tasks. They suggests that each task parameter vector can be expressed as a linear combination of a finite number of underlying basis tasks. However, these initial methodologies face limitations in their applicability and analysis, particularly in scaling to deep neural networks. Finding out related tasks is more dynamically explored in the transfer learning domain Achille et al. (2019; 2021). They find related tasks by measuring task similarity which can be comparing the similarity of features extracted from the same depth of the independent task's network or directly measuring the transfer performance between tasks. Recent research has concentrated on identifying related tasks by directly assessing the relations among them within shared networks. This focus stems from the recognition that the measured inter-task relations in transfer learning fail to fully elucidate the dynamics within the MTL domain Standley et al. (2020); Fifty et al. (2021).

**Multi-Task Architectures.** Multi-task architectures can be classified based on how much the parameters or features are shared across tasks in the network. The most commonly used structure is a shared trunk which consists of a common encoder shared by multiple tasks and a dedicated decoder for each task Dai et al. (2016); Ma et al. (2018); Simonyan & Zisserman (2014); Zhang et al. (2014). A tree-like architecture, featuring multiple division points for each task group, offers a more generalized structure Lu et al. (2017); Vandenhende et al. (2019); Bruggemann et al. (2020); Guo et al. (2020). The cross-talk architecture employs separate symmetrical networks for each task, facilitating feature exchange between layers at the same depth for information sharing between tasks Gao et al. (2019); Xu et al. (2018). The prediction distillation model Eigen & Fergus (2015); Xu et al. (2018); Vandenhende et al. (2020); Zhang et al. (2019) incorporates cross-task interactions at the end of the shared encoder, while the task switching network Sun et al. (2021); Sinha et al. (2018); Fernando et al. (2017); Maninis et al. (2019) adjusts network parameters depending on the task.

**MTL in Vision Transformers.** Recent advancements in multi-task architecture have explored the integration of Vision Transformer Dosovitskiy et al. (2020); Liu et al. (2021c); Wang et al. (2021a); Yang et al. (2021); Xie et al. (2021); Wang et al. (2021b) into MTL. MTFormer Xu et al. (2022) adopts a shared transformer encoder and decoder with a cross-task attention mechanism. MulT Bhattacharjee et al. (2022) leverages a shared attention mechanism to capture task dependencies, inspired by the Swin transformer. InvPT Ye & Xu (2022a) emphasizes global spatial position and multi-task context for dense prediction tasks through multi-scale feature aggregation. The Mixture of Experts (MoE) divides the model into predefined expert groups, dynamically shared or dedicated to specific tasks during the learning phase Riquelme et al. (2021); Zhang et al. (2022); Fan et al. (2022); Mustafa et al. (2022); Chen et al. (2023); Ye & Xu (2023). Task prompter Xu et al. (2023a;b); Ye & Xu (2022c) employs task-specific tokens to encapsulate task-specific information and utilizes cross-task interactions to enhance multi-task performance.

**Multi-Task Domain Generalization.** Task grouping based on their relations has also been explored in the field of domain adaptation. In particular, (Wei et al., 2024) proposes grouping heterogeneous tasks to regularize them, thereby promoting the learning of more generalized features across domain shifts. (Smith et al., 2021) explores generalization strategies at the mini-batch level. (Li et al., 2020) addresses diverse domain shift scenarios by incorporating all possible sequential domain learning paths to generalize features for unseen domains. (Shi et al., 2021) focuses on generalization to unseen domains by reducing dependence on specific domains through inter-domain gradient matching. Additionally, (Hu et al., 2022) analyzes the problem of spurious correlations in MTL and proposes regularization methods to mitigate this issue. The effect of gradient conflicts, which are considered the primary cause of negative transfer between tasks, is thoroughly examined in (Jiang et al., 2024). This work also proposes combining task distributions to identify better network parameters from a generalization perspective. The objectives of conventional multi-task optimization and domain generalization differ fundamentally. Conventional multi-task optimization typically assumes that the source and target domains share similar data distributions. In contrast, domain generalization focuses on scenarios involving significant domain shifts. This distinction leads to different approaches in

leveraging task relations to achieve their respective goals. In multi-task optimization, simultaneously updating heterogeneous tasks with conflicting gradients results in suboptimal optimization. On the other hand, domain generalization leverages task sets as a tool to extract generalized features applicable to various unseen domains. Overfitting to similar tasks can harm performance on unseen domains, making it advantageous to use heterogeneous tasks as a form of regularization.

## C  EXPERIMENTAL DETAILS

We implement our experiments on top of publically available code from Ye & Xu (2022b). We run our experiments on A6000 GPUs.

**Datasets.** We assess our method on multi-task datasets: NYUD-v2 Silberman et al. (2012), PASCAL-Context Mottaghi et al. (2014), and Taskonomy Zamir et al. (2018). These datasets encompass various vision tasks. NYUD-v2 comprises 4 vision tasks: depth estimation, semantic segmentation, surface normal prediction, and edge detection. Meanwhile, PASCAL-Context includes 5 tasks: semantic segmentation, human parts estimation, saliency estimation, surface normal prediction, and edge detection. In Taskonomy, we use 11 vision tasks: Depth Euclidean (DE), Depth Zbuffer (DZ), Edge Texture (ET), Keypoints 2D (K2), Keypoints 3D (K3), Normal (N), Principal Curvature (C), Reshading (R), Segment Unsup2d (S2), and Segment Unsup2.5D (S2.5).

**Metrics.**  To assess task performance, we employed widely used metrics across different tasks. For semantic segmentation, we utilized mean Intersection over Union (mIoU). The performance of surface normal prediction was gauged by computing the mean angle distances between the predicted output and ground truth. Depth estimation task performance was evaluated using Root Mean Squared Error (RMSE). For saliency estimation and human part segmentation, we utilized mean Intersection over Union (mIoU). Edge detection performance was assessed using optimal-dataset-scale-F-measure (odsF). For the Taskonomy benchmark, curvature was evaluated using RMSE, while the other tasks were evaluated using L1 distance, following the settings in Chen et al. (2023).

To evaluate multi-task performance, we adopted the metric proposed in Maninis et al. (2019). This metric measures per-task performance by averaging it with respect to the single-task baseline $b$, as shown in the equation: $\triangle_m = (1/T) \sum_{i=1}^{T} (-1)^{l_i} (M_{m,i} - M_{b,i})/M_{b,i}$ where $l_i = 1$ if a lower value of measure $M_i$ means better performance for task $i$, and 0 otherwise.

Table C.1: Hyperparameters for experiments.

| Hyperparameter | Value |
|---|---|
| Optimizer | Adam Kingma & Ba (2014) |
| Scheduler | Polynomial Decay |
| Minibatch size | 8 |
| Number of iterations | 40000 |
| Backbone (Transformer) | ViT Dosovitskiy et al. (2020) |
| ∟ Learning rate | 0.00002 |
| ∟ Weight Decay | 0.000001 |
| ∟ Affinity decay factor $\beta$ | 0.001 |

**Implementation Details.**  For experiments, we adopt ViT Dosovitskiy et al. (2020) pre-trained on ImageNet-22K Deng et al. (2009) as the multi-task encoder. Task-specific decoders merge the multi-scale features extracted by the encoder to generate the outputs for each task. The models are trained for 40,000 iterations on both NYUD Silberman et al. (2012) and PASCAL Everingham & Winn (2012) datasets with batch size 8. We used Adam optimizer with learning rate $2 \times 10^{-5}$ and $1 \times 10^{-6}$ of a weight decay with a polynomial learning rate schedule. The cross-entropy loss was used for semantic segmentation, human parts estimation, and saliency, edge detection. Surface normal prediction and depth estimation used L1 loss. The tasks are weighted equally to ensure a fair comparison. For the Taskonomy Benchmark Zamir et al. (2018), we use the dataloader from the open-access code provided by Chen et al. (2023), while maintaining experimental settings identical to those used for NYUD-v2 and PASCAL-Context. We use the same experimental setup for the other hyperparameters as in previous works Ye & Xu (2022a;c), as detailed in Table C.1.

# D    ADDITIONAL EXPERIMENTAL RESULTS

**Comparison of optimization results with different backbone sizes.** We compare the results of multi-task optimization on Taskonomy across various sizes of vision transformers, as shown in Tables D.1 to D.3. Our method consistently achieves superior performance across all backbone sizes. Unlike previous approaches that focus on learning shared parameters, our optimization strategy enhances the learning of task-specific parameters. This leads to significant performance improvements, especially with smaller backbones, where competition between tasks is more intense due to the limited number of shared parameters. How tasks are grouped, as visualized in fig. D.1, depends on the backbone size.

**Visualization of Proximal Inter-Task Affinity.** In Figure D.2, we present the tracked proximal inter-task affinity for each pair of tasks in Taskonomy. The changes in proximal inter-task affinity depend on the nature of the task pair, but as the backbone size increases, the affinity tends to become more positive. This trend is more noticeable in NYUD-v2 and PASCAL-Context, where there are fewer tasks, as shown in Figures D.3 and D.4. This pattern also aligns with the number of clustered tasks in Figure 3c, where the number of groups increases as the backbone size decreases.

**Effect of the Decay Rate with Visualization.** In Figure D.5, we visualize the proximal inter-task affinity tracked during optimization with various decay rates $\beta$, ranging from 0.01 to 1e-5 on a logarithmic scale. The decay rate $\beta$ helps stabilize the tracking of proximal inter-task affinity as it fluctuates during optimization. Additionally, it aids in understanding inter-task relations over time, independent of input data. For vision transformers, a decay rate of $\beta = 0.001$ demonstrates stable tracking. In real-world applications, multi-task performance is not highly sensitive to the decay rate $\beta$. In Table D.4, we evaluate how $\beta$ impacts multi-task performance on the Taskonomy benchmark. The results demonstrate that the proposed optimization method consistently improves performance across various $\beta$ values, minimizing the need for extensive hyperparameter tuning in practical scenarios.

**The Influence of Task Grouping Strategy.** In Table D.5, we present results comparing different task grouping strategies. These include randomly grouping tasks with a predefined number, grouping heterogeneous tasks, and grouping homogeneous tasks (our approach). The results clearly demonstrate that grouping homogeneous task sets yields superior performance under the proposed settings. This contrasts with the multi-task domain generalization approach, which groups heterogeneous tasks as a form of regularization to enhance generalization to unseen domains. This difference arises from the fundamentally distinct objectives of conventional multi-task optimization and domain generalization. Conventional multi-task optimization typically assumes that the source and target domains share similar data distributions, while domain generalization addresses scenarios with significant domain shifts. Consequently, the approaches to leveraging task relations differ to meet these distinct goals. As demonstrated in Theorems 1 and 2 of our work, in multi-task optimization, simultaneously updating heterogeneous tasks with low task affinity leads to suboptimal optimization and higher losses compared to updating similar task sets with high task affinity. This observation aligns with findings from previous multi-task optimization studies referenced in the related works section.

**Influence of Batch Size.** In Table D.6, we compare our method with single-gradient descent (GD) to evaluate its robustness in improving multi-task performance across varying batch sizes. The proposed optimization method consistently demonstrates performance improvements ($\triangle_m$ (% $\uparrow$)) of 5.27%, 5.71%, and 6.13% across different batch sizes. These results highlight the robustness and adaptability of the proposed algorithm across diverse scenarios.

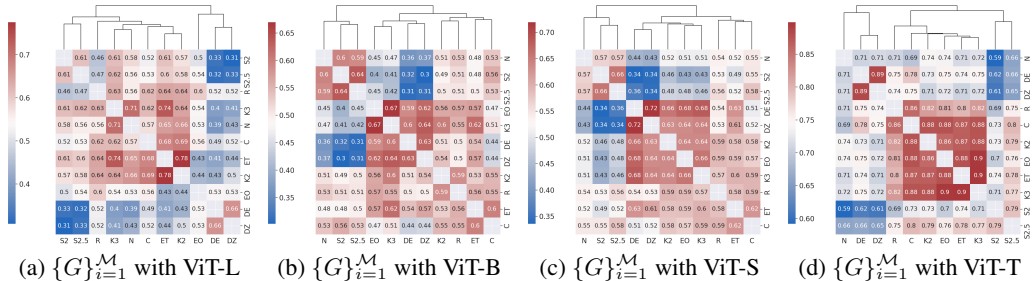

(a) $\{G\}_{i=1}^{\mathcal{M}}$ with ViT-L    (b) $\{G\}_{i=1}^{\mathcal{M}}$ with ViT-B    (c) $\{G\}_{i=1}^{\mathcal{M}}$ with ViT-S    (d) $\{G\}_{i=1}^{\mathcal{M}}$ with ViT-T

Figure D.1: The averaged grouping results $\{G\}_{i=1}^{\mathcal{M}}$ on the Taskonomy benchmark are shown for (a) ViT-L, (b) ViT-B, (c) ViT-S, and (d) ViT-T, respectively.

Table D.1: Comparison with previous multi-task optimization approaches on Taskonomy with ViT-B.

| Task | DE | DZ | EO | ET | K2 | K3 | N | C | R | S2 | S2.5 | $\triangle_m$ (↑) |
| | L1 Dist. ↓ | L1 Dist. ↓ | L1 Dist. ↓ | L1 Dist. ↓ | L1 Dist. ↓ | L1 Dist. ↓ | L1 Dist. ↓ | RMSE↓ | L1 Dist. ↓ | L1 Dist. ↓ | L1 Dist. ↓ | |
|---|---|---|---|---|---|---|---|---|---|---|---|---|
| Single Task | 0.0183 | 0.0186 | 0.1089 | 0.1713 | 0.1630 | 0.0863 | 0.2953 | 0.7522 | 0.1504 | 0.1738 | 0.1530 | - |
| GD | 0.0188 | 0.0197 | 0.1283 | 0.1745 | 0.1718 | 0.0933 | 0.2599 | 0.7911 | 0.1799 | 0.1885 | 0.1631 | -6.35 |
| GradDrop | 0.0195 | 0.0206 | 0.1318 | 0.1748 | 0.1735 | 0.0945 | 0.3018 | 0.8060 | 0.1866 | 0.1920 | 0.1607 | -9.54 |
| MGDA | - | - | - | - | - | - | - | - | - | - | - | - |
| UW | 0.0188 | 0.0198 | 0.1285 | 0.1745 | 0.1719 | 0.0933 | 0.2535 | 0.7915 | 0.1800 | 0.1883 | 0.1629 | -6.19 |
| DTP | 0.0187 | 0.0198 | 0.1283 | 0.1745 | 0.1720 | 0.0933 | 0.2558 | 0.7912 | 0.1804 | 0.1884 | 0.1634 | -6.25 |
| DWA | 0.0188 | 0.0197 | 0.1287 | 0.1745 | 0.1719 | 0.0933 | 0.2570 | 0.7927 | 0.1806 | 0.1887 | 0.1632 | -6.33 |
| PCGrad | 0.0185 | 0.0188 | 0.1285 | 0.1738 | 0.1703 | 0.0928 | 0.2557 | 0.7964 | 0.1810 | 0.1882 | 0.1569 | -5.22 |
| CAGrad | 0.0192 | 0.0196 | 0.1306 | 0.1733 | 0.1654 | 0.0939 | 0.2871 | 0.8147 | 0.1901 | 0.1906 | 0.1659 | -8.34 |
| IMTL | 0.0189 | 0.0200 | 0.1287 | 0.1745 | 0.1720 | 0.0934 | 0.2618 | 0.7928 | 0.1811 | 0.1888 | 0.1635 | -6.75 |
| Aligned-MTL | 0.0191 | 0.0202 | 0.1263 | 0.1729 | 0.1663 | 0.0944 | 0.3061 | 0.8560 | 0.1936 | 0.1872 | 0.1585 | -8.93 |
| Nash-MTL | 0.0175 | 0.0182 | 0.1208 | 0.1730 | 0.1663 | 0.0901 | 0.2686 | 0.7958 | 0.1707 | 0.1839 | 0.1577 | -2.79 |
| FAMO | 0.0189 | 0.0200 | 0.1285 | 0.1745 | 0.1720 | 0.0934 | 0.2715 | 0.7929 | 0.1807 | 0.1891 | 0.1640 | -7.21 |
| Ours | 0.0167 | 0.0169 | 0.1228 | 0.1739 | 0.1695 | 0.0910 | 0.2344 | 0.7600 | 0.1691 | 0.1836 | 0.1571 | -0.64 |

Table D.2: Comparison with previous multi-task optimization approaches on Taskonomy with ViT-S.

| Task | DE | DZ | EO | ET | K2 | K3 | N | C | R | S2 | S2.5 | $\triangle_m$ (↑) |
| | L1 Dist. ↓ | L1 Dist. ↓ | L1 Dist. ↓ | L1 Dist. ↓ | L1 Dist. ↓ | L1 Dist. ↓ | L1 Dist. ↓ | RMSE↓ | L1 Dist. ↓ | L1 Dist. ↓ | L1 Dist. ↓ | |
|---|---|---|---|---|---|---|---|---|---|---|---|---|
| Single Task | 0.0264 | 0.0259 | 0.1348 | 0.1740 | 0.1667 | 0.0973 | 0.3481 | 0.8598 | 0.1905 | 0.1857 | 0.1691 | - |
| GD | 0.0264 | 0.0272 | 0.1574 | 0.1775 | 0.1838 | 0.1038 | 0.4370 | 0.9237 | 0.2475 | 0.2076 | 0.1858 | -11.39 |
| GradDrop | 0.0274 | 0.0280 | 0.1609 | 0.1779 | 0.1856 | 0.1042 | 0.4472 | 0.9366 | 0.2549 | 0.2106 | 0.1821 | -13.14 |
| MGDA | - | - | - | - | - | - | - | - | - | - | - | - |
| UW | 0.0263 | 0.0269 | 0.1570 | 0.1775 | 0.1832 | 0.1037 | 0.4362 | 0.9202 | 0.2465 | 0.2075 | 0.1856 | -11.11 |
| DTP | 0.0262 | 0.0273 | 0.1568 | 0.1778 | 0.1831 | 0.1037 | 0.4884 | 0.9207 | 0.2466 | 0.2073 | 0.1849 | -12.52 |
| DWA | 0.0264 | 0.0271 | 0.1572 | 0.1776 | 0.1834 | 0.1038 | 0.4336 | 0.9215 | 0.2469 | 0.2075 | 0.1856 | -11.20 |
| PCGrad | 0.0271 | 0.0274 | 0.1570 | 0.1766 | 0.1784 | 0.1034 | 0.4522 | 0.9343 | 0.2525 | 0.2071 | 0.1811 | -11.78 |
| CAGrad | 0.0289 | 0.0282 | 0.1611 | 0.1769 | 0.1706 | 0.1062 | 0.4723 | 0.9557 | 0.2689 | 0.2122 | 0.1902 | -15.09 |
| IMTL-L | 0.0255 | 0.0258 | 0.1510 | 0.1744 | 0.1716 | 0.1005 | 0.4339 | 0.9459 | 0.2466 | 0.2036 | 0.1825 | -8.90 |
| Aligned-MTL | 0.0286 | 0.0290 | 0.1603 | 0.1744 | 0.1711 | 0.1033 | 0.4596 | 1.0022 | 0.2783 | 0.2090 | 0.1854 | -15.06 |
| Nash-MTL | 0.0255 | 0.0258 | 0.1510 | 0.1744 | 0.1716 | 0.1005 | 0.4339 | 0.9459 | 0.2466 | 0.2036 | 0.1825 | -8.79 |
| FAMO | 0.0263 | 0.0272 | 0.1573 | 0.1774 | 0.1835 | 0.1035 | 0.4326 | 0.9208 | 0.2464 | 0.2077 | 0.1858 | -11.23 |
| Ours | 0.0225 | 0.0229 | 0.1444 | 0.1762 | 0.1775 | 0.0995 | 0.3983 | 0.8620 | 0.2156 | 0.1997 | 0.1774 | -2.83 |

Table D.3: Comparison with previous multi-task optimization approaches on Taskonomy with ViT-T.

| Task | DE | DZ | EO | ET | K2 | K3 | N | C | R | S2 | S2.5 | $\triangle_m$ (↑) |
| | L1 Dist. ↓ | L1 Dist. ↓ | L1 Dist. ↓ | L1 Dist. ↓ | L1 Dist. ↓ | L1 Dist. ↓ | L1 Dist. ↓ | RMSE↓ | L1 Dist. ↓ | L1 Dist. ↓ | L1 Dist. ↓ | |
|---|---|---|---|---|---|---|---|---|---|---|---|---|
| Single Task | 0.0289 | 0.0290 | 0.1405 | 0.1774 | 0.1682 | 0.0970 | 0.3837 | 0.8968 | 0.2096 | 0.1904 | 0.1729 | - |
| GD | 0.0279 | 0.0285 | 0.1604 | 0.1789 | 0.1860 | 0.1043 | 0.4704 | 0.9488 | 0.2613 | 0.2086 | 0.1914 | -9.21 |
| GradDrop | 0.0287 | 0.0292 | 0.1630 | 0.1795 | 0.1868 | 0.1052 | 0.4795 | 0.9621 | 0.2697 | 0.2118 | 0.1878 | -10.68 |
| MGDA | - | - | - | - | - | - | - | - | - | - | - | - |
| UW | 0.0279 | 0.0285 | 0.1604 | 0.1789 | 0.1859 | 0.1043 | 0.4699 | 0.9488 | 0.2613 | 0.2085 | 0.1914 | -9.21 |
| DTP | 0.0278 | 0.0288 | 0.1603 | 0.1790 | 0.1859 | 0.1042 | 0.4697 | 0.9488 | 0.2614 | 0.2088 | 0.1915 | -9.27 |
| DWA | 0.0279 | 0.0285 | 0.1604 | 0.1789 | 0.1859 | 0.1043 | 0.4693 | 0.9489 | 0.2613 | 0.2086 | 0.1913 | -9.20 |
| PCGrad | 0.0283 | 0.0290 | 0.1604 | 0.1769 | 0.1803 | 0.1036 | 0.4720 | 0.9645 | 0.2683 | 0.2090 | 0.1866 | -9.28 |
| CAGrad | 0.0300 | 0.0304 | 0.1644 | 0.1743 | 0.1721 | 0.1055 | 0.4838 | 0.9818 | 0.2847 | 0.2143 | 0.1974 | -12.12 |
| IMTL | 0.0276 | 0.0282 | 0.1553 | 0.1754 | 0.1743 | 0.1018 | 0.4621 | 0.9809 | 0.2623 | 0.2051 | 0.1878 | -7.55 |
| Aligned-MTL | 0.0296 | 0.0318 | 0.1633 | 0.1765 | 0.1757 | 0.1150 | 0.4806 | 1.0270 | 0.2935 | 0.2109 | 0.1887 | -13.70 |
| Nash-MTL | 0.0276 | 0.0282 | 0.1553 | 0.1754 | 0.1743 | 0.1018 | 0.4621 | 0.9809 | 0.2623 | 0.2051 | 0.1878 | -7.46 |
| FAMO | 0.0279 | 0.0285 | 0.1604 | 0.1789 | 0.1859 | 0.1043 | 0.4718 | 0.9488 | 0.2612 | 0.2085 | 0.1913 | -9.33 |
| Ours | 0.0252 | 0.0257 | 0.1526 | 0.1774 | 0.1827 | 0.1019 | 0.4337 | 0.9100 | 0.2402 | 0.2026 | 0.1845 | -3.67 |

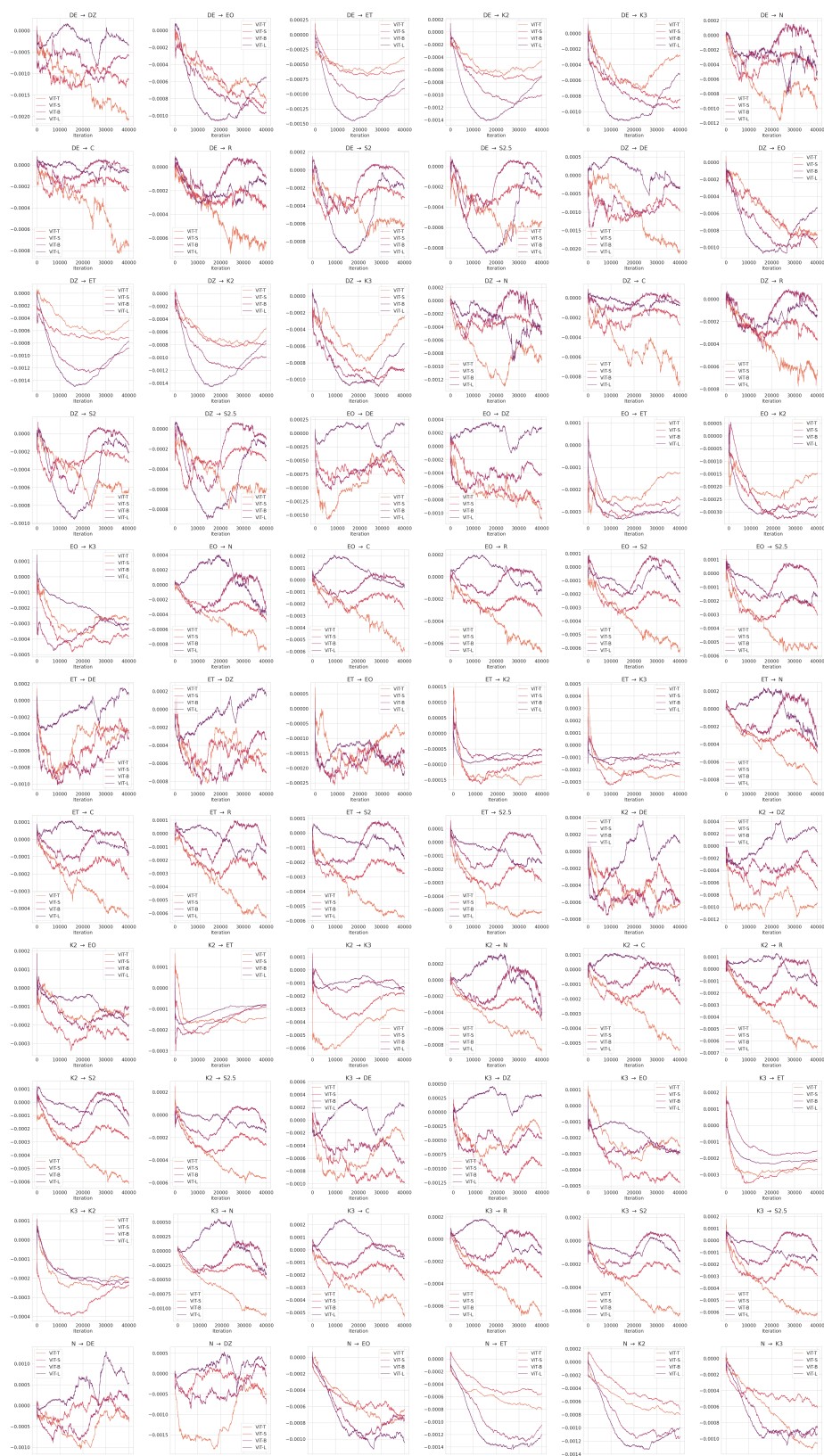

Figure D.2: Changes in proximal inter-task affinity during the optimization process of ViT-L with Taskonomy benchmark.

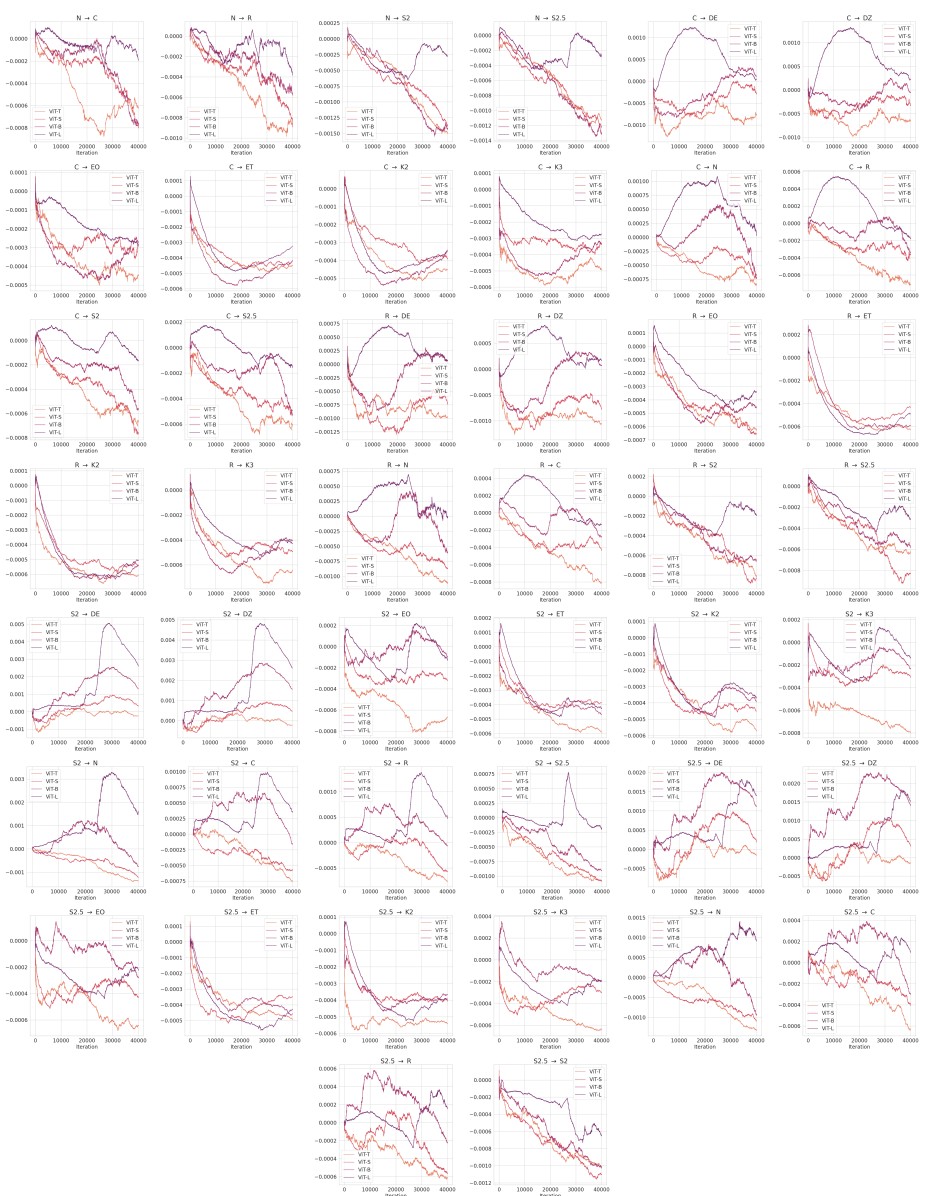

Figure D.2: Changes in proximal inter-task affinity during the optimization process with Taskonomy benchmark.

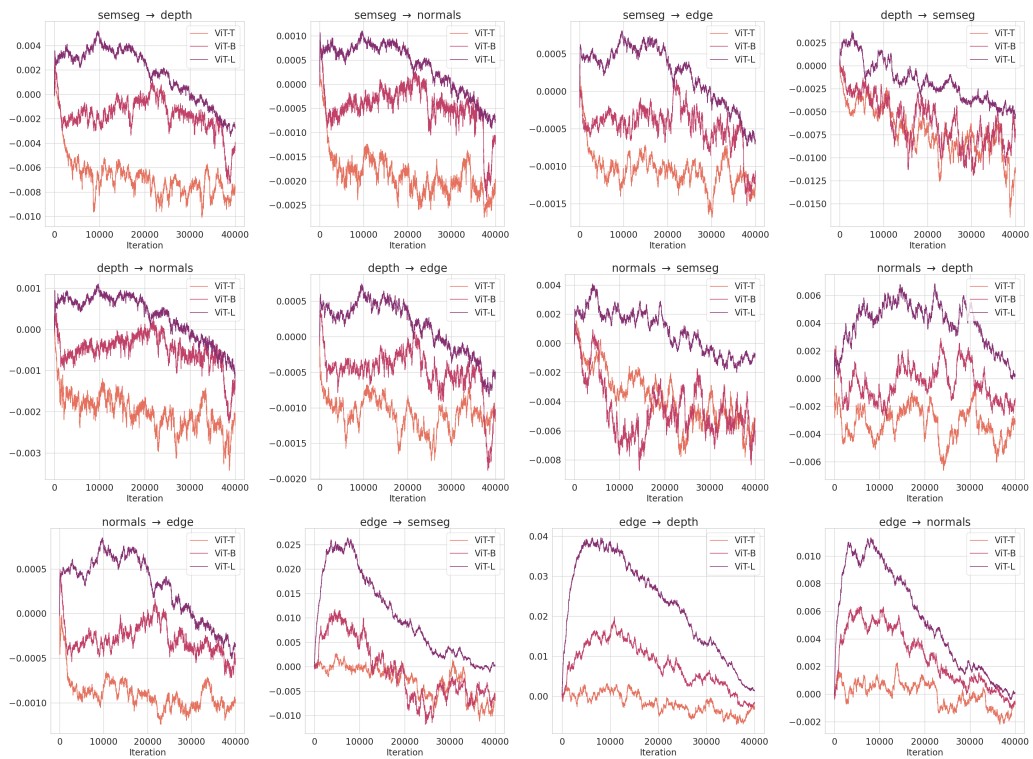

Figure D.3: Changes in the proximal inter-task affinity during the optimization process of different sizes of ViT with NYUD-v2.

Table D.4: Results on Taskonomy with different affinity decay rates $\beta$.

| Task | DE
L1 Dist. ↓ | DZ
L1 Dist. ↓ | EO
L1 Dist. ↓ | ET
L1 Dist. ↓ | K2
L1 Dist. ↓ | K3
L1 Dist. ↓ | N
L1 Dist. ↓ | C
RMSE ↓ | R
L1 Dist. ↓ | S2
L1 Dist. ↓ | S2.5
L1 Dist. ↓ | $\triangle_m$ (↑) |
|---|---|---|---|---|---|---|---|---|---|---|---|---|
| Single Task | 0.0183 | 0.0186 | 0.1089 | 0.1713 | 0.1630 | 0.0863 | 0.2953 | 0.7522 | 0.1504 | 0.1738 | 0.1530 | - |
| GD | 0.0188 | 0.0197 | 0.1283 | 0.1745 | 0.1718 | 0.0933 | 0.2599 | 0.7911 | 0.1799 | 0.1885 | 0.1631 | -6.35 |
| $\beta$=0.0001 | 0.0165 | 0.0168 | 0.1224 | 0.1739 | 0.1693 | 0.0907 | 0.2304 | 0.7581 | 0.1683 | 0.1831 | 0.1571 | -0.18 |
| $\beta$=0.001 | 0.0167 | 0.0169 | 0.1228 | 0.1739 | 0.1695 | 0.0910 | 0.2344 | 0.7600 | 0.1691 | 0.1836 | 0.1571 | -0.64 |
| $\beta$=0.01 | 0.0167 | 0.0171 | 0.1232 | 0.1739 | 0.1698 | 0.0912 | 0.2362 | 0.7623 | 0.1705 | 0.1834 | 0.1576 | -1.01 |
| $\beta$=0.1 | 0.0167 | 0.0171 | 0.1231 | 0.1739 | 0.1695 | 0.0912 | 0.2355 | 0.7631 | 0.1697 | 0.1831 | 0.1575 | -0.87 |

Table D.5: Comparison of different grouping strategies on the Taskonomy benchmark.

| Task | DE
L1 Dist. ↓ | DZ
L1 Dist. ↓ | EO
L1 Dist. ↓ | ET
L1 Dist. ↓ | K2
L1 Dist. ↓ | K3
L1 Dist. ↓ | N
L1 Dist. ↓ | C
RMSE ↓ | R
L1 Dist. ↓ | S2
L1 Dist. ↓ | S2.5
L1 Dist. ↓ | $\triangle_m$ (↑) |
|---|---|---|---|---|---|---|---|---|---|---|---|---|
| Heterogeneous | 0.0172 | 0.0176 | 0.1252 | 0.1741 | 0.1700 | 0.0920 | 0.2475 | 0.7781 | 0.1743 | 0.1849 | 0.1660 | -3.10 |
| Random ($N(\mathcal{M})$=2) | 0.0177 | 0.0180 | 0.1259 | 0.1741 | 0.1707 | 0.0923 | 0.2662 | 0.7807 | 0.1757 | 0.1871 | 0.1617 | -4.24 |
| Random ($N(\mathcal{M})$=3) | 0.0172 | 0.0177 | 0.1250 | 0.1741 | 0.1703 | 0.0920 | 0.2619 | 0.7754 | 0.1749 | 0.1866 | 0.1607 | -3.35 |
| Random ($N(\mathcal{M})$=4) | 0.0183 | 0.0187 | 0.1277 | 0.1746 | 0.1706 | 0.0936 | 0.2812 | 0.7841 | 0.1804 | 0.1882 | 0.1636 | -6.12 |
| Random ($N(\mathcal{M})$=5) | 0.0186 | 0.0184 | 0.1274 | 0.1747 | 0.1708 | 0.0935 | 0.3150 | 0.7842 | 0.1800 | 0.1888 | 0.1640 | -7.17 |
| Random ($N(\mathcal{M})$=6) | 0.0208 | 0.0209 | 0.1349 | 0.1750 | 0.1721 | 0.0961 | 0.3334 | 0.8222 | 0.1976 | 0.1935 | 0.1703 | -13.20 |
| Ours | 0.0167 | 0.0169 | 0.1228 | 0.1739 | 0.1695 | 0.0910 | 0.2344 | 0.7600 | 0.1691 | 0.1836 | 0.1571 | -0.64 |

# E ALGORITHM COMPLEXITY AND COMPUTATIONAL LOAD

We also provide a detailed time comparison of previous multi-task optimization methods on Taskonomy. As shown in Table E.1, our approach effectively optimizes multiple tasks with more efficient training times. Our method converges faster than gradient-based approaches, as the primary bottleneck in optimization lies in backpropagation and gradient manipulation.

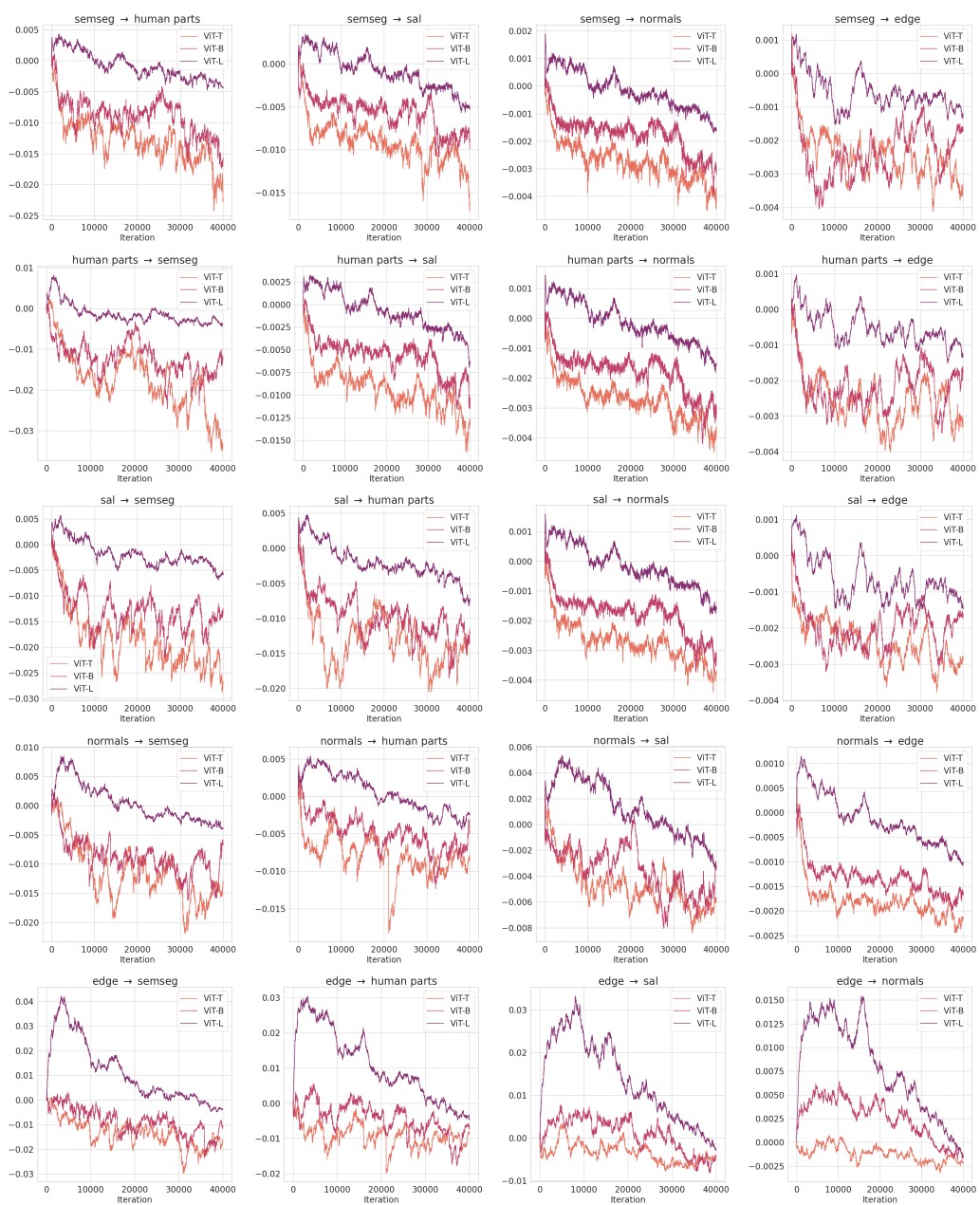

Figure D.4: Changes in proximal inter-task affinity during the optimization process of different sizes of ViT with PASCAL-Context.

Table D.6: Results on Taskonomy with varying batch sizes using ViT-B (batch sizes in brackets).

| Task | DE | DZ | EO | ET | K2 | K3 | N | C | R | S2 | S2.5 | |
|---|---|---|---|---|---|---|---|---|---|---|---|---|
| | L1 Dist. ↓ | L1 Dist. ↓ | L1 Dist. ↓ | L1 Dist. ↓ | L1 Dist. ↓ | L1 Dist. ↓ | L1 Dist. ↓ | RMSE ↓ | L1 Dist. ↓ | L1 Dist. ↓ | L1 Dist. ↓ | $\triangle_m$ (↑) |
| Single Task | 0.0183 | 0.0186 | 0.1089 | 0.1713 | 0.1630 | 0.0863 | 0.2953 | 0.7522 | 0.1504 | 0.1738 | 0.1530 | - |
| GD(4) | 0.0208 | 0.0214 | 0.1323 | 0.1747 | 0.1723 | 0.0952 | 0.2768 | 0.8214 | 0.1936 | 0.1921 | 0.1677 | -10.88 |
| Ours(4) | 0.0185 | 0.0190 | 0.1273 | 0.1741 | 0.1709 | 0.0928 | 0.2739 | 0.7957 | 0.1809 | 0.1888 | 0.1632 | -6.19 |
| GD(8) | 0.0188 | 0.0197 | 0.1283 | 0.1745 | 0.1718 | 0.0933 | 0.2599 | 0.7911 | 0.1799 | 0.1885 | 0.1631 | -6.35 |
| Ours(8) | 0.0167 | 0.0169 | 0.1228 | 0.1739 | 0.1695 | 0.0910 | 0.2344 | 0.7600 | 0.1691 | 0.1836 | 0.1571 | -0.64 |
| GD(16) | 0.0172 | 0.0180 | 0.1248 | 0.1742 | 0.1711 | 0.0920 | 0.2280 | 0.7641 | 0.1706 | 0.1848 | 0.1589 | -1.94 |
| Ours(16) | 0.0153 | 0.0154 | 0.1186 | 0.1737 | 0.1682 | 0.0893 | 0.1967 | 0.7334 | 0.1581 | 0.1780 | 0.1516 | +4.19 |

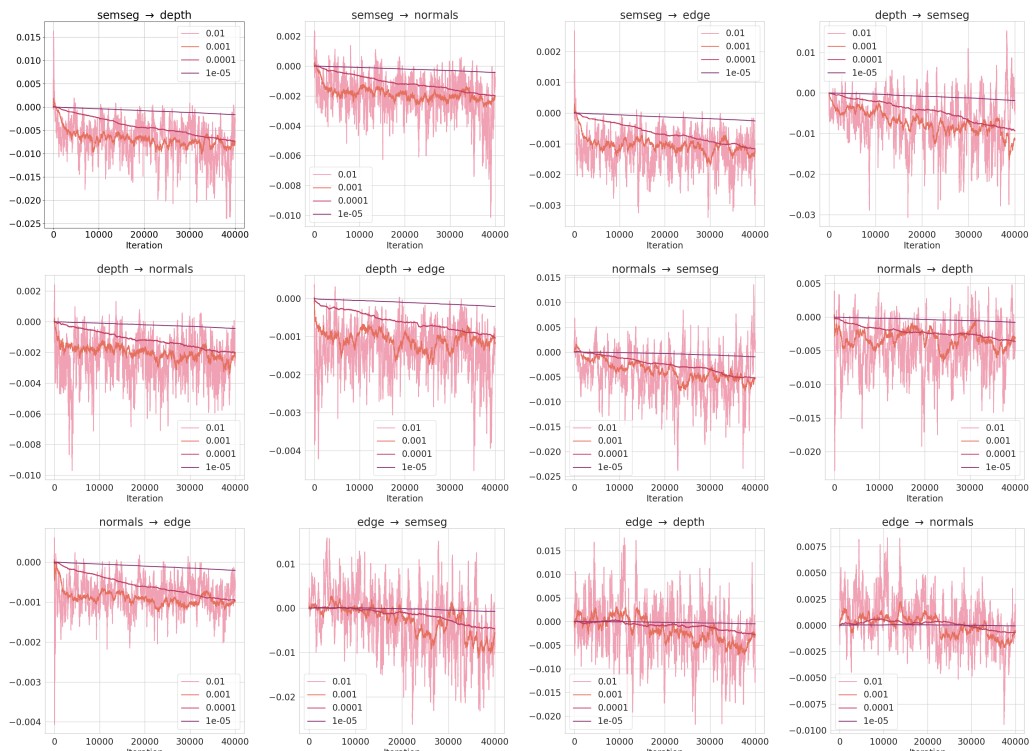

Figure D.5: Changes in proximal inter-task affinity during the optimization process with different decay rates, $\beta$.

Table E.1: Comparison of the average time required by each optimization process to handle a single batch for 11 tasks on Taskonomy.

| Process (sec) | Forward Pass | Backpropagation | Gradient Manipulation | Optimizer Step | Clustering + Affinity Update | Total |
|---|---|---|---|---|---|---|
| GD | 0.030(9.04%) | 0.198(59.26%) | - | 0.106(31.69%) | - | 0.33 |
| UW | 0.030(8.82%) | 0.198(58.24%) | - | 0.112(32.94%) | - | 0.34 |
| DTP | 0.030(8.70%) | 0.199(57.68%) | - | 0.116(33.62%) | - | 0.34 |
| DWA | 0.031(9.01%) | 0.198(57.56%) | - | 0.115(33.43%) | - | 0.34 |
| GradDrop | 0.030(1.13%) | 2.05(80.54%) | 0.411(16.02%) | 0.059(2.30%) | - | 2.57 |
| MGDA | 0.033(0.086%) | 2.06(5.36%) | 36.29(94.47%) | 0.031(0.081%) | - | 38.42 |
| PCGrad | 0.030(0.63%) | 2.07(44.09%) | 2.57(54.62%) | 0.031(0.66%) | - | 4.70 |
| CAGrad | 0.030(0.57%) | 2.06(39.39%) | 3.11(59.44%) | 0.031(0.59%) | - | 5.23 |
| Aligned-MTL | 0.027(0.86%) | 2.07(64.99%) | 1.06(33.20%) | 0.030(0.95%) | - | 3.19 |
| FAMO | 0.030 (8.72%) | 0.198(57.56%) | - | 0.116(33.72%) | - | 0.34 |
| Ours | 0.072 (7.13%) | 0.576(56.72%) | - | 0.323(31.82%) | 0.044(4.33%) | 1.02 |

