# OpenReview forum: "Selective Task Group Updates for Multi-Task Optimization"
_ICLR.cc/2025/Conference — ICLR 2025 Poster_

### Official Review · Reviewer_HAqj · 2024-10-30

**Soundness:** 3
**Presentation:** 2
**Contribution:** 3
**Rating:** 6
**Confidence:** 3

**Summary:**

This paper proposes a novel MTL method in which tasks are dynamically grouped according to the conflict of different tasks during the optimization process and the model is optimized based on these grouped tasks. The method introduces only a limited computation cost and the experiment results show strong performance of their method compared to several existing MTL methods

**Strengths:**

1. The paper proposes a novel optimization method to optimize the multi-task learning process from a new perspective.

2. The method demonstrates promising results on the NYUv2, PASCAL-Context, and Taskonomy datasets.

3. The paper provides a theoretical explanation of the advantages of sequential optimization of task groups and provides an analysis of convergence.

**Weaknesses:**

1. There are some incorrect statements in the article. For example, in Line 092, “This perspective is not addressed in traditional multi-task optimization, which typically focuses solely on the learning of shared parameters.” is wrong, because the learning of task-specific parameters is considered in IMTL [1].

2. The conclusion in Line 373-374, “This suggests that grouping tasks with proximal inter-task affinity and subsequently updating these groups sequentially result in lower multi-task optimization. sequentially result in lower multi-task loss compared to jointly backpropagating all tasks.” does not seem relevant to the theorem above. Can you give a more detailed explanation?

3. For the conclusion in Line 525-526, “We observe that the affinity decay rate ... within a reasonable range.”, there is a lack of experimental results on the performance of models with different $\beta$.

[1] Liu, L., Li, Y., Kuang, Z., Xue, J.-H., Chen, Y., Yang, W., Liao, Q., & Zhang, W. Towards Impartial Multi-task Learning. ICLR, 2021.

**Questions:**

1. Why is the number of groups in Figure 3c not an integer?

2. Line 523, "Table 4c" means "Figure 4c"?

3. Can you compare the performance of random grouping during optimization?

---

> ### Author Response · Authors · 2024-11-24
> **Rebuttal**
>
> Thank you for your time and effort in reviewing our paper. Your thorough feedback was very helpful in refining our work. Specifically, we noticed some typos and statements that could cause misunderstandings for readers. We have revised these thoroughly, incorporating your feedback. We welcome any further feedback or discussion.
>
> ### **Answer to W1**
> Thank you for pointing out the error in our statement. IMTL handles shared and task-specific parameters separately, so our original statement in line 92 was incorrect. What we intended to convey is that, in our approach, the updates to shared and task-specific parameters occur concurrently, with each influencing the other. Unlike previous optimization methods that treat these parameters independently, our approach integrates their interdependence through proximal inter-task affinity for improved multi-task optimization. We have revised the statement to: "This perspective is not addressed in traditional multi-task optimization, which typically deals with the learning of shared and task-specific parameters independently," in the revised version.
>
> ### **Answer to W2**
> The statements explain the main result of Theorem 5, which is the inequality $B_{i,j,k \rightarrow k}^{t+(m-1)/M} \leq B_{j; i,k \rightarrow k}^{t+(m-1)/M}$. Specifically, $B_{i,j,k \rightarrow k}^{t+(m-1)/M}$ represents the proximal inter-task affinity when updating all tasks $\{i,j,k\}$ together, while $B_{j; i,k \rightarrow k}^{t+(m-1)/M}$ represents the affinity when updating tasks $\{j\}$ and $\{i,k\}$ sequentially, with $\{i,k\}$ having positive inter-task affinity. Considering the definition of proximal inter-task affinity, a higher proximal inter-task affinity with respect to task $k$ means lower loss for task $k$. Therefore, when this reasoning is extended across all tasks, grouping tasks with positive proximal inter-task affinity and updating them sequentially leads to lower multi-task loss.
>
>
> ### **Answer to W3**
> We examine the impact of the affinity decay rate $\beta$ in the table below. The results demonstrate that our approach consistently improves performance across a wide range of $\beta$ values, reducing the need for extensive hyperparameter tuning in practical applications. We have revised the statement in the updated version and included the experimental results in the supplementary material.
>
> - Results on Taskonomy with different affinity decay rates $\beta$.
>
> |Task|DE|DZ|EO|ET|K2|K3|N|C|R|S2|S2.5|$\triangle_m$ (% $\uparrow$)|
> |:-|:-:|:-:|:-:|:-:|:-:|:-:|:-:|:-:|:-:|:-:|:-:|:-:|
> |Single Task|0.0183|0.0186|0.1089|0.1713|0.1630|0.0863|0.2953|0.7522|0.1504|0.1738|0.1530|-
> |GD|0.0188|0.0197|0.1283|0.1745|0.1718|0.0933|0.2599|0.7911|0.1799|0.1885|0.1631|-6.35
> |$\beta$=0.0001|0.0165|0.0168|0.1224&|0.1739|0.1693|0.0907|0.2304|0.7581|0.1683|0.1831|0.1571|-0.18
> |$\beta$=0.001|0.0167|0.0169|0.1228|0.1739|0.1695|0.0910|0.2344|0.7600|0.1691|0.1836|0.1571|-0.64
> |$\beta$=0.01|0.0167|0.0171|0.1232|0.1739|0.1698|0.0912|0.2362|0.7623|0.1705|0.1834|0.1576|-1.01
> |$\beta$=0.1|0.0167|0.0171|0.1231|0.1739|0.1695|0.0912|0.2355|0.7631|0.1697|0.1831|0.1575|-0.87
>
>
> ### **Answer to Q1**
> The proposed optimization method divides the task set into multiple groups for each batch. Over the course of 40,000 training iterations, we partition the iterations into approximately 50 intervals and calculate the average number of task groups within each interval for clarity in visualization. This averaging process explains why the number of groups shown in Figure 3c is not an integer.
>
> ### **Answer to Q2**
> Thank you for pointing out the error in our paper. We have revised "Table 4c" in Line 523 to "Figure 4c."
>
> ### **Answer to Q3**
> We present an ablation study on various task grouping strategies and their impact on performance. Specifically, these strategies include randomly grouping tasks with a predefined number ($N(\mathcal{M})$), grouping heterogeneous tasks, and grouping homogeneous tasks (our approach). The results clearly show that our optimization strategy achieves better performance.
>
> - Comparison of Various Grouping Strategies on Taskonomy.
>
> |Task|DE|DZ|EO|ET|K2|K3|N|C|R|S2|S2.5|$\triangle_m$ (% $\uparrow$)|
> |:-|:-:|:-:|:-:|:-:|:-:|:-:|:-:|:-:|:-:|:-:|:-:|:-:|
> |Heterogeneous|0.0172|0.0176|0.1252|0.1741|0.1700|0.0920|0.2475|0.7781|0.1743|0.1849|0.1660|-3.10
> |Random ($N(\mathcal{M})$=2)|0.0177|0.0180|0.1259|0.1741|0.1707|0.0923|0.2662|0.7807|0.1757|0.1871|0.1617|-4.24
> |Random ($N(\mathcal{M})$=3)|0.0172|0.0177|0.1250|0.1741|0.1703|0.0920|0.2619|0.7754|0.1749|0.1866|0.1607|-3.35
> |Random ($N(\mathcal{M})$=4)|0.0183|0.0187|0.1277|0.1746|0.1706|0.0936|0.2812|0.7841|0.1804|0.1882|0.1636|-6.12
> |Random ($N(\mathcal{M})$=5)|0.0186|0.0184|0.1274|0.1747|0.1708|0.0935|0.3150|0.7842|0.1800|0.1888|0.1640|-7.17
> |Random ($N(\mathcal{M})$=6)|0.0208|0.0209|0.1349|0.1750|0.1721|0.0961|0.3334|0.8222|0.1976|0.1935|0.1703|-13.20
> |Ours|0.0167|0.0169|0.1228|0.1739|0.1695|0.0910|0.2344|0.7600|0.1691|0.1836|0.1571|-0.64

---

> > ### Comment · Reviewer_HAqj · 2024-11-26
> >
> > Thank you for your response, which addressed most of my questions. However, I still have some concerns regarding your reply to Q3.
> >
> > 1. I would like to know how you define "heterogeneous" here. Does it refer to two tasks with significant conflicts?
> >
> > 2. From my understanding, heterogeneous should serve as a lower bound for MTL, but the results are the same when randomly divided into two groups.
> >
> > 3. It seems that the more groups there are, the worse the performance becomes in the experimental results. Could you analyze and explain this phenomenon? I am very interested in the reasons behind it.
> >
> > Thank you again for your response. If you can address these concerns, I would be happy to improve my score.

---

> ### Author Response · Authors · 2024-11-27
> **Response to the Comments**
>
> We deeply appreciate your response and once again thank you for your efforts in reviewing our work. Your insightful questions are highly valued, as the grouping strategy and its analysis are crucial aspects of this study. Below are our detailed responses, and we welcome any further questions or discussions.
>
> ---
>
> ### **1. I would like to know how you define "heterogeneous" here. Does it refer to two tasks with significant conflicts?**
>
> Defining "heterogeneous" tasks is indeed critical for explaining the aforementioned experiments, and we appreciate your observation. As you pointed out, we should clarify how the "heterogeneous" setting is implemented.
>
> The similarity between tasks depends on the nature of their objectives, but it can be measured using a metric to further enhance learning algorithms. In our work, we use *proximal inter-task affinity* as this metric. By tracking proximal inter-task affinity during optimization, we can represent the similarity or disparity between tasks in a matrix format. Taking the negative value of the tracked affinity between two tasks represents their disparity instead of similarity.
>
> Using this measure, we cluster tasks based on disparity to create "heterogeneous" task groups. Thus, the "heterogeneous" setting refers to the proposed *selective task group updates algorithm* that clusters tasks with negative proximal inter-task affinity.
>
> ---
>
> ### **2. From my understanding, heterogeneous should serve as a lower bound for MTL, but the results are the same when randomly divided into two groups.**
>
>
> We mistakenly reported the same results for the "heterogeneous" setting and "Random ($N(\mathcal{M}) = 2$)." We sincerely apologize for any confusion caused by this error. The corrected results have now been updated in the respective table.
>
> However, regarding your observation that "heterogeneous should serve as a lower bound," our experimental results reveal a different trend. In this setting, the "heterogeneous" strategy demonstrates better performance compared to random selection criteria. A more detailed explanation of this phenomenon is provided in our response to Question 3.
>
> ---
>
> ### **3. It seems that the more groups there are, the worse the performance becomes in the experimental results. Could you analyze and explain this phenomenon? I am very interested in the reasons behind it.**
>
> Thank you for this insightful question, as it highlights a non-trivial aspect of our method. Before addressing this, let us clarify that these experiments are conducted in *dynamic settings*, where task grouping changes across batches during optimization.
>
> In the "heterogeneous" setting, we track proximal inter-task affinity dynamically during optimization and group heterogeneous tasks for each batch. (The affinity fluctuates continuously during training.) In contrast, in the random grouping strategy, tasks are grouped randomly for each batch.
>
> To explain the observed phenomenon, we consider two key factors:
> 1. **Randomness in the clustering process**
> 2. **Cluster characteristics (e.g., number of clusters and task grouping criteria)**
>
> First, when comparing random grouping scenarios to our method, it becomes clear that random grouping during optimization negatively affects multi-task performance. This is due to the instability introduced by the randomness in forming task groups for each batch. As the number of task groups ($N(\mathcal{M})$) increases, the randomness intensifies because there are more possible combinations of groups. However, when the number of groups equals the number of tasks (the "Separate" case in Table 5 of the main paper, showing -2.48 (% $\uparrow$) in $\triangle_m$), the randomness is eliminated because each task is assigned to its own group, leaving no room for variability in the grouping process. As a result, the performance improves compared to randomly clustered task sets. Similarly, in the "heterogeneous" setting, although the clustering targets heterogeneous tasks, the randomness in grouping is reduced because tasks are grouped based on their tracked affinity during optimization. This explains why the "heterogeneous" setting outperforms random grouping strategies.
>
> Second, we observe that grouping homogeneous tasks (tasks with high proximal inter-task affinity), as adopted in our method, yields better performance than grouping heterogeneous tasks (tasks with low proximal inter-task affinity). This indicates that grouping homogeneous tasks, even though their affinities fluctuate during optimization, consistently outperforms alternative grouping strategies.
>
> ---
>
> We sincerely welcome further questions or discussions to address any remaining uncertainties or areas of interest.

---

> > ### Comment · Reviewer_HAqj · 2024-12-02
> >
> > Thank you for your response. Although you identified two key factors for the observed phenomenon, my concern remains unresolved. As I understand it, the motivation behind your method is that conflicts between different tasks during optimization lead to poor results. However, based on your findings, even when these conflicts are maximized, the performance is still better than random, suggesting that task conflicts might not be the core issue affecting multi-task optimization. This conclusion seems to weaken your motivation.

---

> ### Author Response · Authors · 2024-12-02
> **Response to the Comments**
>
> Thank you for patiently reviewing and paying attention to our work. Task conflicts remain a core issue affecting multi-task optimization. To further clarify our assertions, we present additional experimental results for NYUD-v2 and PASCAL-Context in the tables below.
>
> In these experiments, we use a **static** setting where task groups are predefined before optimization begins, and the optimization proceeds using these fixed task groups.
> In this setup, the clustering process's randomness during optimization is minimized because the task groups remain fixed throughout the optimization. Thus, we can isolate and evaluate the influence of task grouping (specifically, the cluster characteristics, which is the second factor we highlighted) on the optimization methods.
>
> We refer to this random grouping scenario under the static setting as **"Static Random"**, where the number of task groups is predefined as $N(\mathcal{M})$, and the tasks are randomly selected without any changes during optimization. When the randomness in the clustering process across batches is eliminated, the heterogeneous setting serves as a lower bound for the random scenario. Our method, which dynamically clusters homogeneous tasks, consistently outperforms other grouping strategies.
>
> We also emphasize that the performance gap resulting from different grouping strategies is significant. (Please refer to Table 3 of the main paper for comparison.) It shows that **task conflicts remain the core issue affecting multi-task optimization.**
>
> Since task relations dynamically fluctuate during optimization, both the randomness in task grouping and the criteria for forming task groups are critical factors in evaluating our method. We hope that the experiments conducted in the dynamic setting (as addressed in Answer to Q3) and the static setting (as shown in the tables below) address your questions.
>
>
> - Comparison of Various Grouping Strategies on NYUD-v2.
>
> |Task|Semseg (mIoU $\uparrow$)|Depth (RMSE $\downarrow$)|Normal (mErr $\downarrow$)|Edge (odsF $\uparrow$)| $\triangle_m$ (% $\uparrow$)|
> |:-|:-:|:-:|:-:|:-:|:-:|
> |Heterogeneous|38.12|0.620|24.59|52.20|-5.14
> |Static Random ($N(\mathcal{M})$=2)|39.31|0.630|24.36|52.20|-4.50
> |Static Random ($N(\mathcal{M})$=3)|38.29|0.634|24.29|52.60|-5.05
> |Ours|**40.02**|**0.618**|**24.09**|**53.90**|**-2.58**
>
>
>
> - Comparison of Various Grouping Strategies on PASCAL-Context.
>
> |Task|Semseg (mIoU $\uparrow$)|Parsing (mIoU $\uparrow$)|Saliency (maxF $\uparrow$)|Normal (mErr $\downarrow$)|Edge (odsF $\uparrow$)|$\triangle_m$ (% $\uparrow$)|
> |:-|:-:|:-:|:-:|:-:|:-:|:-:|
> |Heterogeneous|66.45|56.25|82.12|17.50|33.80|-9.89
> |Static Random ($N(\mathcal{M})$=2)|67.35|56.71|82.43|17.35|34.50|-8.91
> |Static Random ($N(\mathcal{M})$=3)|66.58|56.23|82.21|17.51|34.80|-9.42
> |Ours|**68.14**|**57.15**|**82.52**|**17.19**|**39.50**|**-6.24**
>
>
> We greatly appreciate your patience. If you permit, we look forward to any further questions or discussions.

---

> ### Author Response · Authors · 2024-12-02
> **Additional Response to the Comments**
>
> We kindly ask you to consider the distinction between static and dynamic settings. In a static setting, where predefined task clusters are used without change during optimization, the randomness caused by the formation of task sets during optimization is eliminated. As a result, grouping heterogeneous tasks serves as a lower bound compared to a random scenario. However, in a dynamic setting, heterogeneous task grouping does not serve as a lower bound because additional randomness exists as the formation of task clusters fluctuates during optimization.
> We validate our method in both settings and demonstrate significant performance improvements in each. This highlights that the grouping strategy, which depends on task conflicts, is one of  key factors for optimizing multi-task learning.
>
> We sincerely appreciate your patience and, if permitted, look forward to any further questions or discussions.

---

> > ### Comment · Reviewer_HAqj · 2024-12-03
> >
> > Thank you for your detailed response and comprehensive experiments. Your reply addressed most of my concerns. Although I still have some doubts about the results under dynamic random, I acknowledge the contribution of the paper. Therefore, I decided to raise my score.

---

> > > ### Author Response · Authors · 2024-12-03
> > > **Response to the comment**
> > >
> > > We sincerely appreciate your generous evaluation of our work and the improved score. We deeply regret any doubts caused by our insufficient explanation, particularly regarding the random scenario. Nonetheless, we greatly value your constructive suggestions and acknowledgment of our contributions.

---

### Official Review · Reviewer_akRc · 2024-10-31

**Soundness:** 3
**Presentation:** 3
**Contribution:** 3
**Rating:** 6
**Confidence:** 4

**Summary:**

This paper discusses the challenges of multi-task learning, where training multiple tasks together in one architecture can lead to negative transfer or performance degradation. Traditional solutions focus on optimizing shared parameters but neglect task-specific ones. The proposed solution involves grouping tasks selectively and updating them in each batch, along with an algorithm that adapts to determine effective task grouping. The concept of proximal inter-task affinity is introduced to track task relations during optimization. This approach is said to improve multi-task performance by enhancing the learning of task-specific parameters and is shown to outperform previous methods, being scalable to different architectures and task numbers.

**Strengths:**

1. This paper investigates an important problem of multi-task learning.
2. This paper is well-written and easy to follow.
3.  Realizing that traditional solutions focus on optimizing shared parameters but neglect task-specific ones, the authors delve into the concept of proximal inter-task affinity, making this paper well-motivated.
4.  The proposed method is new to me and gives a fresh perspective to further improve the performance of MTL.
5. This approach is said to improve multi-task performance by enhancing the learning of task-specific parameters and is shown to outperform previous methods, being scalable to different architectures and task numbers.

**Weaknesses:**

1. The task grouping result in Figure 3c seems out of converge. Will the number of groups further increase as the iteration becomes larger?
2. Why Nash-MTL is not reported in Table 2?
3. In the theoretical analysis (Section 4), the authors explain how this sequential update strategy can improve multi-task performance from an optimization standpoint. What about the generalization standpoint? I think the generalization of a model is more important.
4. In real-world applications, a typical MTL problem may have only a few tasks (e.g., 3). Will the proposed method work in such a circumstance? What is the task grouping result if there are only three tasks?

**Questions:**

Please refer to the weaknesss section.

---

> ### Author Response · Authors · 2024-11-24
> **Rebuttal**
>
> Thank you for your time and effort in reviewing our paper. We are particularly grateful for your positive feedback and the insightful, challenging questions that have helped improve our work. We have addressed some of your concerns in the revised version of the paper and welcome any further feedback or discussion.
> ### **Answer to W1**
> Thank you for your insightful and thought-provoking question. The number of task groups is determined by the proximal inter-task affinity, which is tracked throughout the optimization process. This affinity reflects the relations between tasks as training progresses. From an optimization perspective, as training advances, the task-specific gradients for shared parameters tend to diverge because each task has different objectives. This leads to a decrease in the proximal inter-task affinity, resulting in an increase in the number of task groups.
>
> However, simply increasing the number of training iterations will not result in more task groups, as there is an upper limit to the number of task groups. A similar limit is observed in Figure 3c, where the number of task groups fluctuates during optimization but never exceeds 4. Additionally, our experimental observations show a trade-off between task losses and performance as convergence is approached, suggesting that there is no further capacity for learning task-specific information that would alleviate conflicts among heterogeneous tasks in shared space. Therefore, we think the upper bound on the number of task groups is closely related to the capacity of task-specific parameters for each task, though predicting this upper bound theoretically can be quite challenging. When the number of iterations is increased without changing other conditions, the graph in Figure 3c simply stretches along the x-axis. On the other hand, incorporating additional task-specific parameters would enable the model to handle a more heterogeneous set of tasks, potentially leading to more task groups.
>
>
> ### **Answer to W2**
> In our experiments, Nash-MTL fails to converge on the Taskonomy benchmark when using ViT, resulting in very poor performance, which we denote by the dashed lines in the table. However, it successfully converges on the NYUD-v2 and PASCAL-Context datasets, with particularly promising results on NYUD-v2. We guess that the increasing number of tasks might hinder convergence, a similar issue that was also observed with MGDA.
>
> ### **Answer to W3**
> Thank you for your insightful question. From a generalization perspective, the key issue that might be addressed in this work is how tasks should be grouped and optimized to improve generalization performance. Different viewpoints exist on this matter, depending on the target domain we aim to optimize for.
>
> In conventional optimization settings, where the source domain (used for training) and the target domain (used for testing) have similar data distributions, grouping similar tasks typically reduces multi-task loss and enhances generalization. This is demonstrated in Theorems 1 and 2 of the paper. However, from a domain generalization perspective, grouping heterogeneous tasks may be more advantageous for improving generalization performance.
>
> In this work, we focus on the scenario where the source and target domains share similar distributions, a common assumption in conventional multi-task optimization. As experimental evidence, we present an ablation study on different task grouping strategies and their impact on generalization performance, as shown in the table below. Specifically, these strategies include randomly grouping tasks with a predefined number ($N(\mathcal{M})$), grouping heterogeneous tasks, and grouping homogeneous tasks (our approach). The results clearly demonstrate that grouping homogeneous task sets yields better generalization performance under the proposed settings.
>
> - Comparison of Various Grouping Strategies on Taskonomy.
>
> |Task|DE|DZ|EO|ET|K2|K3|N|C|R|S2|S2.5|$\triangle_m$ (% $\uparrow$)|
> |:-|:-:|:-:|:-:|:-:|:-:|:-:|:-:|:-:|:-:|:-:|:-:|:-:|
> |Heterogeneous|0.0172|0.0176|0.1252|0.1741|0.1700|0.0920|0.2475|0.7781|0.1743|0.1849|0.1660|-3.10
> |Random ($N(\mathcal{M})$=2)|0.0177|0.0180|0.1259|0.1741|0.1707|0.0923|0.2662|0.7807|0.1757|0.1871|0.1617|-4.24
> |Random ($N(\mathcal{M})$=3)|0.0172|0.0177|0.1250|0.1741|0.1703|0.0920|0.2619|0.7754|0.1749|0.1866|0.1607|-3.35
> |Random ($N(\mathcal{M})$=4)|0.0183|0.0187|0.1277|0.1746|0.1706|0.0936|0.2812|0.7841|0.1804|0.1882|0.1636|-6.12
> |Random ($N(\mathcal{M})$=5)|0.0186|0.0184|0.1274|0.1747|0.1708|0.0935|0.3150|0.7842|0.1800|0.1888|0.1640|-7.17
> |Random ($N(\mathcal{M})$=6)|0.0208|0.0209|0.1349|0.1750|0.1721|0.0961|0.3334|0.8222|0.1976|0.1935|0.1703|-13.20
> |Ours|0.0167|0.0169|0.1228|0.1739|0.1695|0.0910|0.2344|0.7600|0.1691|0.1836|0.1571|-0.64

---

> ### Author Response · Authors · 2024-11-24
> **Rebuttal (Part 2)**
>
> ### **Answer to W4**
> Although we have tested our algorithm on a large number of tasks to validate the scalability of the proposed methods, it is seamlessly applicable to case we have at least 2 tasks. For example, we validate our methods on the NYUD-v2 dataset with only three tasks and compare the performance improvements and clustering frequency during optimization in the tables below. It shows meaningful performance improvements even with just three tasks.
>
> - Performance on NYUD-v2 with three tasks
>
> |Task|Semseg (mIoU $\uparrow$)|Depth (RMSE $\downarrow$)|Normal (mErr $\downarrow$)|$\triangle_m$ (% $\uparrow$)|
> |:-|:-:|:-:|:-:|:-:|
> |GD|38.69|0.635|25.61|-4.47
> |ours|39.50|0.622|24.46|-1.40
>
>
> - Clustering frequency for NYUD-v2 with three tasks
>
> | |Semseg|Depth|Normals
> |:-|:-:|:-:|:-:|
> |Semseg|1|-|-
> |Depth|0.411|1|-
> |Normals|0.330|0.276|1

---

> > ### Comment · Reviewer_akRc · 2024-12-03
> > **Response to the authors**
> >
> > Thanks for your feedback. After reading reviews from other reviewers and the authors' responses, I decided to maintain my score.

---

> > > ### Author Response · Authors · 2024-12-03
> > > **Response to the comment**
> > >
> > > We sincerely appreciate your continued interest in our work and your patience. Your constructive feedback, particularly regarding the generalization aspect of the work, is invaluable in helping us improve.

---

### Official Review · Reviewer_hDHX · 2024-11-01

**Soundness:** 3
**Presentation:** 3
**Contribution:** 3
**Rating:** 6
**Confidence:** 3

**Summary:**

The author proposes a method for addressing multi-objective problems by grouping objectives.
In the process of considering relations between tasks (objectives), the concept of inter-task affinity was introduced, but additional computation was reduced by focusing on the update of task-specific parameters.
Additionally, introducing the concept of affinity to group objectives is the author’s original idea and has been thoroughly analyzed both theoretically and experimentally.

**Strengths:**

1. Experimental Analysis
In the field of deep learning, the analysis of batch sequences has not been extensively explored. The author argues that grouping certain objectives in multi-objective problems can be significantly beneficial from a global perspective and has demonstrated this experimentally. In cases where the multi-task learning (MTL) results outperform those of single-task learning (STL), the author’s method consistently achieves the highest performance, which serves as strong empirical support for the validity of the proposed approach.

2. Logical Approach
The author’s approach to deriving proxy task affinity is reasonable.
By utilizing the loss after task parameter updates, the author effectively reduced additional computations
Also, the proposed method has been theoretically proven to remain useful.
Additionally, the explanation of the benefits of grouping derived from Theorem 2 is clearly written and easy to understand.

**Weaknesses:**

In my understanding, some questions remain regarding the actual utility of certain theoretical approaches.
The author addresses the utility of multiple objectives in a local context, but optimization in the field of deep learning is far more complex.
In practice, grouping the same classes together for optimization in classification tasks may be optimal for the currently updated classes locally; however, it is challenging to reach a global optimum.
I would like to see additional experimental evaluation on this matter.
I will leave detailed suggestions in the questions section.

**Questions:**

As I mentioned in the weaknesses section, I do not interpret the author’s theoretical analysis as indicating that the proposed multiple-objective method can be effectively solved on a global scale.
However, I am not suggesting the necessity of a stringent theoretical foundation.
I would like to see evidence that the author’s method can consistently provide an optimal point.
Demonstrating that the proposed method is robust across different batch sizes, numbers of groups, and optimization hyperparameters would effectively support its consistency and reliability.

---

> ### Author Response · Authors · 2024-11-24
> **Rebuttal**
>
> Thank you for your time and effort in reviewing our paper. The feedback provided was invaluable in refining our work. In the revised version, we address your concerns both from a theoretical perspective and through additional supporting experiments. We welcome any further feedback or discussion.
> ### **Answer to W1**
> ### The proposed theoretical analysis can be extended to a global context.
> Thank you for your valuable feedback. We appreciate your thoughtful concerns about the utility of the theoretical approaches and are grateful for your gracious evaluation of our theoretical analysis. Before addressing the experimental results suggested by the reviewer, we would like to defend some aspects of our approach.
>
> Our theoretical analysis demonstrates the utility of selective grouping updates in multi-task optimization at the batch level for simplicity of explanation. However, this approach can be extended to a global scale, as the primary advantage of task grouping updates lies in enhancing the learning of task-specific parameters. Selective updates improve optimization by enabling better learning of task-specific parameters at the batch level, as shown in the proof process of Theorem 5 (via the inequality $B_{i,j,k \rightarrow k}^{t+(m-1)/M} \leq B_{j; i,k \rightarrow k}^{t+(m-1)/M}$, which is driven by updates to task-specific parameters). Since task-specific parameters exclusively influence the outputs of their respective tasks, the reasoning behind our batch-level analysis (Theorem 5) naturally extends to the global scale for all tasks, from t=1 to T and m=1 to $M$.
>
> However, as the reviewer pointed out, this theoretical analysis alone may not be sufficient to fully establish the utility of the proposed methods. Therefore, we focus on demonstrating the robustness of the algorithm under varying batch sizes and affinity decay rates ($\beta$), which is important hyperparmeter in our optimization. In our approach, we do not explicitly determine the number of groups, as it is dynamically determined by the proximal inter-task affinity we track. So, we demonstrate the superiority of our method compared to other grouping strategies across different grouping scenarios with varying numbers of groups.
>
> #### **Additional Experiments**
> We compare our method with single-gradient descent (GD) to evaluate how robustly it improves multi-task performance across varying batch sizes. The proposed optimization method consistently demonstrates performance improvements ($\triangle_m$ (% $\uparrow$)) of 5.27%, 5.71%, and 6.13% across different batch sizes, outperforming previous optimization methods in terms of performance gains.
>
>
> - Results on Taskonomy with varying batch sizes using ViT-B (batch sizes in brackets).
>
> |Task|DE|DZ|EO|ET|K2|K3|N|C|R|S2|S2.5|$\triangle_m$ (% $\uparrow$)|
> |:-|:-:|:-:|:-:|:-:|:-:|:-:|:-:|:-:|:-:|:-:|:-:|:-:|
> |Single Task|0.0183|0.0186|0.1089|0.1713|0.1630|0.0863|0.2953|0.7522|0.1504|0.1738|0.1530|-
> |GD(4)|0.0208|0.0214|0.1323|0.1747|0.1723|0.0952|0.2768|0.8214|0.1936|0.1921|0.1677|-10.88
> |Ours(4)|0.0185|0.0190|0.1273|0.1741|0.1709|0.0928|0.2739|0.7957|0.1809|0.1888|0.1632|-6.19
> |GD(8)|0.0188|0.0197|0.1283|0.1745|0.1718|0.0933|0.2599|0.7911|0.1799|0.1885|0.1631|-6.35
> |Ours(8)|0.0167|0.0169|0.1228|0.1739|0.1695|0.0910|0.2344|0.7600|0.1691|0.1836|0.1571|-0.64
> |GD(16)|0.0172|0.0180|0.1248|0.1742|0.1711|0.0920|0.2280|0.7641|0.1706|0.1848|0.1589|-1.94
> |Ours(16)|0.0153|0.0154|0.1186|0.1737|0.1682|0.0893|0.1967|0.7334|0.1581|0.1780|0.1516|+4.19
>
> We examine the influence of the affinity decay rate $\beta$, an important hyperparameter in the proposed optimization method. The results demonstrate that the proposed optimization method consistently improves performance across varying $\beta$ values, reducing the need for extensive hyperparameter tuning in real-world applications.
>
> - Results on Taskonomy with different affinity decay rates $\beta$.
>
> |Task|DE|DZ|EO|ET|K2|K3|N|C|R|S2|S2.5|$\triangle_m$ (% $\uparrow$)|
> |:-|:-:|:-:|:-:|:-:|:-:|:-:|:-:|:-:|:-:|:-:|:-:|:-:|
> |Single Task|0.0183|0.0186|0.1089|0.1713|0.1630|0.0863|0.2953|0.7522|0.1504|0.1738|0.1530|-
> |GD|0.0188|0.0197|0.1283|0.1745|0.1718|0.0933|0.2599|0.7911|0.1799|0.1885|0.1631|-6.35
> |$\beta$=0.0001|0.0165|0.0168|0.1224|0.1739|0.1693|0.0907|0.2304|0.7581|0.1683|0.1831|0.1571|-0.18
> |$\beta$=0.001|0.0167|0.0169|0.1228|0.1739|0.1695|0.0910|0.2344|0.7600|0.1691|0.1836|0.1571|-0.64
> |$\beta$=0.01|0.0167|0.0171|0.1232|0.1739|0.1698|0.0912|0.2362|0.7623|0.1705|0.1834|0.1576|-1.01
> |$\beta$=0.1|0.0167|0.0171|0.1231|0.1739|0.1695|0.0912|0.2355|0.7631|0.1697|0.1831|0.1575|-0.87

---

> ### Author Response · Authors · 2024-11-24
> **Rebuttal (Part 2)**
>
> We present results comparing different task grouping strategies in the tables below. These strategies include randomly grouping tasks with a predefined number ($N(\mathcal{M})$), grouping heterogeneous tasks, and grouping homogeneous tasks (our approach). The results clearly show that our method achieves superior performance compared to the other grouping scenarios.
>
> - Comparison of Various Grouping Strategies on Taskonomy.
>
> |Task|DE|DZ|EO|ET|K2|K3|N|C|R|S2|S2.5|$\triangle_m$ (% $\uparrow$)|
> |:-|:-:|:-:|:-:|:-:|:-:|:-:|:-:|:-:|:-:|:-:|:-:|:-:|
> |Heterogeneous|0.0172|0.0176|0.1252|0.1741|0.1700|0.0920|0.2475|0.7781|0.1743|0.1849|0.1660|-3.10
> |Random ($N(\mathcal{M})$=2)|0.0177|0.0180|0.1259|0.1741|0.1707|0.0923|0.2662|0.7807|0.1757|0.1871|0.1617|-4.24
> |Random ($N(\mathcal{M})$=3)|0.0172|0.0177|0.1250|0.1741|0.1703|0.0920|0.2619|0.7754|0.1749|0.1866|0.1607|-3.35
> |Random ($N(\mathcal{M})$=4)|0.0183|0.0187|0.1277|0.1746|0.1706|0.0936|0.2812|0.7841|0.1804|0.1882|0.1636|-6.12
> |Random ($N(\mathcal{M})$=5)|0.0186|0.0184|0.1274|0.1747|0.1708|0.0935|0.3150|0.7842|0.1800|0.1888|0.1640|-7.17
> |Random ($N(\mathcal{M})$=6)|0.0208|0.0209|0.1349|0.1750|0.1721|0.0961|0.3334|0.8222|0.1976|0.1935|0.1703|-13.20
> |Ours|0.0167|0.0169|0.1228|0.1739|0.1695|0.0910|0.2344|0.7600|0.1691|0.1836|0.1571|-0.64

---

> > ### Comment · Reviewer_hDHX · 2024-11-26
> >
> > Thank you for the detailed additional explanations. The robustness of performance against hyperparameter changes and comparisons with other grouping methods sufficiently support the significance of the proposed optimization approach. One remaining question concerns the batch size used in your method, which appears to be relatively small. Do you have any comparative experimental results for larger batch sizes?

---

> ### Author Response · Authors · 2024-11-28
> **Rebuttal**
>
> We sincerely appreciate your response and your continued interest in our work. We address your remaining concerns below, supplemented with additional experiments.
>
> ---
>
> ### **One remaining question concerns the batch size used in your method, which appears to be relatively small. Do you have any comparative experimental results for larger batch sizes?**
>
> As you pointed out, most multi-task optimization research, including our work, typically employs smaller batch sizes compared to single-task domains. This is mainly due to the significant memory and computational costs associated with optimizing multi-task networks as the number of tasks increases.
>
> In our experiments, we used batch sizes similar to those in previous studies within this field. However, your concern about evaluating the proposed approach with larger batch sizes is both reasonable and valuable for further assessing our work. Therefore, we have conducted additional experiments to evaluate multi-task performance with larger batch sizes, as shown in the tables below. With increasing batch sizes, our method consistently demonstrates better performance compared to baselines. Moreover, larger batch sizes further enhance multi-task performance as they contribute to more stable tracking of proximal inter-task affinity during optimization (The performance gap increases even further by 7.45% with a batch size of 128). As highlighted in Table 1 of the main paper, our optimization method has a complexity of $\mathcal{O}(1)$, in contrast to other optimization methods that require $\mathcal{O}(\mathcal{K})$ memory. This makes other methods prohibitive for large batch sizes.
>
>
> - Results on Taskonomy with varying batch sizes using ViT-B (batch sizes in brackets).
>
> |Task|DE|DZ|EO|ET|K2|K3|N|C|R|S2|S2.5|$\triangle_m$ (% $\uparrow$)|
> |:-|:-:|:-:|:-:|:-:|:-:|:-:|:-:|:-:|:-:|:-:|:-:|:-:|
> |GD(64)|0.0164|0.0164|0.1183|0.1738|0.1691|0.0893|0.1749|0.7235|0.1552|0.1759|0.1486|+4.34
> |Ours(64)|0.0151|0.0139|0.1105|0.1732|0.1648|0.0880|0.1598|0.6946|0.1450|0.1706|0.1397|+9.57
> |GD(128)|0.0163|0.0157|0.1151|0.1737|0.1681|0.0879|0.1576|0.7082|0.1490|0.1715|0.1437|+6.85
> |Ours(128)|0.0124|0.0125|0.1071|0.1732|0.1641|0.0845|0.1415|0.6799|0.1361|0.1623|0.1346|+14.30
>
>
> We sincerely welcome any additional questions or discussions you may have.

---

> > ### Author Response · Authors · 2024-12-03
> > **Comments for reviewer**
> >
> > We appreciate your constructive feedback and thoughtful opinions on our work, as your suggestions are invaluable in helping us make further improvements. To address your comments, we conducted additional experiments using a relatively larger batch size, although we are unsure if we have fully addressed your concerns. Given the time limit, we hope our additional comments have addressed your concerns effectively, and we sincerely thank you for the effort and care you have devoted to reviewing our work.

---

### Official Review · Reviewer_KUiJ · 2024-11-04

**Soundness:** 3
**Presentation:** 2
**Contribution:** 2
**Rating:** 6
**Confidence:** 3

**Summary:**

This paper presents a multi-task learning method that adaptively groups tasks based on proximal inter-task affinity and then sequentially updates each group. It provides a theoretical explanation of the benefits of sequentially updating task groups and the role of incorporating task-specific parameters in reducing conflicts. Experimental results demonstrate the method's superior performance across various benchmarks.

**Strengths:**

1. Solid analysis from a theoretical perspective: The paper provides theoretical insights to explain the effectiveness of the proposed method, including (i) the benefits of sequential updating of groups, and (ii) the role of incorporating task-specific parameters in reducing conflicts.

2. The paper is well-written and well-organized.

**Weaknesses:**

1. Based on proximal inter-task affinity, what principle do we use for task grouping? Discussion on other principles should be included. For example, in [1], they use the Fisher Information Matrix, grouping the most heterogeneous tasks to mitigate conflicts.

2. The motivation for introducing proximal inter-task affinity: After reading Appendix A.1, I still find it difficult to understand the motivation for introducing proximal inter-task affinity.

3. Sequential learning on tasks [1], domains [3,4], and mini-batches [2] for alignment has been studied previously. It would be beneficial to compare the different theoretical perspectives between the proposed method and these prior studies.

[1] Task Groupings Regularization: Data-Free Meta-Learning with Heterogeneous Pre-trained Models, ICML 2024
[2] On the Origin of Implicit Regularization in Stochastic Gradient Descent, ICLR 2021
[3] Sequential Learning for Domain Generalization, ECCV 2022
[4] Gradient Matching for Domain Generalization, ICLR 2022

**Questions:**

Please see the weaknesses.

---

> ### Author Response · Authors · 2024-11-24
> **Rebuttal**
>
> Thank you for your time and effort in reviewing our paper. The perspectives you provided have been incredibly helpful in refining our work. In the revised version, we address the points raised and highlight the key differences between our work and previous studies. We welcome any further feedback or discussion.
> ### **Answer to W1, W3**
> ### Discussion on Task Grouping Strategy and Theoretical Differences with Domain Generalization Approaches.
> Thank you for your insightful questions about our work. Task grouping is a critical aspect of both multi-task optimization and multi-task domain generalization. To enhance our discussion, we have included relevant works suggested by the reviewer in our paper (Additional Related Works). The key distinction between our work and these studies is that they focus on inter-task relations from a domain generalization perspective.
>
> In particular, [1] proposes grouping heterogeneous tasks to regularize them, promoting the learning of more generalized features across domain shifts. [2] explores generalization strategies at the mini-batch level. [3] addresses diverse domain shift scenarios by incorporating all possible sequential domain learning paths to generalize features for unseen domains. [4] focuses on generalization to unseen domains by reducing dependence on specific domains through inter-domain gradient matching.
>
> The objectives of conventional multi-task optimization and domain generalization differ fundamentally. Conventional multi-task optimization typically assumes that the source and target domains share similar data distributions, whereas domain generalization focuses on scenarios with significant domain shifts. This distinction leads to different approaches in utilizing task relations to achieve their respective goals. As demonstrated in Theorems 1 and 2 of our work, in multi-task optimization, simultaneously updating heterogeneous tasks with low task affinity results in suboptimal optimization and increased loss compared to updating similar task sets with high task affinity. This aligns with findings from prior multi-task optimization studies introduced in the related works section. In contrast, domain generalization uses task sets as a tool to extract generalized features that can be applied across various unseen domains. Overfitting to similar tasks can degrade performance on unseen domains, making it beneficial to use heterogeneous tasks as a form of regularization.
>
> Thus, in conventional multi-task optimization settings, it is more effective to group homogeneous tasks together. To support this with experimental evidence, we present results comparing different task grouping strategies in the tables below. These strategies include randomly grouping tasks with a predefined number ($N(\mathcal{M})$), grouping heterogeneous tasks, and grouping homogeneous tasks (our approach). The results clearly demonstrate that grouping homogeneous task sets achieves superior performance under the proposed settings.
>
>
> - Comparison of Various Grouping Strategies on Taskonomy.
>
> |Task|DE|DZ|EO|ET|K2|K3|N|C|R|S2|S2.5|$\triangle_m$ (% $\uparrow$)|
> |:-|:-:|:-:|:-:|:-:|:-:|:-:|:-:|:-:|:-:|:-:|:-:|:-:|
> |Heterogeneous|0.0172|0.0176|0.1252|0.1741|0.1700|0.0920|0.2475|0.7781|0.1743|0.1849|0.1660|-3.10
> |Random ($N(\mathcal{M})$=2)|0.0177|0.0180|0.1259|0.1741|0.1707|0.0923|0.2662|0.7807|0.1757|0.1871|0.1617|-4.24
> |Random ($N(\mathcal{M})$=3)|0.0172|0.0177|0.1250|0.1741|0.1703|0.0920|0.2619|0.7754|0.1749|0.1866|0.1607|-3.35
> |Random ($N(\mathcal{M})$=4)|0.0183|0.0187|0.1277|0.1746|0.1706|0.0936|0.2812|0.7841|0.1804|0.1882|0.1636|-6.12
> |Random ($N(\mathcal{M})$=5)|0.0186|0.0184|0.1274|0.1747|0.1708|0.0935|0.3150|0.7842|0.1800|0.1888|0.1640|-7.17
> |Random ($N(\mathcal{M})$=6)|0.0208|0.0209|0.1349|0.1750|0.1721|0.0961|0.3334|0.8222|0.1976|0.1935|0.1703|-13.20
> |Ours|0.0167|0.0169|0.1228|0.1739|0.1695|0.0910|0.2344|0.7600|0.1691|0.1836|0.1571|-0.64
>
>
>
> Additionally, a notable difference between previous works and ours is that prior studies mainly focus on static settings, where inter-task relations remain fixed during learning. In contrast, our approach continuously tracks the fluctuating inter-task relations during optimization and leverages them for further optimization. The concept of proximal inter-task affinity enables this continuous tracking to occur concurrently during the optimization process.

---

> > ### Author Response · Authors · 2024-11-24
> > **Rebuttal (Part 2)**
> >
> > ### **Answer to W2**
> > The main challenge in implementing selective task group updates for multi-task optimization lies in concurrently tracking task relations and effectively utilizing them during optimization.
> >
> > The inter-task affinity between tasks is represented as:
> > $$\mathcal{A}^t_{i\rightarrow k} = 1- \frac{L_k(z^t, \Theta_{s|i}^{t+1}, \Theta_k^t)}{L_k(z^t, \Theta_{s}^{t}, \Theta_k^t)}$$
> >
> > The proximal inter-task affinity between tasks is represented as:
> > $$\mathcal{B}^t_{G\rightarrow k} = 1- \frac{L_k(z^t, \Theta_{s|G}^{t+1}, \Theta_k^{t+1})}{L_k(z^t, \Theta_{s}^{t}, \Theta_k^t)}$$
> >
> > In conventional multi-task optimization settings, both $\Theta_s^t$ and $\Theta_k^t$ are updated concurrently. However, $\mathcal{A}^t_{i\rightarrow k}$, which tracks pairwise inter-task relations, does not account for updates to the task-specific parameters $\Theta_k^t$. Specifically, after updating the shared parameters from $\Theta_s^t$ to $\Theta_{s|i}^{t+1}$, it requires revisiting past optimization states $\Theta_s^t$ to compute the relations for other tasks or to proceed with further updates of both $\Theta_s^t$ and $\Theta_k^t$. This process introduces significant computational overhead, making it challenging to directly integrate inter-task affinity into multi-task optimization.
> >
> > In contrast, proximal inter-task affinity expands the scope of inter-task relations to include groups of tasks and integrates updates to task-specific parameters. This approach enables efficient tracking of task relations during the optimization process without the need to revisit previous states. By leveraging proximal inter-task affinity, task relations can be seamlessly incorporated into optimization, making the process computationally efficient and well-suited for fast convergence—an essential requirement in optimization.

---

> ### Author Response · Authors · 2024-12-02
> **Rebuttal (Part 3)**
>
> The core novelty of our work lies in incorporating inter-task affinity into multi-task optimization and demonstrating its role in facilitating selective group updates through the proposed grouping strategy to enhance multi-task performance. The concept of proximal inter-task affinity serves as a technical tool, enabling the concurrent tracking of task affinities and seamless multi-task optimization.
>
> We greatly value your insights and look forward to engaging in further discussions. Thank you for your thoughtful feedback!

---

> > ### Comment · Reviewer_KUiJ · 2024-12-03
> >
> > Thanks for your response. I have raised my score to 6.

---

> > > ### Author Response · Authors · 2024-12-03
> > > **Response to the comment**
> > >
> > > We sincerely appreciate your generous evaluation of our work and the improved score. Your constructive suggestions and insightful questions are incredibly helpful in guiding us to enhance our work further.

---

### Author Response · Authors · 2024-11-24
**Reply to All Reviewers**

We greatly appreciate all the reviewers for their thorough review and constructive suggestions. Their feedback has been invaluable in guiding us to refine our work. We have addressed all the reviewers' concerns in the final version, and the key changes are summarized below:

1. **Interpretation of Our Work in the Generalization Perspective (KUiJ, akRc):**
   We have incorporated a generalization perspective in our work, comparing it with previous approaches. Specifically, we discuss the differences between multi-task optimization and domain generalization, particularly regarding which grouping strategy is beneficial for each respective goal. We include comparisons with previous multi-task domain generalization approaches in the theoretical perspective and present additional experiments showing which grouping strategy is advantageous for multi-task optimization in the revised version.


2. **Robustness of Optimization on a Global Scale and Under Varying Conditions (hDHX, akRc, HAqj):**
   We extend our theoretical analysis to the global level by further developing batch-level analysis. We also prove the robustness of our optimization across different conditions, including variations in batch size, decay rate ($\beta$), and grouping strategy. Furthermore, we include a three-task setting to demonstrate that our algorithm can boost multi-task performance even with fewer tasks.

3. **Grouping Strategy and Its Effects on Multi-task Performance (KUiJ, akRc, HAqj):**
   We present an ablation study on how different grouping strategies impact multi-task performance and generalization. We compare our method, which groups homogeneous tasks for optimization, with random grouping, predefined grouping, and heterogeneous grouping scenarios. Our results confirm that the proposed grouping strategy provides the best performance. These experiments have been incorporated into the revised version.

4. **Addressing Minor Errors (HAqj):**
   We have corrected typos and reworded statements that could lead to misunderstandings, improving the readability and clarity of the paper.

---

### Meta-Review · Area_Chair_1f2r · 2024-12-19

**Metareview:**

The paper address the task of multi-task learning where negative transfer is a well known issues. The paper proposes a method to group tasks and then update each group. The process of grouping is based on the notion of proximal task affinity. Most reviewers agreed that the paper was well written and investigates an important problem statement. The  the questions raised by the reviewers mainly fell into to themes of generalization and clarification of aspects of the theorem. The authors rebuttal addressed these issues and all reviewers are unanimous in their final recommendation. Hence, this paper should be accepted to ICLR.

**Additional Comments On Reviewer Discussion:**

After the rebuttal, both Reviewer KUiJ and  Reviewer HAqj raised their scores. There was a robust discussion period with interaction from all the reviewers and the authors.

---

### Decision · Program_Chairs · 2025-01-22

Accept (Poster)